# Towards Auto-Regressive Next-Token Prediction: In-context learning Emerges from Generalization

**Zixuan Gong**[*], **Xiaolin Hu**[*], **Huayi Tang**, **Yong Liu**[†]
Gaoling School of Artificial Intelligence
Renmin University of China
Beijing, China
`{zxgong,xiaolinhu,huayitang,liuyonggsai}@ruc.edu.cn`

## Abstract

Large language models (LLMs) have demonstrated remarkable in-context learning (ICL) abilities. However, existing theoretical analysis of ICL primarily exhibits two limitations: **(a) Limited *i.i.d.* Setting.** Most studies focus on supervised function learning tasks where prompts are constructed with *i.i.d.* input-label pairs. This *i.i.d.* assumption diverges significantly from real language learning scenarios where prompt tokens are interdependent. **(b) Lack of Emergence Explanation.** Most literature answers *what* ICL does from an implicit optimization perspective but falls short in elucidating *how* ICL emerges and the impact of pre-training phase on ICL. In our paper, to extend (a), we adopt a more practical paradigm, *auto-regressive next-token prediction (AR-NTP)*, which closely aligns with the actual training of language models. Specifically, within AR-NTP, we emphasize prompt token-dependency, which involves predicting each subsequent token based on the preceding sequence. To address (b), we formalize a systematic pre-training and ICL framework, highlighting the layer-wise structure of sequences and topics, alongside a two-level expectation. In conclusion, we present data-dependent, topic-dependent and optimization-dependent PAC-Bayesian generalization bounds for pre-trained LLMs, investigating that *ICL emerges from the generalization of sequences and topics*. Our theory is supported by experiments on numerical linear dynamic systems, synthetic GINC and real-world language datasets.

## 1 Introduction

Large language models (LLMs) have exhibited intriguing emergent capabilities in in-context learning (ICL) (Brown et al., 2020), which allows effective predictions on downstream tasks only based on a short context without any parameter fine-tuning (Black et al., 2022; Rae et al., 2021). Since then, more scholars have increasingly focused on the intrinsic mechanisms of ICL (Chan et al., 2022; Garg et al., 2022; Von Oswald et al., 2023), aiming to gain a better understanding of LLMs.

For simple supervised function learning tasks, the analysis framework for ICL has been well-established, where independently and identically distributed (*i.i.d.*) input-label pairs are stacked into a prompt so that the model directly gives the predicted label for query input without parameter updates. Following that, empirically, Garg et al. (2022) demonstrates that pre-trained LLMs can approximate linear functions with a performance that is nearly equivalent to the least squares estimator. Theoretically, many studies reveal that ICL implicitly employs optimization algorithms. Among these, a prominent viewpoint demonstrates that pre-trained LLMs performing ICL is equivalent to mimicking a single step of gradient descent on linear regression tasks (Ahn et al., 2024; Akyürek et al., 2022; Dai et al., 2023; Nichani et al., 2024; Von Oswald et al., 2023; Zhang et al., 2023a). However, there are two main limitations in this literature: (a) Limited *i.i.d.* Setting. The *i.i.d.* assumption in prompt tokens is potentially strong and unrealistic in language tasks where prompt

---

[*]Equal contribution.
[†]Corresponding author.

tokens are dependent, making it challenging to easily extend the aforementioned analysis framework for supervised learning tasks to language modeling. (b) Lack of Emergence Explanation. Most literature answers **what** ICL does from an optimization algorithm perspective but falls short in explaining **how** pre-trained LLMs can be good enough to emerge ICL ability as well as the impact of pre-training phase on ICL. Therefore, the following fundamental questions remain relatively underexplored:

*(a) How can we model language tasks with token-dependency, going beyond the i.i.d. limitation?*

*(b) How can ICL emerge from pre-trained LLMs?*

For question (a), in extending existing work on supervised function learning, we are eager to explore research on the ***auto-regressive next-token prediction (AR-NTP)*** paradigm, which is key to the success of modern LLMs (Achiam et al., 2023; Brown et al., 2020) in practical language tasks. Specifically, there are generally successive tokens in both training sequences and ICL prompts, which are drawn from the unsupervised corpus. Through AR-NTP, each subsequent token in sequences or prompts is generated based on the preceding tokens. Drawing inspiration from the Bayesian perspective in statistical field, we utilize conditional probability distribution (Han et al., 2023; Jiang, 2023; Li et al., 2023; Wang et al., 2023; Wei et al., 2022; Wu et al., 2023; Xie et al., 2021) to theoretically model AR-NTP. In the line of Bayesian research, most view ICL as a process of implicit Bayesian inference, where the pre-trained LLM is thought to subconsciously deduce a concept while generating a prediction. These works assume tokens are generated from Hidden Markov Models. In our paper, we consider a more relaxed generation mode, AR-NTP, where each token depends on all the preceding tokens rather than just one preceding token. Consequently, our core task is to model language tasks, and the core challenge of modeling language tasks is to consider *prompt token-dependency*.

To analyze question (b), we intuitively recognize that the prompt sequence may be new or unseen and the corresponding ICL topic for the sequence is generally unknown. We desire a well-pretrained in-context learner, *i.e.*, the LLM can effectively utilize ICL to generate high-quality responses to any sequence under any topic, regardless of whether the sequence and topic are seen or unseen during pre-training. This necessitates examining population loss by considering expectations over distribution, rather than focusing solely on empirical loss. A low population loss indicates that the model possesses strong generalization ability for diverse sequences and topics, thereby facilitating the emergence of ICL. Therefore, it is natural to ***explore the origin of ICL from the perspective of measuring generalization ability***. Specifically, we formalize a systematic pre-training and ICL framework that incorporates data distribution and topic distribution, allowing us to establish the population loss with a two-level expectation. By adopting PAC-Bayesian generalization analysis techniques, we gain a clearer understanding of how ICL emerges.

Based on the above analysis, we summarize our main contributions as follows.

**1. Pre-training and ICL Framework under AR-NTP Paradigm.** Towards practical AR-NTP paradigm rather than *i.i.d.* setting, we establish a systematic pre-training and ICL framework considering layer-wise structure of sequences and topics (Section 3.1). Meanwhile, we propose two-level expectation over data and topic distribution to link pre-training and ICL phase, thereby providing well-defined population loss based on empirical loss (Section 3.2 and 3.3).

**2. ICL Emerges from Generalization.** Our theoretical results of population loss reveal that model generalization, tightly with ICL abilities, is influenced by model size, optimization iterations, pre-training data and prompt length. This further demonstrates that ICL emerges from the excellent generalization of sequences and topics (Section 4.1 and Section 4.2).

**3. Generalization Analysis.** By dealing with prompt token-dependency and employing continuous mathematical techniques such as Stochastic Differential Equation (SDE), we present data-dependent and topic-dependent, as well as optimization-dependent PAC-Bayesian generalization bounds for population loss (Section 4.1 and Section 4.2).

**4. Empirical Verification of Theory.** We perform experiments on numerical linear dynamic system, synthetic GINC and real-word language datasets (Section 5 and Appendix D). By discussing the effects of pre-training data, optimization process and prior model initialization, we verify our theoretical results and offer practical implications (Appendix E).

## 2 RELATED WORK

**Optimization Perspective on ICL.** The field of ICL in transformers has been extensively explored from various analytical perspectives. A prominent approach is to view ICL as an implicit execution of gradient descent algorithm (Akyürek et al., 2022; Von Oswald et al., 2023; Dai et al., 2023; Zhang et al., 2023a). This concept is well-illustrated (Akyürek et al., 2022; Von Oswald et al., 2023), which demonstrates that pre-trained transformers can mimic a single step of gradient descent on linear regression tasks. Huang et al. (2023); Zhang et al. (2023a) specifically provide evidence that learning linear models via gradient flow aligns with transformers learning in-context, based on optimization convergence analysis. However, all this literature falls short in explaining how LLMs develop the ability of ICL and the connection between the pre-training and ICL phases.

**Bayesian Perspective on ICL.** There is some existing work from Bayesian view enriching the understanding of ICL (Han et al., 2023; Jiang, 2023; Wang et al., 2023; Wies et al., 2023; Xie et al., 2021). Xie et al. (2021) interpreter ICL as implicit Bayesian inference, where the pre-trained LLM is seen as intuitively deducing a concept during prediction. Following Xie et al. (2021), the assumption that the pre-training distribution is a Hidden Markov Model, is relaxed in Wies et al. (2023). Further, Zhang et al. (2023a) consider the pre-training and ICL phase and assume that prior and posterior satisfy a uniform distribution. In our study, we adopt data-dependent and topic-dependent prior without relying on some predetermined distribution assumptions. A topic distribution is considered in our pre-training and ICL framework, which weakens the assumption that the pre-training topic distribution covers the ICL topic distribution in Zhang et al. (2023a) to some extent.

**From Multi-Task Learning to Meta-Learning.** Training LLMs to perform ICL can be viewed as an approach for addressing the wider tasks of meta-learning or learning-to-learn (Naik & Mammone, 1992; Schmidhuber, 1987). In pre-training phase, the LLM is trained on multiple tasks. We expect that a well-pretrained LLM serves as a good *meta-learner* possessing the ICL ability to generalize to new unseen tasks, not only as a *multi-task learner* (Radford et al., 2019). Theoretical analysis of meta-learning has received significant attention (Chua et al., 2021; Denevi et al., 2018; Ji et al., 2020; Tripuraneni et al., 2020). Drawing inspiration from the assumption of an unknown task distribution in meta-learning analysis, we establish a pre-training and ICL framework with topic/task distribution and data distribution, to describe the model's generalization ability to new test prompts and unseen topics (Details in Section 3.1). However, it is worth emphasizing that our ICL generalization analysis under AR-NTP cannot be equivalent to meta-learning generalization, since the expectation over sequence would be specially spilt into two parts due to the prompt token-dependency (Details in Section 3.3). We defer more discussion in Appendix C.

## 3 PROBLEM SETUP

In this section, for question (a), Section 3.1 establishes the pre-training and ICL framework under AR-NTP paradigm. Following that, to address question (b), Section 3.2 and 3.3, formalize the optimization objective and generalization of pre-trained LLMs, to illustrate how pre-trained LLMs can be good enough to emerge ICL ability.

**Notations.** Let $\mathbb{E}[\cdot]$ be the expectation of random variables. The KL divergence between distribution $\mu$ and $\nu$ is $D_{\mathrm{KL}}(\mu \parallel \nu) = \mathbb{E}_{\theta \sim \mu}[\log \mu(\theta)/\nu(\theta)]$ and total variation (TV) distance is $D_{\mathrm{TV}}(\mu, \nu) = 1/2 \sum_{\theta \in \Theta} |\mu(\theta) - \nu(\theta)|$. The detailed notations is shown in Appendix A Table 1.

### 3.1 AR-NTP PARADIGM AND PRE-TRAINING AND ICL FRAMEWORK

We model the practical AR-NTP paradigm in language learning tasks, where any training sequence or ICL prompt consists of successive tokens drawn from the unsupervised corpus. Within AR-NTP, each subsequent token in the sequence is generated based on the preceding tokens, referred to as prefix sequences. Thus, the data distribution of each token is represented as a conditional probability distribution in statistics. From the analysis to question (b), it's necessary to consider the impact of pre-training and formalize a systematic pre-training and ICL framework to facilitate the generalization analysis where ICL emerges. The overview of pre-training and ICL framework, including topic distribution and data distribution, is shown in Figure 1.

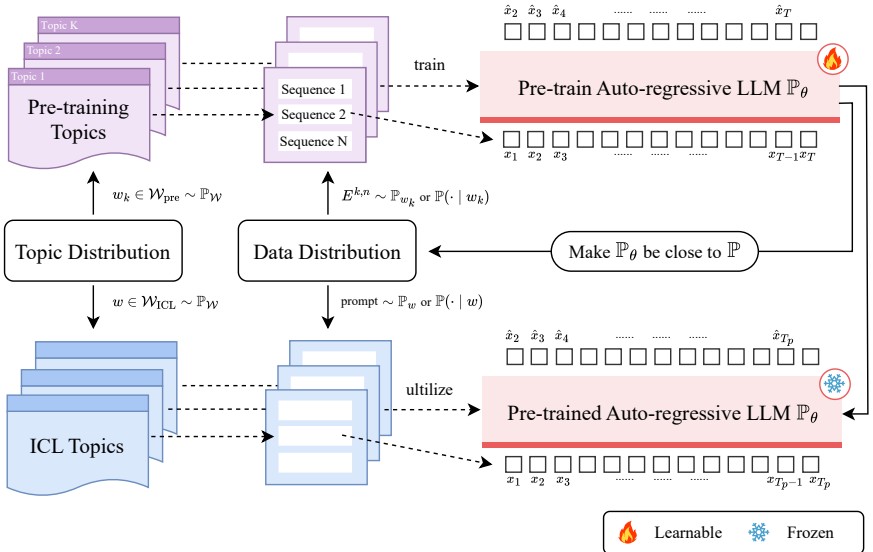

Figure 1: Overview of Pre-training and In-context Learning Framework.

**Pre-training Phase.** In pre-training phase, the training data typically encompasses sequences from various topics. To model the training process more realistically, it's essential to characterize and distinguish the training sequences and their respective topics in detail. In the following, we describe the generating process of all training sequences.

1. **Generate One Training Sequence Under a Pre-training Topic:** Under the assumption of topic distribution $\mathbb{P}_{\mathcal{W}}$ and a fixed topic $w_k \sim \mathbb{P}_{\mathcal{W}}$, the $n$-th sequence $E^{k,n}$ satisfies the true data distribution $\mathbb{P}_{w_k}$, or denoted by $\mathbb{P}(\cdot \mid w_k)$. According to the auto-regressive generating process, $(t+1)$-th token $x_{t+1}^{k,n}$ in $E^{k,n}$ is generated depending on the prefix sequence $E_t^{k,n} = \{x_1^{k,n}, x_2^{k,n}, \cdots, x_t^{k,n}\}$. It also means $x_{t+1}^{k,n} \sim \mathbb{P}(\cdot \mid E_t^{k,n}, w_k)$. When the token count reaches $T_{k,n}$, the sequence $E^{k,n} = \{(E_t^{k,n}, x_{t+1}^{k,n})\}_{t=1}^{T_{k,n}-1}$ is already formed.

2. **Generate $N_k$ Training Sequences Under a Pre-training Topic:** Repeat step 1, $N_k$ training sequences following the same data distribution $\mathbb{P}_{w_k}$ or $\mathbb{P}(\cdot \mid w_k)$, are *i.i.d.* sampled. The pre-training sequences under topic $w_k$ can be denoted by $E^k = \{E^{k,n}\}_{n=1}^{N_k}$.

3. **Generate Complete Training Sequences Under $K$ Pre-training Topics:** Considering that the set of topics used for pre-training is $\mathcal{W}_{\text{pre}}$ which contains $K$ topics, then repeat Step 2, the complete pre-training sequences can be denoted by $E = \{E^k\}_{k=1}^K = \{E^{k,n}\}_{k,n=1}^{K,N_k}$.

Note that the number of sequences for different topics ($N_k$) and sequence length ($T_{k,n}$) vary from each other. We give more discussion for $N_k$ and $T_{k,n}$ in Remark F.2. For theoretical convenience, we unify $N$ and $T$ in our main analysis. Using pre-training data $E$ containing a total of $KN$ sequences, the model gives predictions that still follow the AR-NTP methods. Then the LLM $\mathbb{P}_\theta$ parameterized by $\theta \in \Theta$ is pre-trained by establishing AR-NTP loss.

*Note:* Throughout our paper, the subscripts or superscripts $k$, $n$, and $t$ represent the topic index, sequence index and token index, respectively.

**ICL Phase.** In ICL phase, for any ICL topic $w$ that satisfies the same topic distribution $\mathbb{P}_{\mathcal{W}}$ as pre-training topics, prompt $= \{x_1, x_2, \cdots, x_{T_p}\}$ is generated from data distribution $\mathbb{P}_w$ (or $\mathbb{P}(\cdot \mid w)$). Similarly to the above generation process of pre-training sequence $E^{k,n}$, $x_t \sim \mathbb{P}(\cdot \mid \text{prompt}_{t-1}, w)$. The goal for an ICL learner is to make the prediction $\mathbb{P}_\theta(x_{T_p} \mid \text{prompt}_{T_p-1}, w)$, given by the pre-trained LLM $\mathbb{P}_\theta$, as close as possible to $\mathbb{P}(x_{T_p} \mid \text{prompt}_{T_p-1}, w)$. To test the performance of ICL on different topics, a set of ICL topics $\mathcal{W}_{\text{ICL}}$ is adopted. Note that different numbers of demonstrations may be used in standard ICL. In our theoretical modeling, we consider directly concatenating demonstrations into ICL prompts. The distinction between zero-shot ICL and few-shot ICL is reflected in the prompt length $T_p$, and our theoretical results reveal the impact of prompt length on model generalization and ICL emergence.

## 3.2 OPTIMIZATION OBJECTIVE: EMPIRICAL LOSS

Considering the pre-training phase, finite topics and finite sequences are *i.i.d.* sampled from topic distribution $\mathbb{P}_{\mathcal{W}}$ and data distribution $\mathbb{P}_{w_k}$ or $\mathbb{P}(\cdot \mid w_k)$. During the training process, for any sequence $E^{k,n}$ under topic $w_k$, each token $x_{t+1}^{k,n}$ can be predicted depending on the prefix sequence $E_t^{k,n}$, optimizing the negative log-likelihood loss $-\log \mathbb{P}_\theta(x_{t+1}^{k,n}|E_t^{k,n}, w_k)$ in practice. When with fixed the true data distribution $\mathbb{P}(\cdot \mid w_k)$, minimizing $-\log \mathbb{P}_\theta(x_{t+1}^{k,n}|E_t^{k,n}, w_k)$ is equivalent to minimize the $\log \frac{\mathbb{P}(x_{t+1}^{k,n}|E_t^{k,n}, w_k)}{\mathbb{P}_\theta(x_{t+1}^{k,n}|E_t^{k,n}, w_k)}$. It is expected that the prediction of LLM could be close to the true sequence.

Under $k$-th topic, we average the prediction loss of all tokens for the $n$-th sequence and then average this over $N$ sequences, with the definition of $L_{E^k}(\theta, w_k)$. Finally, averaging over $K$ topics, the optimization objection (empirical loss) of the pre-training phase is defined as

$$L_E(\theta, \mathcal{W}_{\text{pre}}) = \frac{1}{K} \sum_{k=1}^{K} \underbrace{\left( \frac{1}{NT} \sum_{n=1}^{N} \sum_{t=1}^{T} \log \frac{\mathbb{P}(x_{t+1}^{k,n}|E_t^{k,n}, w_k)}{\mathbb{P}_\theta(x_{t+1}^{k,n}|E_t^{k,n}, w_k)} \right)}_{L_{E^k}(\theta, w_k)}, \tag{1}$$

where $L_{E^{k,n}}(\theta, w_k) = \frac{1}{T} \sum_{t=1}^{T} \log \mathbb{P}(x_{t+1}^{k,n}|E_t^{k,n}, w_k)/\mathbb{P}_\theta(x_{t+1}^{k,n}|E_t^{k,n}, w_k)$, represents the average loss of one sequence. Define the minimum of empirical loss as

$$\hat{\theta} = \text{argmin}_\theta L_E(\theta, \mathcal{W}_{\text{pre}}). \tag{2}$$

In our theoretical analysis, LLMs perform Stochastic Gradient Descent (SGD) as an optimization algorithm to update parameters $\theta$ in order to get the minimum $\hat{\theta}$. We formalize optimization error $\epsilon_{\text{opt}}$ with the logarithmic distribution distance between the pre-trained model $\mathbb{P}_\theta$ and the ideal model $\mathbb{P}_{\hat{\theta}}$,

$$\frac{1}{KNT} \sum_{k=1}^{K} \sum_{n=1}^{N} \sum_{t=1}^{T} \left( \log \mathbb{P}_{\hat{\theta}}(x_{t+1}^{k,n}|E_t^{k,n}, w_k) - \log \mathbb{P}_\theta(x_{t+1}^{k,n}|E_t^{k,n}, w_k) \right). \tag{3}$$

## 3.3 GENERALIZATION ANALYSIS: TWO-LEVEL EXPECTATION

We expect a good ICL learner, which means that the pre-trained LLM has the ability to identify new topics and predict new sequences. As introduced in Section 3.1, we hope that the pre-trained LLM can infer unseen sequences under unseen ICL topics with the assumption of data distribution and topic distribution. Therefore, it's natural to define a two-level expectation, aiming to minimize the expected / population loss.

**The First-level Expectation over Sequence.** The first-level expectation (*i.e.* inner expectation) is taken over sequence $E^{k,n}$, indicating a sufficient number of sequences for each topic to facilitate comprehensive learning in the ideal case so that the pre-trained model can perform excellently when faced with new sequences **under seen topics**. In Equation 1, rather than using $L_{E^k}(\theta, w_k)$ with $N$ sequences, we define $L(\theta, \mathcal{W}_{\text{pre}})$ with sufficient sequences as

$$L(\theta, \mathcal{W}_{\text{pre}}) = \frac{1}{K} \sum_{k=1}^{K} \mathbb{E}_{E^{k,n}} \left[ \log \frac{\mathbb{P}(x_{t+1}^{k,n}|E_t^{k,n}, w_k)}{\mathbb{P}_\theta(x_{t+1}^{k,n}|E_t^{k,n}, w_k)} \right].$$

More concretely, the *prompt token-dependency* in AR-NTP (*i.e.* the tokens are dependently generated in sequence $E^{k,n}$) motivates that the first-level expectation $\mathbb{E}_{E^{k,n}}$ needs to be divided into two parts: expectation over each token when given prefix sequences $\mathbb{E}_{x_{t+1}^{k,n} \sim \mathbb{P}(\cdot|E_t^{k,n}, w_k)}$ and expectation over prefix sequences $\mathbb{E}_{E_t^{k,n}}$. Then combining the definition of KL divergence, it can be transformed into $\frac{1}{K} \sum_{k=1}^{K} \mathbb{E}_{E_t^{k,n}} \left[ D_{\text{KL}} \left( \mathbb{P}(\cdot \mid E_t^{k,n}, w_k) \parallel \mathbb{P}_\theta(\cdot \mid E_t^{k,n}, w_k) \right) \right]$. Using any prefix sequence $P$ to replace $E_t^{k,n}$, we simply the representation and the first-level expected loss finally becomes

$$L(\theta, \mathcal{W}_{\text{pre}}) = \frac{1}{K} \sum_{k=1}^{K} \mathbb{E}_P \left[ D_{\text{KL}} \left( \mathbb{P}(\cdot \mid P, w_k) \parallel \mathbb{P}_\theta(\cdot \mid P, w_k) \right) \right]. \tag{4}$$

**The Second-level Expectation over Topic.** The second-level expectation (*i.e.* outer expectation) is taken over topic $w_k$. A well-trained LLM with the objective of minimizing the population loss over infinite topics will demonstrate good generalization of topics, which will be directly reflected in the model's accuracy in predicting ICL prompts **under unseen topics** during the ICL phase, provided these unseen ICL topics satisfy the assumption of topic distribution. Therefore, rather than using $K$ topics in Equation 4, we define $L(\theta)$ with sufficient topics as

$$L(\theta) = \mathbb{E}_w \mathbb{E}_P \left[ D_{\mathrm{KL}} \left( \mathbb{P}(\cdot \mid P, w) \parallel \mathbb{P}_\theta(\cdot \mid P, w) \right) \right],$$

which is called population loss with two-level expectation. To specifically align to the ICL phase and test the impact of different prompt lengths, we calculate the average loss over each token, similar to pre-training, *i.e.*,

$$L(\theta) = \frac{1}{T_p} \sum_{t=1}^{T_p} \mathbb{E}_w \mathbb{E}_{\mathrm{prompt}_t} \left[ D_{\mathrm{KL}} \left( \mathbb{P}(\cdot \mid \mathrm{prompt}_t, w) \parallel \mathbb{P}_\theta(\cdot \mid \mathrm{prompt}_t, w) \right) \right]. \tag{5}$$

# 4 ICL EMERGES FROM GENERALIZATION OF PRE-TRAINED LLMS

In this section, we sequentially present Theorems for the generalization of sequences and topics. Specifically, Theorem 4.3 considers the generalization of sequences, providing the upper bound of the first-level expected loss defined in Equation 4. Theorem 4.6 further considers the generalization of topics and provides the upper bound of the two-level expected loss (*i.e.* population loss defined in Equation 5) by integrating Theorem 4.3. Thus, we answer question (b) that ICL emerges from the excellent generalization of sequences and topics.

**Summary of Challenges.** Before diving into the details of Theorems, we summarize the challenges in both modeling and theoretical proof, in comparison to previous research.

**(1) The consideration of two-level expectation.** In contrast to focusing solely on the ICL process, we model the entire process of training and utilization, aiming to mirror real-world training scenarios and explore the origin of ICL from the perspective of generalization. The consideration of two-level expectation over sequence and topic under a reasonable pre-training and ICL framework significantly amplifies our workload.

**(2) The dealment of prompt token-dependency.** Under the setting of AR-NTP, we make great efforts to address the dependency between the current token and its preceding tokens by constructing ghost sequences (see the detailed construction in Appendix G.2.1, where we summarize the proof sketch), thereby enabling the possibility of taking expectation over each token within all possible sequences. It's worth noting that such a dependency is not present in the supervised function learning tasks in other ICL research.

**(3) The connection of negative logarithm likelihood, KL divergence and TV distance.** We examine the primary optimization objective: negative logarithm likelihood. Naturally, this leads to a connection with KL divergence, thereby formalizing the expression of population loss. Furthermore, in addressing the aforementioned token-dependency, we establish connections between TV distance and the expectation over a single token when given its predecessors. Therefore, it's necessary to establish connections between the two key distribution metrics: TV distance and KL divergence (see in Lemma G.7), to obtain our final generalization bounds. The AR-NTP setup necessitates the establishment of the above series of connections, which are not considered in the previous ICL work.

## 4.1 GENERALIZATION OF SEQUENCES: THE FIRST-LEVEL EXPECTATION

Under finite $(K)$ pre-training topics, $L(\theta, \mathcal{W}_{\mathrm{pre}})$ defined in Equation 4, represents the first-level expected loss where infinite sequences per topic are utilized. It describes comprehensive learning for each pre-training topic in the ideal case so that the pre-trained model can give excellent answers for new sequences on the seen topics in ICL phase. In the following theorem, we present the upper bound of $L(\theta, \mathcal{W}_{\mathrm{pre}})$.

Based on basic notations of general PAC-Bayesian theory, in our discussion, we define the posterior distribution of model parameters as $\mu(\theta)$, which is obtained by training the LLM using $K$ topics

and $N$ sequences per topic. Define the prior distribution as $\nu(\theta)$, which is an assumed probability distribution before some evidence is taken into account. In the formal Theorem for $L(\theta, \mathcal{W}_{\text{pre}})$, we derive the KL distance between the posterior and prior in the upper bound, specifically with a data-dependent prior (Li et al., 2019). Furthermore, continuous mathematical analysis tools such as SDE are used to detail the KL divergence between posterior and data-dependent prior, which further considers the optimization algorithm. Since then, we can provide data-dependent and optimization-dependent generalization bound of the first-level expected loss.

**Data-Dependent Prior.** We employ the following method for generating a data-dependent prior (Li et al., 2019). Let $J$ include $N'$ indexes uniformly sampled from $[N]$ without replacement and $I$ is $[N] \setminus J$, splitting pre-training sequences under fixed topic $w_k$ into two parts $E_I^k$ and $E_J^k$. Under all pre-training topics, we have $E_I = \{E_I^k\}_{k=1}^K$ and $E_J = \{E_J^k\}_{k=1}^K$. The prior distribution of model parameters $\theta$ depends on the subset $E_J$, which is denoted by $\nu_J$ and the posterior distribution of $\theta$ depends on $E_I$ denoted by $\mu$. Thus, a parallel training process with $E_J$ are conducted, and after that, a data-dependent prior $\nu_J$ will be obtained. We emphasize that extracting a portion of training data to learn the prior distribution of model parameters has significant implications. Specifically, this approach allows the prior to adapt to specific features and trends in the data, enhancing the model's ability to capture and learn from these nuances. Even if we sacrifice a portion of the training data, the prior will lead to a posterior distribution that is better aligned with the actual data distribution. In high-dimensional spaces, a data-dependent prior provides a more informed starting point.

**Assumption 4.1** (Bounded Loss Function). Given fixed topic $w_k$ and prefix sequence $E_t^{k,n}$, for the true data distribution $\mathbb{P}(\cdot \mid w_k)$ (or $\mathbb{P}_{w_k}$) and pre-trained LLM $\mathbb{P}_\theta$, we have

$$\log \frac{\mathbb{P}(x_{t+1}^{k,n} \mid E_t^{k,n}, w_k)}{\mathbb{P}_\theta(x_{t+1}^{k,n} \mid E_t^{k,n}, w_k)} \leq S.$$

This assumption shows that the logarithm ratio of $\mathbb{P}(\cdot \mid w_k)$ and $\mathbb{P}_\theta$ is bounded suggesting that the learned model is expected to closely approximate the true data distribution. According to the true data distribution, the probability of $x_{t+1}^{k,n}$ tends to 1. Thus by scaling law (Kaplan et al., 2020), the training loss for specific tokens $-\log \mathbb{P}_\theta(x_{t+1}^{k,n} \mid E_t^{k,n}, w_k)$ equals to $\left(\frac{N_c}{N_{\text{param}}}\right)^{\alpha_N}$, where $N_{\text{param}}$ represents the number of parameters in the model, $N_c$ and $\alpha_N$ are constants obtained through statistical fitting. Thus, $S$ can further be measured with $\left(\frac{N_c}{N_{\text{param}}}\right)^{\alpha_N}$.

**Assumption 4.2** (Bounded Gradient). Suppose that for topic $w_k$ and model parameters $\theta_t$ at step $t$ (for any $0 \leq t \leq T'$, $T'$ is the total iteration steps), we have $\|\nabla L_{E^{k,n}}(\theta_t, w_k)\| \leq L$.

Assumption 4.2 is the classical $L$-Lipschitz continuous condition, which is widely used in generalization analysis (Elisseeff et al., 2005; Li et al., 2019). This suggests that the gradient of an average loss of one sequence (see in Equation 1) is bounded.

**Theorem 4.3** (Data-Dependent and Optimization-Dependent Generalization Bound of the First-level Expected Loss). *Let the auto-regressive LLM $\mathbb{P}_\theta$ be the empirical solution of Equation 1, and $\mathbb{P}(\cdot \mid w)$ denotes the true data distribution under topic $w$. Under Assumptions 4.1 and 4.2, for any $0 < \delta < 1$, with probability at least $1 - \delta$, the first-level expected loss with $K$ topics and infinite sequences per topic, denoted by $L(\theta, \mathcal{W}_{pre})$ (see in Equation 4), satisfies,*

$$\mathbb{E}_\mu \left[ L(\theta, \mathcal{W}_{pre}) \right] = \mathcal{O} \left\{ \sqrt{\frac{\log 1/\delta}{KNT}} + \sqrt{\frac{1}{KNT} \left( D_{\text{KL}}(\mu \parallel \nu) + \log \frac{1}{\delta} \right) - \epsilon_{opt}} \right\},$$

*then considering data-dependent prior $\nu_J$ and detailing the term $D_{\text{KL}}(\mu \parallel \nu_J)$, $L(\theta, \mathcal{W}_{pre})$ is further bounded by*

$$\mathcal{O} \left\{ \sqrt{\frac{\log 1/\delta}{K(N - N')T}} + \sqrt{\frac{1}{K(N - N')T} \left( \frac{L^2 C(\frac{1}{N_{param}}, T')}{N'} + \log \frac{1}{\delta} \right) - \epsilon_{opt}} \right\}, \quad (6)$$

*where $C(\frac{1}{N_{param}}, T') = \frac{\beta}{2} e^{8\beta S} \left( 1 - e^{-\frac{T'}{\exp(8\beta S)}} \right)$. $\epsilon_{opt}$ is the optimization error (see in Equation 3). $K$, $N (N')$ and $T$ denote the number of topics, the number of sequences per topic and the sequence length utilized in the optimization process of Equation 1. $T'$ denotes the total training iterations. $N_{param}$ denotes the number of model parameters.*

**Remark 4.4.** Theorem 4.3 reveals that when considering the first-level expectation over sequence, the expected loss achieves $\mathcal{O}\{1/\sqrt{KNT}\}$ rate. This indicates that an increase in the number of training topics ($K$), the number of sequences per topic ($N$), and the sequence length ($T$) leads to a reduction in the first-level expected loss, aligning with both intuitive understanding and empirical evidence. Furthermore, from the term $C(\frac{1}{N_{\text{param}}}, T')$, the expected loss is smaller with a larger model size (*i.e.*, larger $N_{\text{param}}$). It reveals why LLMs outperform small language models in emerging ICL, even though they adopt a similar AR-NTP paradigm. $T'$ is related to the optimization process. As $T'$ increases, $C(\beta, T')$ increases, *i.e.*, the generalization error increases. This reflects the influence of total training iterations $T'$ on testing loss, corresponding to the classical viewpoint 'train faster, generalize better' (Hardt et al., 2016; Lei & Ying, 2020; Zhang et al., 2022). We defer more detailed discussion to Appendix F.4 and proof to Appendix G.2.1 and G.2.2.

## 4.2 GENERALIZATION OF SEQUENCES AND TOPICS: TWO-LEVEL EXPECTATION

Up to now, we have analyzed the first-level expected loss with $K$ topics and infinite sequences per topic. With small first-level expected loss, the pre-trained LLM can perform excellently on the new test prompt under **seen** topics in ICL phase. In this section, we use similar techniques to further consider the second-level expectation with infinite topics, so that the pre-trained LLM with small population loss can perform well on **unseen** topics. At this moment, ICL emerges from the generalization of sequences and topics.

In the following theorem for the two-level expected loss (population loss) $L(\theta)$, similarly, we derive the KL distance between the posterior $\mu$ and prior $\nu$ in the upper bound, specifically propose a topic-dependent prior whose core idea comes from data-dependent prior (Li et al., 2019), *i.e.*, a portion of $K$ topics will be used for calculating model prior and other topics will be used for obtaining posterior. Based on SDE analysis, we detail the KL divergence between posterior and topic-dependent prior. Since then, we can provide data-dependent, topic-dependent and optimization-dependent generalization bound of the population loss.

**Topic-Dependent Prior.** We employ the following method for generating a topic-dependent prior, similar to data-dependent prior (Li et al., 2019). We split topics into two parts and let $J$ include $K'$ indexes uniformly sampled from $[K]$ without replacement and let $I$ be $[K] \setminus J$, then the total sequences are divided into $E^I = \{E^k\}_{k \in \mathcal{W}_{\text{pre}, I}}$ and $E^J = \{E^k\}_{k \in \mathcal{W}_{\text{pre}, J}}$. Assume that the posterior distribution of model parameters $\theta$ depends on $E^I$ denoted by $\mu$ and the prior distribution of $\theta$ depends on the topic subset $E^J$ denoted by $\nu_J$. A parallel training process is performed with $E^J$ based on the same LLM architecture, and after that, a topic-dependent prior $\nu_J$ will be obtained.

**Assumption 4.5** (Bounded Expected Gradient). Suppose that for topic $w_k$ and model parameters $\theta_t$ at step $t$ (for any $0 \le t \le T'$, $T'$ is the total iteration steps), we have $\|\mathbb{E}_{E^{k,n}} [\nabla L_{E^{k,n}}(\theta_t, w_k)]\| \le \sigma$.

Note that $L_{E^{k,n}}$ denotes the average loss of one sequence (Equation 1). Then $\mathbb{E}_{E^{k,n}} [\nabla L_{E^{k,n}}(\theta_t, w_k)]$ denotes the gradient averaging over all possible sequences $E^{k,n}$, therefore $\sigma$ is less than the common Lipschitz constant $L$, which bounds the gradient at individual sample points.

**Theorem 4.6** (Data-Dependent, Topic-Dependent and Optimization-Dependent Generalization Bound of the Two-level Expected Loss). *Let the auto-regressive LLM $\mathbb{P}_\theta$ be the empirical solution of Equation 1, and $\mathbb{P}(\cdot \mid w)$ is the true data distribution under topic $w$. Under Assumptions 4.1, 4.2 and 4.5, for any $0 < \delta < 1$, with probability at least $1 - \delta$, the two-level expected loss (population loss) with infinite topics and infinite sequences per topic, denoted by $L(\theta)$ (see in Equation 5), satisfies,*

$$\mathbb{E}_\mu [L(\theta)] = \mathcal{O} \left\{ \sqrt{\frac{1}{KT_p} \left( D_{\text{KL}}(\mu \parallel \nu) + \log \frac{1}{\delta} \right)} + U(\mathcal{W}_{pre}, K, N, N', T) \right\},$$

*then considering data-dependent and topic-dependent prior $\nu_J$ and detailing the term $D_{\text{KL}}(\mu \parallel \nu_J)$, $L(\theta)$ is further bounded by*

$$\mathcal{O} \left\{ \sqrt{\frac{1}{(K - K')T_p} \left( \frac{\sigma^2 C(\frac{1}{N_{param}}, T')}{K'} + \log \frac{1}{\delta} \right)} + U(\mathcal{W}_{pre}, K, N, N', T) \right\}, \quad (7)$$

*where $C(\frac{1}{N_{param}}, T') = \frac{\beta}{2} e^{8\beta S} \left( 1 - e^{-\frac{T'}{\exp(8\beta S)}} \right)$, $U(\mathcal{W}_{pre}, K, N, N', T)$ denotes the right hand of Equation 6. $K(K')$, $N(N')$ and $T$ denote the number of topics, the number of sequences per topic*

*and the sequence length utilized in the optimization process of Equation* 1. $T'$ *denotes the total training iterations.* $N_{param}$ *denotes the number of model parameters.*

**Remark 4.7** (Optimality Analysis). The term $U(\mathcal{W}_{\text{pre}}, K, N, N', T)$ comes from Theorem 4.3 whose analysis can refer to Remark 4.4. As for the first term in the result, with order $\mathcal{O}\{1/\sqrt{KT_p}\}$, it illustrates the impact of training with a finite number of topics on the model's predictive ability for unseen topics in ICL. In addition, by directly concatenating demonstrations into the ICL prompt in our setting, ICL prompt length reflects the distinction between zero-shot ICL and few-shot ICL. Our theorem exhibits that longer prompts (*i.e.* larger $T_p$) with more demonstrations lead to smaller population loss, facilitating the emergence of ICL. In total, our guarantees reveal the impact of pre-training on the generalization performance on unseen topics and sequences in ICL, with order $\mathcal{O}\{C(\frac{1}{N_{\text{param}}}, T')(1/\sqrt{KT_p} + 1/\sqrt{KNT})\}$. In comparison, Li et al. (2023) derive a generalization bound on unseen topics based on algorithm stability technique, with order $\mathcal{O}\{1/\sqrt{T} + 1/\sqrt{nMT}\}$ where $n, M, T$ denote the sequence length, number of sequences per topic and number of source topics. Our bound is tighter than Li et al. (2023) in the first term, with a compatible second term. We defer the proof to Appendix G.3.1 and G.3.2.

**More Insights Beyond Recent ICL Research.** Our PAC-Bayesian approach offers statistical insights into model performance, emphasizing the impact of pre-training topics, sequences and sequence length. The data-dependent and topic-dependent prior uniquely enhances optimization and may provide more practical guidance on model training, data selection and deduplication, distinguishing our work from related generalization studies (Li et al., 2023; Zhang et al., 2023b). Detailed practical implications are discussed in Appendix E.

## 5 EXPERIMENTS

In this section, we primarily provide some typical experiments to verify our theory, observe the optimization process and prior model initialization. We defer more experiments on linear dynamic systems, synthetic language dataset GINC and real-world language datasets to Appendix D [1].

**Experiments on Synthetic Language Dataset GINC.** Inspired by Xie et al. (2021), we first perform experiments on the synthetic language dataset GINC to verify our theory. GINC is a small-scale language dataset generated from uniform Hidden Markov Models (HMMs) over topics, where distinct state transition matrices represent the unique topics for each HMM, without defining topics explicitly. We train the GPT-2 model with GINC dataset using a single 24GB NVIDIA GeForce RTX 3090. Detailed data-generating process, model and Hyperparameter settings are provided in Appendix D.2.

In the following, we arrange groups of comparative experiments to explore the separate effects of the number of topics ($K$), number of sequences per topic ($N$), sequence length ($T$) and prompt length ($T_p$). We also provide an interesting case where ICL failed.

**Observation (1): Separate Effects of $K$, $N$, $T$ and $T_p$.** In Figure 2, we first present four groups of experiments 2(a)-2(d) to analyze the impact of different factors on generalization. *In Figure 2(a):* For pre-training, take $K = 10$ topics and generate $N \in \{20, 40, 60, 80, 100\}$ pre-training sequences per topic with varying sequence length $T \in \{1280, 2560, 5120, 10240\}$. The ICL performance of pre-trained model is then tested with $T_p = 128$ prompt length. Each line exhibits a growing trend, indicating a better generalization performance with increasing sequences per topic. Comparing the four lines, a larger sequence length also brings better generalization. *From Figure 2(b)-2(d)*, we vary $K \in \{10, 20, 30\}$. Under each $K$, keep $T = 10240$, adjust $N \in \{20, 40, 60, 80, 100\}$ and $T_p \in \{48, 80, 128, 160\}$. Combining these experiments, we validate the effects of $K, N, T_p$ on generalization to emerge ICL ability, closely aligning our Theorems.

**Observation (2): An Interesting Case that ICL Fails.** *In Figure 2(e)*, when the pre-training data contains random transitions, the model observes all token transitions, yet ICL fails. This suggests that the pre-trained models cannot extract information when data distributions do not match the topic, thus failing to achieve ICL.

---

[1]Our code is available at https://github.com/zx-gong/ICL-Emerge.

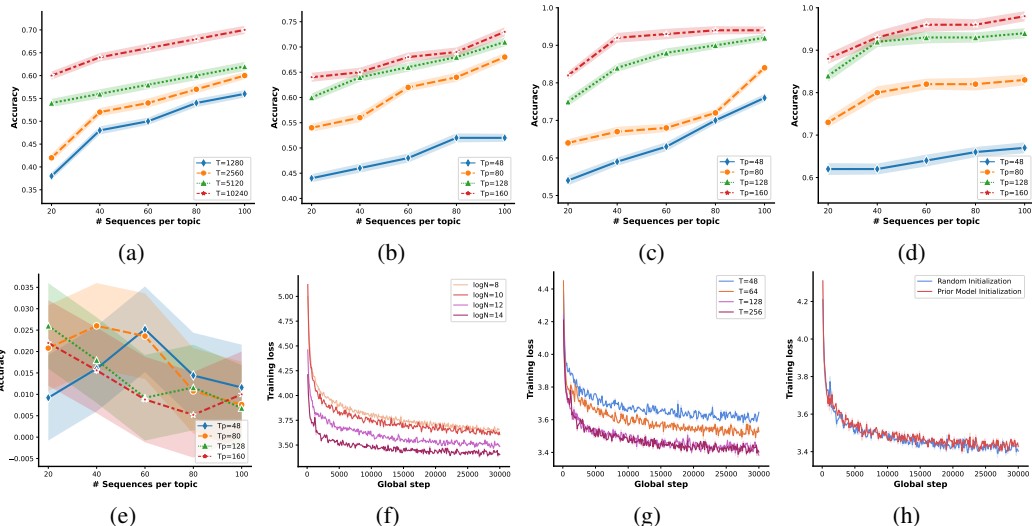

Figure 2: Experiments on GINC and Real-world Language Datasets.

**Experiments on Real-world Language Dataset.**   We further perform experiments on real-world language datasets, inspired by (Min et al., 2021; Wang et al., 2023). We train the GPT2-large model over 20 diverse pre-training datasets covering sentiment analysis, question answering and reasoning tasks. We defer the detailed description of datasets, model and Hyperparameter settings, and observations on the effects of training data, to Appendix D.3. Here, we focus on more insightful experiments regarding the effects of optimization on generalization, as well as potential benefits of effective prior model initialization, guided by the KL term in generalization bounds.

**Observation (1): Optimization Process.**   Through continuous analysis of optimization trajectory, our generalization bounds are optimization-dependent, extending beyond the influence of training data. *In Figure 2(f)*, we present four training processes with varying $N \in \{2^8, 2^{10}, 2^{12}, 2^{14}\}$, while keeping $K = 20$ and $T = 256$ fixed. We observe that larger $N$ brings faster convergence in addition to better performance. Similarly, *in Figure 2(g)*, we take varied $T \in \{48, 64, 128, 256\}$ and keep $K = 20$ and $N = 2^{14}$ fixed. All these observations align with our Theorems that faster training leads to better generalization.

**Observation (2): Prior Model Initialization.**   Building on our generalization results with data-dependent and topic-dependent prior, we design experiments to observe the effects of prior model initialization on training and performance (detailed designation is deferred to Appendix D.3). Our results show that in the random initialization regime, where all pre-training data is used, training for 30,000 steps takes nearly ***7 hours*** on four A100 GPUs. In contrast, under the prior model initialization regime, where a smaller model is used for warmup and serves as the prior for initializing the larger model, training the GPT2-large model takes only ***4 hours*** for the same 30,000 steps on four A100 GPUs, with 0.5 hours required for training the GPT2-small model for 15,000 steps. Furthermore, as shown in the optimization loss curve in Figure 2(h), prior model initialization not only accelerates training but also stabilizes the training process (especially in the early stages), leading to comparable model performance. This approach demonstrates the effectiveness of leveraging prior knowledge in enhancing both training efficiency and model performance, supporting the KL term in our generalization bounds and offering more practical insights.

## 6   CONCLUSION

In this paper, under the AR-NTP paradigm, we propose a systematic pre-training and ICL framework with a layer-wise structure of sequences and topics, alongside two-level expectation. By employing PAC-Bayesian analysis and continuous mathematical techniques like SDE, we provide a comprehensive analysis of data-dependent, topic-dependent and optimization-dependent generalization bounds, demonstrating that ICL emerges for the excellent generalization of sequences and topics. Ultimately, our work aims to take an initial exploration of the origin of ICL ability from the perspective of generalization, supported by both theoretical and experimental results.

## ACKNOWLEDGMENTS

This research was supported by National Natural Science Foundation of China (No.62476277), National Key Research and Development Program of China (NO.2024YFE0203200), CCF-ALIMAMA TECH Kangaroo Fund (No.CCF-ALIMAMA OF 2024008), and Huawei-Renmin University joint program on Information Retrieval. We also acknowledge the support provided by the fund for building worldclass universities (disciplines) of Renmin University of China and by the funds from Beijing Key Laboratory of Big Data Management and Analysis Methods, Gaoling School of Artificial Intelligence, Renmin University of China, from Engineering Research Center of Next-Generation Intelligent Search and Recommendation, Ministry of Education, from Intelligent Social Governance Interdisciplinary Platform, Major Innovation & Planning Interdisciplinary Platform for the 'DoubleFirst Class' Initiative, Renmin University of China, from Public Policy and Decision-making Research Lab of Renmin University of China, and from Public Computing Cloud, Renmin University of China.

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

# Appendix

# A  TABLE OF NOTATIONS

Table 1: Table of Notations.

| Notation | Description |
| --- | --- |
| $K$ | Number of pre-training topics |
| $K'$ | Number of pre-training topics used to compute topic-dependent prior |
| $N$ | Number of pre-training sequences per topic |
| $N'$ | Number of pre-training sequences per topic used to compute data-dependent prior |
| $T$ | Pre-training Sequence length |
| $T_p$ | ICL Prompt length |
| $w_k$ | A pre-training topic with index $k$ |
| $w$ | A ICL topic |
| $\mathcal{W}_{\text{pre}}$ | The set of pre-training topics |
| $\mathcal{W}_{\text{ICL}}$ | The set of ICL topics |
| $\mathbb{P}_{\mathcal{W}}$ | Topic distribution, each topic $w_k \in \mathcal{W}_{\text{pre}}$, $w \in \mathcal{W}_{\text{ICL}}$ is *i.i.d.* drawn from the topic distribution. |
| $E^{k,n}$ | The $n$-th pre-training sequence under the $k$-th topic |
| $E_t^{k,n}$ | The subsequence consisting of the first $t$ tokens of $E^{k,n}$ |
| $E^k$ | The set of pre-training sequences under the $k$-th topic, $|E^k| = N$. |
| $E$ | The set of all pre-training sequences, $E = \{E^k\}_{k=1}^K = \{E^{k,n}\}_{k,n=1}^{K,N}$, $|E| = KN$. |
| $E_{T_p}$ | ICL prompt under ICL topic $w$ |
| $\mathbb{P}_{w_k}$ or $\mathbb{P}(\cdot \mid w_k)$ | Data distribution, each pre-training sequence $E^{k,n} \in E^k$ is *i.i.d.* drawn from the data distribution. |
| $\mathbb{P}_w$ or $\mathbb{P}(\cdot \mid w)$ | Data distribution, ICL prompt $E_{T_p}$ is drawn from the data distribution. |
| $x_{t+1}^{k,n}$ | The $t+1$-th token of pre-training sequence $E^{k,n}$, generated depending on the prefix sequence $E_t^{k,n}$. |
| $x_t$ | The $t$-th token of ICL prompt $E_T$ |
| $\theta$ | The parameters of the pre-trained LLM |
| $\hat{\theta}$ | The optimal parameters of the pre-trained LLM |
| $\mathbb{P}(x_{t+1}^{k,n} \mid E_t^{k,n}, w_k)$ | The true data distribution of token $x_{t+1}^{k,n}$ when given topic $w_k$ and the prefix sequence $E_t^{k,n}$. |
| $\mathbb{P}_\theta(x_{t+1}^{k,n} \mid E_t^{k,n}, w_k)$ | The prediction of token $x_{t+1}^{k,n}$, made from the pre-trained model, when given topic $w_k$ and the prefix sequence $E_t^{k,n}$. |
| $\mathbb{P}_{\hat{\theta}}(x_{t+1}^{k,n} \mid E_t^{k,n}, w_k)$ | The prediction of token $x_{t+1}^{k,n}$, made from the optimal pre-trained model, when given topic $w_k$ and the prefix sequence $E_t^{k,n}$. |
| $L_E(\theta, \mathcal{W}_{\text{pre}})$ | The empirical loss of all pre-training sequences in $E$, see in Equation 1. |
| $L_{E^k}(\theta, w_k)$ | The loss of sequences in $E^k$, see in Equation 1. |
| $L_{E^{k,n}}(\theta, w_k)$ | The loss of sequence $E^{k,n}$, see in Equation 1. |
| $L(\theta, \mathcal{W}_{\text{pre}})$ | The first-level expected loss, take expectation over sequence, see in Equation 4. |
| $L(\theta)$ | The population loss, take expectation over topic and sequence, see in Equation 5. |

# B  OVERVIEW OF TWO-LEVEL EXPECTATION

**Decomposition of Population Loss.**   No matter the inner or outer expectation, the expected loss $L(\theta)$ is incalculable since the data distribution $\mathbb{P}_{w_k}$ and topic distribution $\mathbb{P}_{\mathcal{W}}$ are both unknown (as introduced in Section 3.2, finite sequences and topics are utilized to optimize the empirical loss in

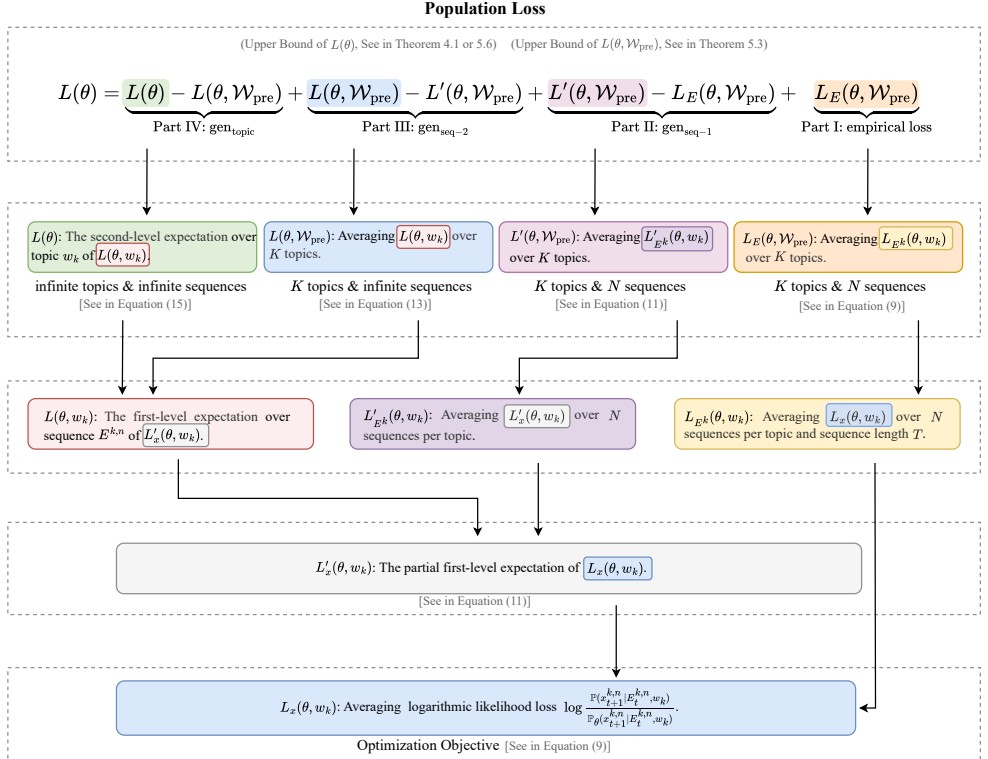

Figure 3: **Overview of Two-Level Expectation. From a horizontal perspective: The first box (from top to bottom):** according to Equation 8, the population loss is decomposed into four parts. We ultimately obtain the upper bound of the population loss by separately defining the upper bound for each part. Combining Part I, Part II and Part III, we obtain Theorem 4.3; further combining with Part IV, we obtain Theorem 4.6. **The second box:** comparing $L(\theta)$ and $L(\theta, \mathcal{W}_{\text{pre}})$, we aim to describe the second-level expectation defined over topic. **The third box:** comparing $L(\theta, w_k)$ and $L'_x(\theta, w_k)$, we aim to describe the complete first-level expectation defined over sequence. **The fourth box:** comparing $L'_x(\theta, w_k)$ and $L_x(\theta, w_k)$, $L'_{E^{k,n}}(\theta, w_k)$ is a partial first-level expectation over token $x^{k,n}_{t+1}$ conditioned on $E^{k,n}_t$. **The fifth box:** Negative logarithmic likelihood loss, the optimization objective for a token. **From a vertical perspective**, the formulas described in the four columns can be found in Equation 15, 13, 11 and 9, respectively. **The first column:** the chain of $L(\theta) \rightarrow L(\theta, w_k) \rightarrow L'_x(\theta, w_k) \rightarrow L_x(\theta, w_k)$. **The second column:** the chain of $L(\theta, \mathcal{W}_{\text{pre}}) \rightarrow L(\theta, w_k) \rightarrow L'_x(\theta, w_k) \rightarrow L_x(\theta, w_k)$. **The third column:** the chain of $L'(\theta, \mathcal{W}_{\text{pre}}) \rightarrow L'_{E^k}(\theta, w_k) \rightarrow L'_x(\theta, w_k) \rightarrow L_x(\theta, w_k)$. **The fourth column:** the chain of $L_E(\theta, \mathcal{W}_{\text{pre}}) \rightarrow L_{E^k}(\theta, w_k) \rightarrow L_x(\theta, w_k)$.

practical). ICL ability can be measured by population loss, which can be decomposed by simply adding and subtracting three terms $L_E(\theta, \mathcal{W}_{\text{pre}})$, $L'(\theta, \mathcal{W}_{\text{pre}})$ and $L(\theta, \mathcal{W}_{\text{pre}})$ in Equation 8. A good ICL learner means a small population loss, i.e. a small value in all four parts. The overview of two-level expectation is shown in Figure 3 and the table of notations is shown in Table 2.

$$L(\theta) = \underbrace{L(\theta) - L(\theta, \mathcal{W}_{\text{pre}})}_{\text{Part IV: gen}_{\text{topic}}} + \underbrace{L(\theta, \mathcal{W}_{\text{pre}}) - L'(\theta, \mathcal{W}_{\text{pre}})}_{\text{Part III: gen}_{\text{seq-2}}}$$
$$+ \underbrace{L'(\theta, \mathcal{W}_{\text{pre}}) - L_E(\theta, \mathcal{W}_{\text{pre}})}_{\text{Part II: gen}_{\text{seq-1}}} + \underbrace{L_E(\theta, \mathcal{W}_{\text{pre}})}_{\text{Part I: empirical loss}} \quad (8)$$

**Part I: empirical loss.** For Part I, the training of the LLM takes into account $K$ topics and $N$ sequences per topic. In this setting, finite topics and finite sequences could affect the performance

of model so that the training loss is called as empirical loss (optimization objective). For a detailed explanation of empirical loss, the same as Equation 1,

$$
\begin{aligned}
L_E(\theta, \mathcal{W}_{\text{pre}}) &= \frac{1}{K} \sum_{k=1}^{K} L_{E^k}(\theta, w_k), \\
L_{E^k}(\theta, w_k) &= \frac{1}{N} \sum_{n=1}^{N} L_{E^{k,n}}(\theta, w_k), \\
L_{E^{k,n}}(\theta, w_k) &= \frac{1}{T} \sum_{t=1}^{T} L_x(\theta, w_k), \\
L_x(\theta, w_k) &= \log \frac{\mathbb{P}(x_{t+1}^{k,n} \mid E_t^{k,n}, w_k)}{\mathbb{P}_\theta(x_{t+1}^{k,n} \mid E_t^{k,n}, w_k)}.
\end{aligned}
\tag{9}
$$

**Part II** : $\text{gen}_{\text{seq-1}}$. Through Part I, we have obtained the empirical loss with finite sequences and finite topics. To address the first-level expectation, it's necessary to evaluate the expected loss over sequence, that is, utilizing an infinite number of sequences for each pre-training topic. Given that the sequential dependence in token generation or prediction, where each subsequent token relies on the preceding tokens, our approach involves initially calculating the expectation of token $x_{t+1}^{k,n}$ conditioned on $E_t^{k,n}$ in this Part II. It's a partial generalization error for the first-level expected loss. This is followed by taking expectation over $E_t^{k,n}$ in the Part III, thereby achieving the comprehensive first-level expectation over sequence $E^{k,n}$.

According to the definition of KL divergence, the partial first-level expectation over sequences $\mathbb{E}_{x_{t+1}^{k,n} \sim \mathbb{P}(\cdot \mid E_t^{k,n}, w_k)} [L_x(\theta, w_k)]$ can be related to $D_{\text{KL}} \left( \mathbb{P}(\cdot \mid E_t^{k,n}, w_k) \parallel \mathbb{P}_\theta(\cdot \mid E_t^{k,n}, w_k) \right)$, i.e.

$$
\begin{aligned}
\mathbb{E}_{x_{t+1}^{k,n} \sim \mathbb{P}(\cdot \mid E_t^{k,n}, w_k)} [L_x(\theta, w_k)] &= \mathbb{E}_{x_{t+1}^{k,n} \sim \mathbb{P}(\cdot \mid E_t^{k,n}, w_k)} \left[ \log \frac{\mathbb{P}(x_{t+1}^{k,n} \mid E_t^{k,n}, w_k)}{\mathbb{P}_\theta(x_{t+1}^{k,n} \mid E_t^{k,n}, w_k)} \right] \\
&= D_{\text{KL}} \left( \mathbb{P}(\cdot \mid E_t^{k,n}, w_k) \parallel \mathbb{P}_\theta(\cdot \mid E_t^{k,n}, w_k) \right) \\
&\triangleq L_x'(\theta, w_k).
\end{aligned}
\tag{10}
$$

Then, taking average of all tokens in a sequence, $N$ sequences per topic and $K$ topics and combining with Equation 10, we define a partial first-level expected loss $L'(\theta, \mathcal{W}_{\text{pre}})$ as

$$
\begin{aligned}
L'(\theta, \mathcal{W}_{\text{pre}}) &= \frac{1}{K} \sum_{k=1}^{K} L_{E^k}'(\theta, w_k), \\
L_{E^k}'(\theta, w_k) &= \frac{1}{N} \sum_{n=1}^{N} L_{E^{k,n}}'(\theta, w_k), \\
L_{E^{k,n}}'(\theta, w_k) &= \frac{1}{T} \sum_{t=1}^{T} L_x'(\theta, w_k), \\
L_x'(\theta, w_k) &= D_{\text{KL}} \left( \mathbb{P}(\cdot \mid E_t^{k,n}, w_k) \parallel \mathbb{P}_\theta(\cdot \mid E_t^{k,n}, w_k) \right).
\end{aligned}
\tag{11}
$$

Finally, a partial generalization error for the first-level expected loss can be described as

$$
\text{gen}_{\text{seq-1}} = L'(\theta, \mathcal{W}_{\text{pre}}) - L_E(\theta, \mathcal{W}_{\text{pre}}).
\tag{12}
$$

**Part III** : $\text{gen}_{\text{seq-2}}$. Through Part II, we derived a partial first-level expected loss $L'(\theta, \mathcal{W}_{\text{pre}})$. Subsequently, in this part, by taking expectation over $E_t^{k,n}$, we will achieve a comprehensive first-level expectation over prefix sequence $E^{k,n}$. Utilizing infinite sequences per topic rather than $N$

Table 2: Table of Notations in Figure 3.

|  | Notation | Description |
|---|---|---|
| $L_E(\theta, \mathcal{W}_{\text{pre}})$ | $L_E(\theta, \mathcal{W}_{\text{pre}})$ | Averaging $L_{E^k}(\theta, w_k)$ over $K$ topics, see in Equation 9. |
|  | $L_{E^k}(\theta, w_k)$ | Averaging $L_{E^{k,n}}(\theta, w_k)$ over $N$ sequences per topic. |
|  | $L_{E^{k,n}}(\theta, w_k)$ | Averaging $L_x(\theta, w_k)$ over sequence length. |
|  | $L_x(\theta, w_k)$ | Negative logarithmic likelihood loss $\log \frac{\mathbb{P}(x_{t+1}^{k,n} \mid E_t^{k,n}, w_k)}{\mathbb{P}_\theta(x_{t+1}^{k,n} \mid E_t^{k,n}, w_k)}$. |
| $L'(\theta, \mathcal{W}_{\text{pre}})$ | $L'(\theta, \mathcal{W}_{\text{pre}})$ | Averaging $L'_{E^k}(\theta, w_k)$ over $K$ topics, see in Equation 11. |
|  | $L'_{E^k}(\theta, w_k)$ | Averaging $L'_{E^{k,n}}(\theta, w_k)$ over $N$ sequences per topic. |
|  | $L'_{E^{k,n}}(\theta, w_k)$ | Averaging $L'_x(\theta, w_k)$ over sequence length. |
|  | $L'_x(\theta, w_k)$ | **Taking the partial first-level expectation** over token $x_{t+1}^{k,n} \sim \mathbb{P}(\cdot \mid E_t^{k,n}, w_k)$. |
| $L(\theta, \mathcal{W}_{\text{pre}})$ | $L(\theta, \mathcal{W}_{\text{pre}})$ | Averaging $L(\theta, w_k)$ over $K$ topics, see in Equation 13. |
|  | $L(\theta, w_k)$ | **Taking the complete first-level expectation** over prefix sequence $E_t^{k,n}$ and token $x_{t+1}^{k,n} \sim \mathbb{P}(\cdot \mid E_t^{k,n}, w_k)$. |
|  | $L'_x(\theta, w_k)$ | The partial first-level expectation over token $x_{t+1}^{k,n}$. |
| $L(\theta)$ | $L(\theta)$ | **Taking the second-level expectation** over topic $w_k$ of $L(\theta, w_k)$, see in Equation 15. |
|  | $L(\theta, w_k)$ | The first-level expectation over sequence $E^{k,n}$. |

sequences, the first-level expected loss $L(\theta, \mathcal{W}_{\text{pre}})$ can be more concretely described as

$$L(\theta, \mathcal{W}_{\text{pre}}) = \frac{1}{K} \sum_{k=1}^{K} L(\theta, w_k),$$
$$L(\theta, w_k) = \mathbb{E}_{E_t^{k,n}} \left[ L'_x(\theta, w_k) \right]. \tag{13}$$

Compared with $L(\theta, \mathcal{W}_{\text{pre}})$ and $L'(\theta, \mathcal{W}_{\text{pre}})$, the difference lies in the second line of Equation 13 and 11 with infinite sequences or $N$ sequences. This difference represents the complete generalization error of sequences which can be denoted as $\text{gen}_{\text{seq-2}}$,

$$\text{gen}_{\text{seq-2}} = L(\theta, \mathcal{W}_{\text{pre}}) - L'(\theta, \mathcal{W}_{\text{pre}}). \tag{14}$$

**Part IV** : $\textbf{gen}_{\textbf{topic}}$. In this part, we further consider the second-level expectation over topic, that is, considering the population loss with infinite sequences and infinite topics. According to the difference between Equation 13 and population loss lies in the number of topics with infinite or $K$, we have the population loss,

$$L(\theta) = \mathbb{E}_{w_k} \left[ L(\theta, w_k) \right]. \tag{15}$$

After which ICL will emerge from the good generalization of sequences and topics. It can be denoted as $\text{gen}_{\text{topic}}$,

$$\text{gen}_{\text{topic}} = L(\theta) - L(\theta, \mathcal{W}_{\text{pre}}). \tag{16}$$

## C  MORE RELATED WORK

**From Multi-Task Learning to Meta-Learning.**    Although drawing inspiration from the assumption of an unknown task distribution in meta-learning analysis, it is worthy to emphasize that ICL generalization analysis under auto-regressive next-token prediction cannot be equivalent to meta-learning generalization. We conduct our analysis under the unique setup of auto-regressive pre-trained LLMs. The prompt token-dependency issue brought by auto-regressive language modeling implies that we cannot directly apply the general meta-learning analysis to ICL generalization analysis. For instance, the study in Bai et al. (2024) directly applied the general approach of meta-learning, assuming that a prompt consists of $N + 1$ *i.i.d.* samples, which is unreasonable for AR-NTP problem

we investigate. For a prompt $(x_1, x_2, \cdots, x_T)$ under AR-NTP, we do not require $x_1 \sim x_T$ to be independent of each other; instead, subsequent tokens depend on previously generated tokens. As mentioned in Section 4.2, addressing prompt token-dependency is one of the significant contributions and challenges compared to other works, including meta-learning works in non-ICL domains. This is the key distinction from traditional meta-learning approach.

**Generalization Analysis.** Understanding the generalization error in learning algorithm which meansures the model performance on unseen data with population loss, has led to the development of several classic methods for establishing its upper bounds. Among these, uniform convergence (including VC dimension, Rademacher complexity) (Bartlett et al., 2017; Shalev-Shwartz et al., 2010; Vapnik et al., 1994), algorithm stability (Bousquet & Elisseeff, 2002; Feldman & Vondrak, 2018; Hardt et al., 2016; Lei & Ying, 2020; Zhang et al., 2022), information-theoretic bounds (Russo & Zou, 2016; 2019; Xu & Raginsky, 2017), and PAC-Bayesian (Catoni, 2007; Dziugaite & Roy, 2017; McAllester, 1998) are prominent techniques. For VC dimension, it depends solely on the hypothesis class which offers the worst-case analysis. For Rademacher complexity, it depends both on the hypothesis class and on the unknown distribution which can be understood as an average-case analysis. The above obtained bounds almost depend on the size of hypothesis space, and become vacuous hence may be unable to explain generalization in deep learning with over-parameterized neural network (Nagarajan & Kolter, 2019; Zhang et al., 2021). Additionally, compared with algorithm stability theory, it considers worst-case and fails in analyzing the relationship between input data and output model. Therefore, we turn to the PAC-Bayesian approach for its unique data-dependent and hypothesis space-independent analysis. In our work, we specifically incorporate a topic-dependent prior within the PAC-Bayesian framework, adding a novel dimension to this analysis. Furthermore, by detailing the KL divergence when considering the optimization process, we obtain optimization algorithm-dependent generalization bound, naturally combining the advantage of algorithm stability technique.

# D  MORE EXPERIMENTS

## D.1  EXPERIMENTS ON LINEAR DYNAMIC SYSTEM

We conduct numerical experiments of linear dynamic system. Our experimental setup follows Li et al. (2019): All ICL experiments are trained and evaluated using the same GPT-2 architecture with 12 layers, 8 attention heads, and 256 dimensional embeddings, on NVIDIA 3090 GPUs.

For a partially-observed dynamical system, the mathematical model can be represented by state and observation equation. Consider the state equation $x_{t+1} = W x_t + \zeta_t$, where $x_t$ represents the state vector at time $t$ in a $d$-dimensional space. This is analogous to the tokens in our analysis. $W$ denotes the state transition matrix and $\zeta_t$ is the process noise satisfying $\mathcal{N}(0, \sigma^2 I_d)$. The observation equation is given by $y_{t+1} = C x_{t+1}$, where $C$ is the observation matrix, indicating that only partial dimensions of the state vector are observable. The uniqueness of different topics is reflected in the parameters $W$ and $C$. Within this linear dynamic system setting, we examine how the number of pre-training topics $(K)$, the number of sequences per topic $(N)$, and the sequence length $(T)$ significantly affect the generalization performance of auto-regressive LLMs. Additionally, we highlight the advantages of both data-dependent and topic-dependent priors.

**The Comparison of Overall Loss and In-context Learning Loss**. Before embarking on our main experiments, we conduct a preliminary comparison between the absolute values of the overall loss and the in-context learning (ICL) loss. In the pre-training phase, we predict all tokens in a sequence and consider the average of these predictions as the overall loss. According to our theoretical proof, this average prediction loss can be naturally generalized to the ICL phase to represent the ability of ICL. Although in more scenarios, the focus often shifts to the predicted outcome of the last token, here the prediction loss of the last token is denoted as ICL loss. In the left of Figure 4, our observations reveal that the ICL loss is consistently lower than the overall loss with a different number of sequences per topic. It's because the prediction loss decreases as the sequence length increases, corresponding to our theory. Consequently, our theoretical bounds hold validity and significance under both overall and ICL loss assessments.

**Separate Effects of $K$, $N$ and $T$.** In our experimental design, we manipulate the variables $K$, $N$, and $T$ independently and the experimental results are shown in Figure 5. In Figure 5(a), with fixed

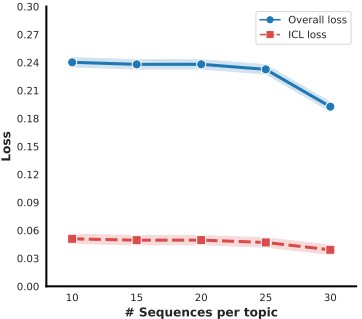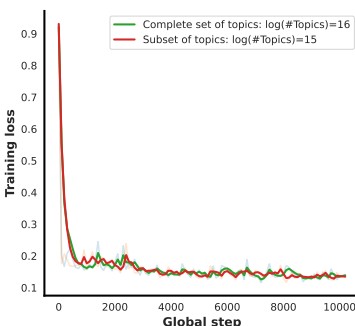

Figure 4: Experiments on Linear Dynamic System. Left: The comparison of overall loss and in-context learning loss. Right: The comparison of experiments conducted on complete topic set and subset of topics.

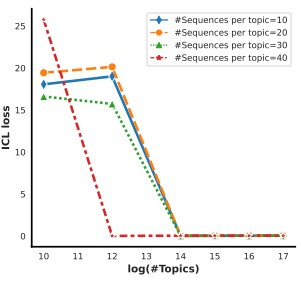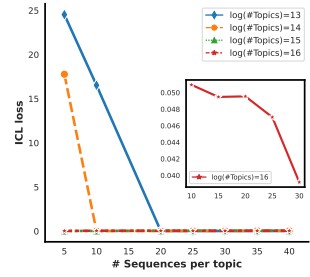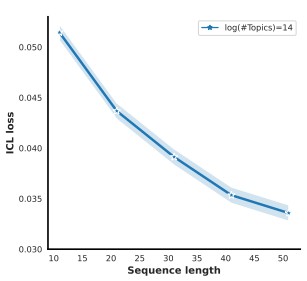

(a) Vary number of topics.  (b) Vary number of sequences per topic.  (c) Vary sequence length.

Figure 5: Experiments on Linear Dynamic System: The effect of the number of pre-training topics ($K$), the number of sequences per topic ($N$) and sequence length ($T$).

number of sequences per topic and sequence length ($T = 11$) for each line ($N = 10, 20, 30, 40$), we vary the number of topics within the range of $K = 2^{10} \sim 2^{17}$. As $K$ increases, it's noticeable that the ICL loss consistently show a downward trend across all four lines. Furthermore, the sharp drops in ICL loss observed in these cases suggests that LLMs exhibit emerging abilities when the accumulated topic count reaches certain thresholds. In Figure 5(b), holding the number of topics and sequence length ($T = 11$) constant for each line (with topics set at $K = 2^{13}, 2^{14}, 2^{15}, 2^{16}$), we adjust the number of sequences per topic, varying it within a range of $N = 5 \sim 40$. Comparing the four cases, the ICL loss diminishes as $N$ grows. Notably, in cases with less sufficient topics (like $K = 2^{13}$ and $2^{14}$), a larger $N$ leads to significant reductions in ICL loss. Specially, the decrease trend of ICL loss is particularly evident in the magnified view of the case where $K = 2^{16}$. In Figure 5(c), maintaining a constant number of topics ($K = 2^{14}$) and sequence per topic ($N = 40$), we modify the sequence length, allowing it to vary within a range of $T = 11 \sim 51$. We can find that ICL loss clearly decreases as the sequence length grows.

**The Advantages of Both Data-dependent and Topic-dependent Priors.** As introduced before, data-dependent and topic-dependent priors provide a chance to make the generalization bound computable. To illustrate this, we take the example of topic-dependent prior and two experiments are conducted: one with a complete set of topics ($K = 2^{16}$) and another with its subset ($K' = 2^{15}$). Observing the results in the right of Figure 4, both training processes eventually converge to nearly identical steady states. This suggests that using a subset of topics to obtain a topic-dependent prior in preliminary experiments yields a prior that is closer to the posterior than a randomly selected prior. Then for the KL divergence between prior and posterior distribution of model parameters in our generalization bounds, assume these distributions are either uniform or Gaussian allows us to derive the closed-form expressions for the KL divergence.

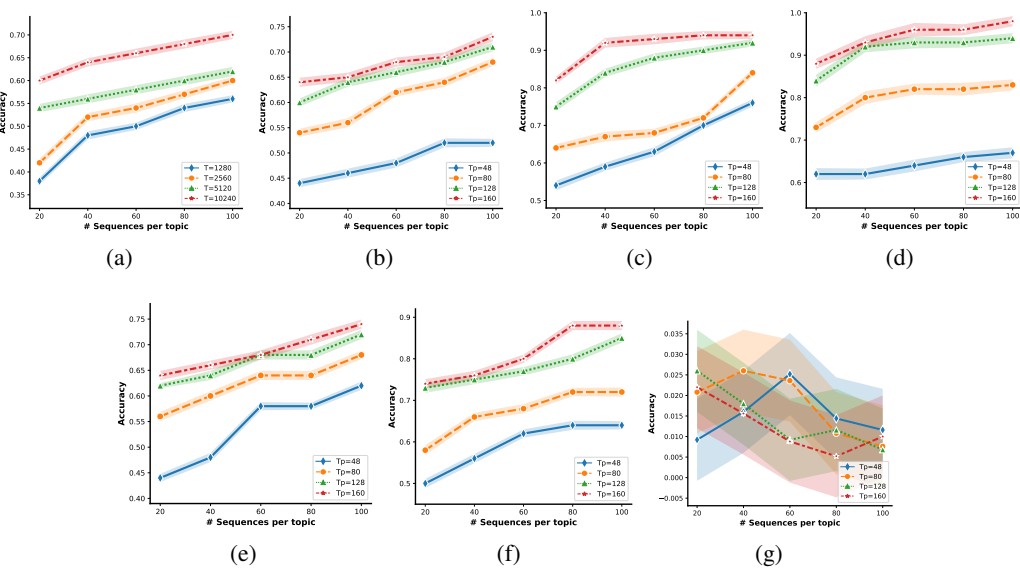

(a)       (b)       (c)       (d)

(e)       (f)       (g)

Figure 6: Experiments on GINC Synthetic Language Dataset.

## D.2 EXPERIMENTS ON GINC SYNTHETIC LANGUAGE DATASET

Inspired by Xie et al. (2021), we perform experiments on a synthetic language dataset GINC to verify our theoretical results.

**GINC Dataset.** GINC is a small-scale language dataset generated from a uniform mixture of Hidden Markov Models (HMMs) over a family of topics/concepts (Xie et al., 2021). The generation steps are as follows: *(1) Prepare transition matrix for HMMs:* The topic/concept determines the state transition matrix in HMM. For simulation, the transition matrix is randomly generated for each topic (each HMM), respectively; *(2) Prepare vocabulary:* The vocabulary is generated as combinations of letters starting from 'a' to 'z', 'aa' to 'az', and so on. We can obtain vocabularies with different sizes; *(3) Prepare memory matrix:* A unique matrix is created that records the mapping of vocabulary and state; *(4) Generate sequences:* Given a fixed topic and an initial state, generate the next state based on the transition matrix, and then obtain the observed token using the memory matrix. In total, each sequence is sampled from a random HMM in the family.

**Model and Hyperparameter Settings.** Our transformer model is based on the GPT-2 architecture with 4 layers, 12 attention heads, and 768-dimensional embeddings (Wolf, 2019). We train the model for 20 epochs using the AdamW optimizer with a batch size of 8 and a linear learning rate schedule. The schedule includes a warmup phase of 1000 steps, up to the learning rate of 8e-4. Notably, we adopt a large portion of the code from Xie et al. (2021). All experiments on GINC are conducted using a single 24GB NVIDIA GeForce RTX 3090.

In the following, We empirically explore the separate effects of the number of topics ($K$), number of sequences per topic ($N$), sequence length ($T$) and prompt length ($T_p$). We detail $K \in \{10, 20, 30\}$, $N \in \{20, 40, 60, 80, 100\}$, $T \in \{1280, 2560, 5120, 10240\}$, $T_p \in \{48, 80, 128, 160\}$, where ranging $T$ with directly masking the token that exceeds the specified length and do not taking special consideration. In totoal, we arrange groups of comparative experiments to verify that increasing $K, N, T, T_p$ individually improves the model's generalization performance as demonstrated in our Theorems. Additionally, we discuss the effect of vocabulary size and provide an interesting case involving with a failed ICL.

**Observation (1): Separate Effects of $K$, $N$, $T$ and $T_p$.** We first present four groups of experiments 6(a)-6(d) in Figure 6. *In Figure 6(a):* For pre-training data, take $K = 10$ topics and generate $N \in \{20, 40, 60, 80, 100\}$ pre-training sequences/documents per topic, in addition with varying sequence length $T \in \{1280, 2560, 5120, 10240\}$. Then with the pre-trained model, test ICL performance on the prompt with $T_p = 128$ prompt length. Each line exhibits a growing trend, indicating a better

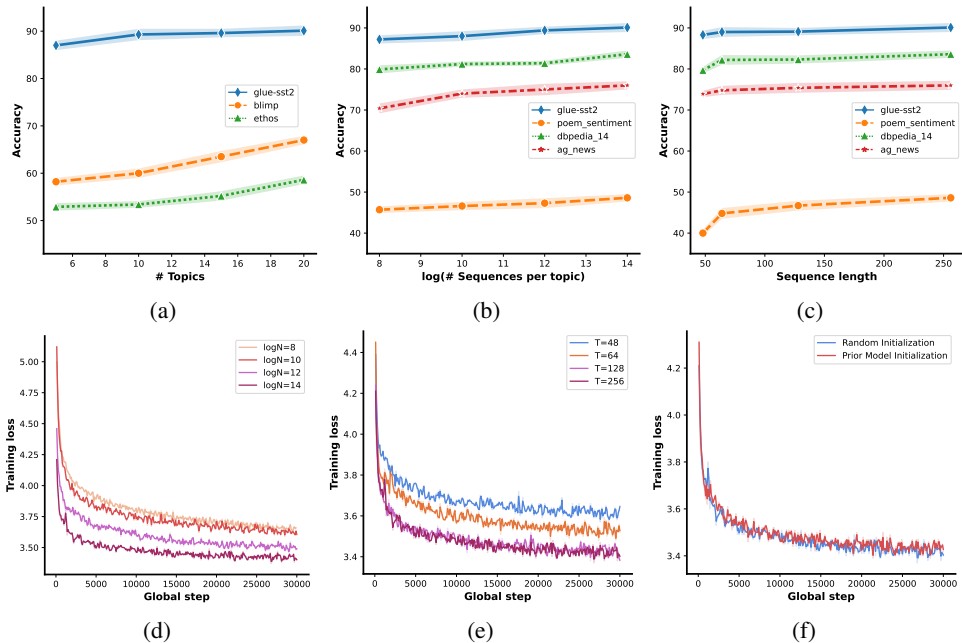

Figure 7: Experiments on Real-world Language Dataset.

generalization performance with increasing sequences per topic. Comparing the four lines from bottom to top, a larger sequence length also brings better generalization. *From Figure 6(b)-6(d)*, we vary $K \in \{10, 20, 30\}$. Under each $K$, keep sequence length $T = 10240$, with varying $N \in \{20, 40, 60, 80, 100\}$ and $T_p \in \{48, 80, 128, 160\}$. Combining these three groups of experiments, we validate the effects of $K, N, T_p$ on generalization, closely aligning our Theorems.

**Observation (2): Effect of Vocabulary Size and an Interesting Case that ICL Fails.** *In Figure 6(b), 6(e) and 6(f)*, We vary the vocabulary size within $\{50, 100, 150\}$. With fixed $K = 10$ topics, we vary $N \in \{20, 40, 60, 80, 100\}$ and $T_p \in \{48, 80, 128, 160\}$. Apart from the observations similar to Figure 6(b)-6(d) about $N, T_p$, we surprisingly find that a larger vocabulary size leads to higher ICL prediction accuracy. This aligns with our understanding that the number of possible token combinations in sequences grows with increased vocabulary size. It also implies that more diverse training data improves ICL performance. This is further implicitly supported by our theory, which suggests increasing the training sample size as much as possible to ensure sample diversity. Furthermore, we conduct an interesting experiment *in Figure 6(g)*. When the pre-training data contains random transitions, the model observes all token transitions, yet ICL fails. This suggests that the pre-trained models cannot extract information when data distributions do not match the topic, thus failing to achieve ICL.

### D.3    EXPERIMENTS ON REAL-WORLD LANGUAGE DATASETS.

We further perform experiments on real-world language datasets, inspired by (Min et al., 2021; Wang et al., 2023).

**Datasets, Model and Hyperparameter Settings.** In the pre-training phase, we consider a mixture of a series of language tasks, mainly including 20 datasets. Classified by task types, including sentiment analysis (glue-sst2, poem_sentiment, yelp_polarity and emotion), linguistic analysis (glue-cola, blimp), text classification (ag_news, dbpedia_14, ethos), question answering (tweet_qa) and commonsense reasoning (swag). Different datasets are considered as different topics (reflected in $K$ from our framework). In ICL phase, we test ICL performance with different datasets. All the datasets are obtained from Hugging Face. We train the GPT2-large model with a batch size of 16 and a learning rate of 1e-4 for total 30,000 iterations. Notably, we adopt a large portion of the code from

Wang et al. (2023). All experiments are conducted using four 24GB NVIDIA GeForce RTX 3090 and 40GB A100 GPUs.

In the following, we empirically explore the separate effects of the number of topics ($K$), number of sequences per topic ($N$) and sequence length ($T$). By detailing $K \in \{5, 10, 15, 20\}$, $N \in \{2^8, 2^{10}, 2^{12}, 2^{14}\}$, $T \in \{48, 64, 128, 256\}$, we arrange groups of comparative experiments to verify that increasing $K, N, T$ individually improves the model's generalization performance as demonstrated in our Theorems. Additionally, we observe the impact of optimization process and prior model initialization.

**Observation (1): Separate Effects of $K$, $N$ and $T$.** *In Figure 7(a)*, we investigate the impact of varying the number of topics $K$. Specifically, varying $K \in \{5, 10, 15, 20\}$, keeping fixed $N = 2^{14}$ sequences per topic and sequence length $T = 256$. The results show that for ICL test prompts from different datasets, increasing $K$ consistently improves ICL accuracy, as expected from our theoretical analysis. *In Figure 7(b)*, we examine the effect of varying $N \in \{2^8, 2^{10}, 2^{12}, 2^{14}\}$, with fixed $K = 20, T = 256$. We observe that increasing $N$ leads to better performance in ICL phases, reinforcing the idea that more sequences per topic enhances model generalization and further benefits ICL. Similarly, *in Figure 7(c)*, we explore the impact of varying $T \in \{48, 64, 128, 256\}$ while keeping fixed $K = 20$ and $N = 2^{14}$. Increasing $T$ also brings better ICL performance.

**Observation (2): Optimization Process.** Through continuous optimization trajectory analysis, our generalization bounds are also optimization-dependent. Thus beyond the influence of training data, we investigate whether optimization properties align with our theory. *In Figure 7(d)*, we present four different training processes where $N \in \{2^8, 2^{10}, 2^{12}, 2^{14}\}$ is varied, with fixed $K = 20$ and $T = 256$. This setting mirrors Figure 7(b) where we have demonstrated that increasing $N$ leads to better generalization performance. Furthermore, we observe that larger $N$ also brings faster convergence during training. This aligns with our Theorems that with a smaller number of training iterations $T'$ to converge, *i.e.*, the model trains faster, and further generalizes better. Similarly, *Figure 7(e)* takes the same configuration with Figure 7(c), which also exhibits the connection between optimization and generalization that 'trains faster, generalize better'.

**Observation (3): Prior Model Initialization.** Based on our generalization analysis with a data-dependent prior, we propose that leveraging prior model initialization could accelerate model training. Specifically, consider the following setup: our training data consists of $K = 20$ pre-training topics, $N = 2^{14}$ training sequences per topic and sequence length $T = 256$.

- **Step 1:** Train the GPT2-small model for 15,000 steps using $K = 5$ pre-training topics, $N = 2^{14}$ training sequences per topic and sequence length $T = 256$.
- **Step 2:** Transfer the weights from GPT2-small model to the corresponding weight matrices of GPT2-large, ensuring dimension compatibility. Initialize the weights randomly for the additional transformer layers in GPT2-large.
- **Step 3:** Train the GPT2-large model for an additional 30,000 steps using the full pre-training data ($K = 20, N = 2^{14}, T = 256$)

According to our experimental results, the random initialization regime with all pre-training data requires nearly **7 hours** on four A100 GPUs to complete 30,000 steps. However, under the prior model initialization regime, where a smaller model is used for warmup and serves as the prior model initialization for the larger model, training the GPT2-large model takes only **4 hours** for 30,000 steps on four A100 GPUs under the same setting of $K, N, T$ (with 0.5 hours needed for training the GPT2-small model for 15,000 steps).

Furthermore, as shown in the optimization loss curve in Figure 7(f), the prior model initialization not only accelerates training but also stablizes the training process (especially at the early stage), leading to comparable or even improved model performance. This approach demonstrates how effectively leveraging prior knowledge can contribute to the training process and performance, supporting the KL term in our generalization bounds and presenting more practical insights.

# E    PRACTICAL IMPLICATIONS

We first provide guidance for the quantitative selection of $K$, $N$ and $T$ based on the upper bound of expected loss described by theoretical results.

**Increase the Number of Pre-training Topics.**    For the ICL ability of LLMs, it relies on examples within a given prompt to adjust its behavior, so more topics (or tasks) provide richer information and learning opportunities. As the number of tasks increases, the model is able to learn from a broader range of contexts, thereby enhancing its generalization ability. This is different from general multi-task learning that it aims to learn multiple tasks simultaneously and if the tasks are too different or unrelated, it may lead to task interference, thereby reducing overall performance (i.e., under multitask learning, having more topics does not necessarily lead to better model generalization performance). Instead, our defined topics satisfy the assumption of topic distribution, implying a correlation between pre-training and ICL topics. This also leads to our conclusion that 'more topics lead to better model generalization performance', which differs from general multi-task learning. Furthermore, when increasing the number of topics, our goal is to cover as many different types of topics as possible, to guarantee the diversity of topics, which will enrich the model's learning experience and help the model better understand new contexts with unseen topics. It potentially explains why one can improve ICL performance by selecting appropriate kind of 'few-shot' examples or exemplars to optimize performance (i.e. retrieving shots best suited to the topic/task).

**Expand the Scale of Pre-training Sequences.**    Using a large amount of training sequences per topic can provide more topic information, which helps the model better understand the language patterns for this topic. This guarantees the ability to perform well on the new sequences with a seen topic. This opinion is similar to the classical machine learning problem, where more training data helps the model perform excellently on the test data.

**Increase Sequence Length or Prompt Length.**    Training the model to process longer sequences can enable it to better understand the context and details of lengthy texts, especially for topics or tasks that require an in-depth understanding of long articles, such as text summarization and extended question answering. We hold that longer sequence length help the model maintain coherence and completeness of information when dealing with complex problems.

Furthermore, in our PAC-Bayesian generalization bounds, the key term $D_{KL}(\mu \parallel \nu)$ surely offers possibilities to quantify the information contained in the model and data, thereby providing practical guidance for model training, training data selection and deduplication.

**Practical Guidance for Model Training - Prior Model Initialization.**    Typically, randomly initialized parameters follow uniform or standard normal distributions, which lack any specific information about the data. In contrast, during pre-training, we begin with a small-scale subset of data to train a prior model. The parameters of this prior model can then serve as an informative starting point for longer and more sufficient training with greatly-large-scale pre-training data. When using a data-dependent prior rather than random initializations, this results in a smaller $D_{KL}(\mu \parallel \nu)$, which in our theorems represents the distance between model posterior $\mu$ and prior $\nu$, contributing to a better *generalization*. Furthermore, a lower $D_{KL}(\mu \parallel \nu)$ also enhances the *optimization*, by detailing this term with continuous optimization trajectory analysis. Specifically for example in Theorem 4.6, when with topic-dependent prior $\nu_J$, $D_{KL}(\mu \parallel \nu_J) \approx \sigma^2 C(\frac{1}{N_{param}}, T')/K'$, where $C(\frac{1}{N_{param}}, T') = \frac{\beta}{2} e^{8\beta S}(1 - e^{-\frac{T'}{exp(8\beta S)}})$. A smaller $D_{KL}(\mu \parallel \nu_J)$ means that this favorable initialization brings a stable training (with reduced gradient norm $\sigma$) and avoids exploring the entire parameter space (with fewer optimization iterations $T'$). This aligns with our understanding that data patterns guide the model toward appropriate directions during training, reducing the likelihood of encountering unsuitable local minima or saddle points.

In total, using a data-dependent and topic-dependent prior for model initialization can significantly *improve training stability, model convergence, and generalization.* This approach is particularly useful in multi-task learning, where it helps establish relevant priors for each task/topic in advance. Although employing more strategies to choose the subset $K'$ can further refine the prior, excessive refinement may introduce new computational costs and efficiency trade-offs. We emphasize that even

without careful data selection for prior model learning, a data-dependent prior generally outperforms random initialization. Particularly, when random initialization does not yield good performance, a data-dependent prior model may provide a new opportunity.

**Practical Guidance for Training Data Selection and Deduplication.** It is well-known that the vast amount of data obtained from the internet serves as input for compressing world knowledge into LLMs (Delétang et al., 2023). The data quality among redundant data determines the upper limit of the performance of LLMs. Therefore, considering a data-dependent pre-training and ICL generalization framework has immense potential for guiding data. In our theory, to explicitly show the impact of data, we adopt a data-dependent and topic-dependent prior $\nu_J$ and further detail $D_{KL}(\mu \parallel \nu_J)$ with optimization analysis. We have discussed this in detail before: in *'Practical Guidance for Model Training'* part, we emphasize the advantages of prior model initialization over random initialization in model training. Here, we aim to further explore its guidance for training data *from the perspective of compression*.

Specifically, we select a subset of size $K'$ from the $K$ pre-training topics to estimate a prior distribution. If a smaller $K'$ can estimate a prior that is very close to the posterior distribution, it indicates that the information from the $K$ topics can actually be compressed into a smaller subset of $K'$ topics. This reflects the compressibility of the data, and can thus *backward guide pre-training data* to further undergo data selection and deduplication, such as through topic clustering, data diversity, or information gain metrics (e.g., $D_{KL}(\nu(D) \parallel \nu(D_i))$, if this value is small, the data block $D_i$ is considered redundant and can be reduced in weight or removed to decrease the model's reliance on redundant information.) The reprocessed pretraining data may exclude some noise interference, further improving model performance, saving computational resources, and facilitating training for new models.

# F COMPLETE THEOREMS: ICL EMERGES FROM GENERALIZATION OF PRE-TRAINED LLMs

In Section 4, we have introduced Theorem 4.3 and 4.6. Here, in the following Section F.1, we divide Theorem 4.3 into two parts: Theorem F.1 and Theorem F.3. Similarly, in the following Section F.2, we divide Theorem 4.6 into two parts: Theorem F.5 and Theorem F.7.

## F.1 GENERALIZATION OF SEQUENCES: THE FIRST-LEVEL EXPECTATION

Under finite $(K)$ pre-training topics, we consider the first-level expectation where infinite sequences per topic are utilized. It describes comprehensive learning for each pre-training topic in the ideal case so that the pre-trained model can perform excellently on the seen topics in ICL phase. For this first-level expected loss $L(\theta, \mathcal{W}_{\text{pre}})$ with two partial expectation, it's represented as $\frac{1}{K} \sum_{k=1}^{K} \mathbb{E}_{E_t^{k,n}} \left[ D_{\text{KL}}\left( \mathbb{P}(\cdot \mid E_t^{k,n}, w_k) \parallel \mathbb{P}_\theta(\cdot \mid E_t^{k,n}, w_k) \right) \right]$ (details see in Equation 13). The following Theorem will give the upper bound of $L(\theta, \mathcal{W}_{\text{pre}})$.

In the following Theorem, we first give an general result that KL distance between posterior $\mu$ and prior $\nu$ is kept in the upper bound of the first-level expected loss. Here, the prior is a general prior distribution rather than a data-dependent prior.

**Theorem F.1** (Generalization Bound of the First-Level Expected Loss). *Let the auto-regressive LLM $\mathbb{P}_\theta$ be the empirical solution of Equation 1, and $\mathbb{P}(\cdot \mid w)$ denotes the true data distribution under topic $w$. Under Assumptions 4.1, for any $0 < \delta < 1$, with probability at least $1 - \delta$, the first-level expected loss with $K$ topics and infinite sequences per topic, denoted by $L(\theta, \mathcal{W}_{pre})$ (see in Equation 4 or Equation 13), satisfies,*

$$\mathbb{E}_\mu \left[ L(\theta, \mathcal{W}_{pre}) \right] = \mathcal{O} \left\{ \sqrt{\frac{\log 1/\delta}{KNT}} + \sqrt{\frac{1}{KNT} \left( D_{\text{KL}}(\mu \parallel \nu) + \log \frac{1}{\delta} \right)} - \epsilon_{opt} \right\},$$

*where $\epsilon_{opt}$ is the optimization error (see in Equation 3). $\mu$ and $\nu$ are the posterior and prior distribution of model parameters $\theta$, respectively. $K$, $N$ and $T$ denote the number of topics, the number of sequences per topic and the sequence length utilized in the optimization process of Equation 1.*

**Remark F.2.** Theorem F.1 reveals that when considering the first-level expectation over sequences, the expected loss achieves $\frac{1}{\sqrt{KNT}}$ rate. This indicates that an increase in the number of training topics ($K$), the number of sequences per topic ($N$), and the sequence length ($T$) leads to a reduction in the first-level expected loss, aligning with both intuitive understanding and empirical evidence. Note that the length of different sequences $T_{k,n}$ vary from each other which implies the potential for sampling imbalanced sequences from various topics. Moreover, the number of sequences $N_k$ per topic can also be different. If sequences under a specific theme are notably short, balancing can be achieved by sampling a greater number of these sequences, i.e. increasing $N_k$, ensuring that the product of $N_k T_{k,n}$ for all themes maintains a level of equilibrium. This approach ensures that the final representation of $NT$ conveys an averaged meaning. If certain themes yield fewer sequences, it indicates a lower probability of occurrence for those themes. Under the framework of theme distribution (as defined by the second level expectation), the contribution of such themes (smaller $N_k T_{k,n}$) to $NT$ won't be dominant. In conclusion, themes with higher occurrence probabilities are predominant and more sequences can be more readily sampled. Even if these sequences are shorter, we can compensate by sampling more sequences to achieve an average level $NT$ which corresponds to our result.

In the next Theorem, we carefully consider a data-dependent prior (Li et al., 2019), replacing $D_{\mathrm{KL}}(\mu \parallel \nu)$ with $D_{\mathrm{KL}}(\mu \parallel \nu_J)$ in Theorem F.1 and further deriving a more detailed upper bound.

**Data-Dependent Prior.** We employ the following method for generating a data-dependent prior (Li et al., 2019). Let $J$ include $N'$ indexes uniformly sampled from $[N]$ without replacement and $I$ is $[N] \setminus J$, splitting pre-training sequences under fixed topic $w_k$ into two parts $E_I^k$ and $E_J^k$. Under all pre-training topics, we have $E_I = \{E_I^k\}_{k=1}^K$ and $E_J = \{E_J^k\}_{k=1}^K$. The prior distribution of model parameters $\theta$ depends on the subset $E_J$, which is denoted by $\nu_J$ and the posterior distribution of $\theta$ depends on $E_I$ denoted by $\mu$. Thus, a parallel training process with $E_J$ are conducted, and after that, a data-dependent prior $\nu_J$ will be obtained. We emphasize that extracting a portion of training data to learn the prior distribution of model parameters has significant implications. Specifically, this approach allows the prior to adapt to specific features and trends in the data, enhancing the model's ability to capture and learn from these nuances. In addition, even if we sacrifice a portion of the training data, the prior will lead to a posterior distribution that is better aligned with the actual data distribution. In high-dimensional spaces, a data-dependent prior provides a more informed starting point.

**Theorem F.3** (Data-Dependent and Optimization-Dependent Generalization Bound of the First-Level Expected Loss). *Under the conditions maintained in Theorem F.1 and Assumption 4.2, when considering data-dependent prior $\mu_J$, for any $0 < \delta < 1$, with probability at least $1 - \delta$, the first-level expected loss with $K$ topics and infinite sequences per topic, denoted by $L(\theta, \mathcal{W}_{pre})$ (see in Equation 4 or Equation 13), satisfies,*

$$\mathbb{E}_\mu \left[ L(\theta, \mathcal{W}_{pre}) \right] = \mathcal{O} \left\{ \sqrt{\frac{\log 1/\delta}{K(N - N')T}} + \sqrt{\frac{1}{K(N - N')T} \left( D_{\mathrm{KL}}(\mu \parallel \nu_J) + \log \frac{1}{\delta} \right)} - \epsilon_{opt} \right\},$$

*then detailing the term $D_{\mathrm{KL}}(\mu \parallel \nu_J)$, $L(\theta, \mathcal{W}_{pre})$ further satisfies,*

$$\mathcal{O} \left\{ \sqrt{\frac{\log 1/\delta}{K(N - N')T}} + \sqrt{\frac{1}{K(N - N')T} \left[ \frac{L^2 C(\frac{1}{N_{param}}, T')}{N'} + \log \frac{1}{\delta} \right]} - \epsilon_{opt} \right\}, \qquad (17)$$

*where $C(\frac{1}{N_{param}}, T') = \frac{\beta}{2} e^{8\beta S} \left( 1 - e^{-\frac{T'}{\exp(8\beta S)}} \right)$. $\epsilon_{opt}$ is the optimization error (see in Equation 3). $K$, $N (N')$ and $T$ denote the number of topics, the number of sequences per topic and the sequence length utilized in the optimization process of Equation 1. $T'$ denotes the total training iterations. $N_{param}$ denotes the number of model parameters.*

**Remark F.4.** The PAC-Bayesian generalization bound of the first-level expected loss can be bounded by the KL divergence between the distribution of the model obtained by the real training process and data-dependent prior, i.e. $D_{\mathrm{KL}}(\mu \parallel \nu_J)$. Analyzing the continuous Langevin dynamic of model parameters $\theta$, Fokker-Planck Equation is used to describe the KL distance between two probability density function of two optimization processes, furthermore, referring to the proof of Lemma G.5 in Li et al. (2019), we demonstrate that the integral of the gradient difference of

$\left\| \nabla L_{E_I}(\theta_t, \mathcal{W}_{\text{pre}}) - \nabla L_{E_J}(\theta_t, \mathcal{W}_{\text{pre}}) \right\|_2^2$. Consequently, we bound $D_{\text{KL}}(\mu \parallel \nu_J)$ with $\frac{L^2 C(\frac{1}{N_{\text{param}}}, T')}{N'}$, which is related to optimization algorithm. As $T'$ increases, $C(\beta, T')$ increases, i.e., the generalization error increases. This reflects the influence of total training iterations $T'$ on testing loss, corresponding to the classical viewpoint 'train faster, generalize better' (Hardt et al., 2016; Lei & Ying, 2020; Zhang et al., 2022). In addition, the constant $L$ related to optimization reflects that the upper bound of the gradient of AR-NTP loss also impacts the generalization performance. Observing the derived upper bound, we notice that the last term, $\sqrt{\frac{\log 1/\delta}{K(N-N')T}} \sim \frac{1}{\sqrt{K(N-N')T}}$, provides similar insights to

$\frac{1}{\sqrt{KNT}}$ in Theorem F.1. In total, by detailing the KL divergence, we establish a more refined bound which is data-dependent and optimization-dependent.

**In summary, the proof of Theorem 4.3 is provided in Appendix G.2.1 and Appendix G.2.2.**

### F.2 Generalization of Sequences and Topics: Two-Level Expectation

Up to now, we have analyzed the first-level expected loss with $K$ topics and infinite sequences per topic. In this ideal case, the pre-trained LLM can perform excellently on the new test prompt under seen topics in ICL phase. In this section, we use similar techniques to further consider the second level expectation with infinite topics, so that the pre-trained LLM could perform well on unseen topics under the topic distribution assumption. At this moment, ICL emerges from the generalization of sequences and topics.

We first give a general result in Theorem F.5 with $D_{\text{KL}}(\rho(\theta) \parallel \pi(\theta))$, which extends beyond Theorem F.3 by incorporating infinite topics.

**Theorem F.5** (Data-Dependent and Optimization-Dependent Generalization Bound of the Two-Level Expected Loss). *Let the auto-regressive LLM $\mathbb{P}_\theta$ be the empirical solution of Equation 1, and $\mathbb{P}(\cdot \mid w)$ is the true data distribution under topic $w$. Under Assumptions 4.1, 4.2 and 4.5, for any $0 < \delta < 1$, with probability at least $1 - \delta$, the two-level expected loss (population loss) with infinite topics and infinite sequences per topic, denoted by $L(\theta)$ (see in Equation 5), satisfies,*

$$\mathbb{E}_\mu[L(\theta)] = \mathcal{O}\left\{ \sqrt{\frac{1}{KT_p}\left(D_{\text{KL}}(\mu \parallel \nu) + \log\frac{1}{\delta}\right)} + U(\mathcal{W}_{pre}, K, N, N', T) \right\},$$

*where $U(\mathcal{W}_{pre}, K, N, N', T)$ denotes the right hand of equality 6 or equality 17. $\mu$ and $\nu$ are the posterior and prior distribution of model parameters $\theta$, respectively. $K$, $N(N')$ and $T$ denote the number of topics, the number of sequences per topic and the sequence length utilized in the optimization process of Equation 1.*

**Remark F.6.** The term $U(\mathcal{W}_{\text{pre}}, K, N, N', T)$ comes from Theorem F.3 whose analysis can refer to Remark F.4. As for the first term in the result, with order $\mathcal{O}\{\frac{1}{\sqrt{KT_p}}\}$, it illustrates the impact of training with a finite number of topics on the model's predictive ability for unseen topics in ICL. In addition with larger prompt length, ICL emerges much easier from the generalization of pre-trained LLMs.

Next, we propose topic-dependent prior whose core idea comes from data-dependent prior (Li et al., 2019), i.e., a portion of $K$ topics will be used for calculating model prior and other topics will be used for obtaining posterior. $D_{\text{KL}}(\rho \parallel \pi)$ in Theorem F.5 will be replaced by $D_{\text{KL}}(\rho \parallel \pi_J)$ and further derives a more detailed upper bound. Since then, we can provide data-dependent, topic-dependent and optimization algorithm-dependent generalization error bound of the two-level expected loss.

**Topic-Dependent Prior.** We employ the following method for generating a topic-dependent prior, similar to data-dependent prior (Li et al., 2019). We split topics into two parts and let $J$ include $K'$ indexes uniformly sampled from $[K]$ without replacement and let $I$ be $[K] \setminus J$, then the total sequences are divided into $E^I = \{E^k\}_{k \in \mathcal{W}_{\text{pre},I}}$ and $E^J = \{E^k\}_{k \in \mathcal{W}_{\text{pre},J}}$. Assume that the posterior distribution of model parameters $\theta$ depends on $E^I$ denoted by $\rho$ and the prior distribution of $\theta$ depends on the topic subset $E^J$, which is denoted by $\pi_J$. A parallel training process is performed with $E^J$ based on the same LLM architecture, and after that, a topic-dependent prior $\pi_J$ will be obtained.

**Theorem F.7** (Data-Dependent, Topic-Dependent and Optimization-Dependent Generalization Bound of the Two-Level Expected Loss). *Under the conditions maintained in Theorem F.5 and Assumption*

4.5, when further considering topic-dependent prior, for any $0 < \delta < 1$, with probability at least $1 - \delta$, the two-level expected loss (population loss) with infinite topics and infinite sequences per topic, denoted by $L(\theta)$ (see in Equation 5), satisfies,

$$\mathbb{E}_\mu\left[L(\theta)\right] = \mathcal{O}\left\{\sqrt{\frac{1}{(K-K')T_p}\left(D_{\mathrm{KL}}(\mu \parallel \nu_J) + \log\frac{1}{\delta}\right)} + U(\mathcal{W}_{pre}, K, N, N', T)\right\},$$

then detailing the term $D_{\mathrm{KL}}(\mu \parallel \nu_J)$, $L(\theta)$ further satisfies,

$$\mathcal{O}\left\{\sqrt{\frac{1}{(K-K')T}\left(\frac{\sigma^2 C(\frac{1}{N_{param}}, T')}{K'} + \log\frac{1}{\delta}\right)} + U(\mathcal{W}_{pre}, K, N, N', T)\right\},$$

where $C(\frac{1}{N_{param}}, T') = \frac{\beta}{2}e^{8\beta S}\left(1 - e^{-\frac{T'}{\exp(8\beta S)}}\right)$, $U(\mathcal{W}_{pre}, K, N, N', T)$ denotes the right hand of equality 6 or equality 38. $\mu$ and $\nu_J$ are the posterior and topic-dependent prior distribution of model parameters $\theta$, respectively. $K(K')$, $N(N')$ and $T$ denote the number of topics, the number of sequences per topic and the sequence length utilized in the optimization process of Equation 1. $T'$ denotes the total training iterations. $N_{param}$ denotes the number of model parameters.

**Remark F.8.** In Theorem F.7, we establish a comprehensive upper bound of population loss combining the results in Theorem F.1, F.3 and F.5.

**In summary, the proof of Theorem 4.6 is provided in Appendix G.3.1 and Appendix G.3.2.**

## G  PROOF OF THEOREMS

### G.1  USEFUL DEFINITIONS, LEMMAS AND PROPOSITIONS

**Definition G.1** (Entropy). For random variable $\theta$, which takes value in $\Theta$ and its probability distribution is $\mu$, the entropy of random variable $\theta$ is

$$H(\theta) = -\sum_{\theta \in \Theta} \mu(\theta) \log \mu(\theta) = \mathbb{E}_{\theta \sim \mu}\left[-\log \mu(\theta)\right].$$

**Definition G.2** (Kullback–Leibler Divergence). The Kullback–Leibler (KL) divergence between two probability distributions $\mu$ and $\nu$ is defined by

$$D_{\mathrm{KL}}(\mu \parallel \nu) = \mathbb{E}_{\theta \sim \mu}\left[\log\frac{\mu(\theta)}{\nu(\theta)}\right].$$

**Lemma G.3** (Donsker–Varadhan representation in Belghazi et al. (2018) Theorem 1). *The KL divergence between probability distribution $\mu$ and $\nu$ obeys the following dual representation:*

$$D_{\mathrm{KL}}(\mu \parallel \nu) = \sup_{T:\mathcal{A}\to\mathbb{R}}\left\{\mathbb{E}_\mu\left[T\right] - \log(\mathbb{E}_\nu[e^T])\right\},$$

*where the compact set $\mathcal{A} \subseteq \mathbb{R}^d$ is the support of distribution $\mu$ and $\nu$, and the supremum is calculated across all functions $T$ for which both expectations are finite.*

Let $\mathcal{F}$ be any class of functions $T : \mathcal{A} \to \mathbb{R}$ satisfying the integrability constraints of the lemma. Then for any defined function $T$, it's straightforward to get the lower-bound of the KL divergence between $\mu$ and $\nu$

$$D_{\mathrm{KL}}(\mu \parallel \nu) \geq \mathbb{E}_\mu\left[T\right] - \log(\mathbb{E}_\nu[e^T]),$$

which would be used in the proof of Theorem F.1.

**Definition G.4** (Total Variation Distance in Levin & Peres (2017)). The total variation (TV) distance between two probability distributions $\mu$ and $\nu$ on events set $\mathcal{B}$ is defined by

$$D_{\mathrm{TV}}(\mu, \nu) = \max_{B \in \mathcal{B}} |\mu(B) - \nu(B)|.$$

This definition is explicitly probabilistic: It quantifies the divergence between $\mu$ and $\nu$ as the maximum disparity in the probabilities assigned to a single event $B$ by the two distributions.

**Proposition G.5** (Total Variation Distance in Levin & Peres (2017)). *Let $\mu$ and $\nu$ be two probability distributions on $\mathcal{A}$. Then*

$$D_{\mathrm{TV}}(\mu, \nu) = \frac{1}{2} \sum_{a \in \mathcal{A}} |\mu(a) - \nu(a)|.$$

*Proof.* Let $A$ be any event and event $B$ be $B = \{a : \mu(a) \geq \nu(a)\}$. Since $A = A \cap (B \cup B^c) = (A \cap B) \cup (A \cap B^c)$, then we have

$$\mu(A) - \nu(A) \leq \mu(A \cap B) - \nu(A \cap B)$$

Since including more elements of $B$ cannot decrease the difference in probability, we have

$$\mu(A \cap B) - \nu(A \cap B) \leq \mu(B) - \nu(B)$$

Combine the above two inequality, we have

$$\mu(A) - \nu(A) \leq \mu(B) - \nu(B)$$

Similarly,

$$\nu(A) - \mu(A) \leq \nu(B^c) - \mu(B^c).$$

Thus

$$D_{\mathrm{TV}}(\mu, \nu) = \frac{1}{2} \left[ \mu(B) - \nu(B) + \nu(B^c) - \mu(B^c) \right] = \frac{1}{2} \sum_{a \in \mathcal{A}} |\mu(a) - \nu(a)|.$$

$\square$

**Lemma G.6** (Lemma 22 in Agarwal et al. (2020)). *For any two conditional probability distribution $\mu$ and $\nu$, we have*

$$D_{\mathrm{TV}}\left(\mu(\cdot|x), \nu(\cdot|x)\right)^2 \leq -2 \log \mathbb{E}_{y \sim \mu(\cdot|x)} \left[ \exp\left( -\frac{1}{2} \log \frac{\mu(y|x)}{\nu(y|x)} \right) \right].$$

This lemma provides an upper bound on the total variation distance, which is related to the expectation of the logarithmic ratio of two conditional probability distributions. It would be used in the proof of Theorem F.1.

**Lemma G.7** (Upper Bound of KL divergence). *For any two conditional probability distribution $\mu$ and $\nu$, if $\frac{\mu(a)}{\nu(a)} \leq C$, we have*

$$D_{\mathrm{KL}}(\mu(a) \| \nu(a)) \leq \frac{2C \log C}{C - 1} D_{\mathrm{TV}}(\mu(a), \nu(a)).$$

This lemma provides the relationship between KL divergence and TV distance.

*Proof.* Let $f(t) = \log t$, $g(t) = |\frac{1}{t} - 1|$. According to the definition of KL divergence and TV distance (see in G.2 and G.5), we have

$$D_{\mathrm{KL}}(\mu(a) \| \nu(a)) = \mathbb{E}_{a \sim \mu} \left[ \log \frac{\mu(a)}{\nu(a)} \right] = \mathbb{E}_{a \sim \mu} \left[ f\left( \frac{\mu(a)}{\nu(a)} \right) \right]$$

$$D_{\mathrm{TV}}(\mu(a), \nu(a)) = \frac{1}{2} \sum_a |\mu(a) - \nu(a)| = \frac{1}{2} \sum_a \mu(a) \left| 1 - \frac{\nu(a)}{\mu(a)} \right| = \frac{1}{2} \mathbb{E}_{a \sim \mu} \left[ g\left( \frac{\mu(a)}{\nu(a)} \right) \right]$$

For $0 < t \leq C (t \neq 1)$, we have

$$\sup_{0 < t \leq C, t \neq 1} \frac{f(t)}{g(t)} = \sup_{0 < t \leq C, t \neq 1} \frac{\log t}{|\frac{1}{t} - 1|} = \sup_{1 < t \leq C} \frac{t \log t}{t - 1}$$

Based on the derivative chain rule, we have that if $1 < t \leq C$, $\frac{t \log t}{t - 1} \leq \frac{C \log C}{C - 1}$. Thus, we conclude that

$$D_{\mathrm{KL}}(\mu(a) \| \nu(a)) \leq \frac{2C \log C}{C - 1} D_{\mathrm{TV}}(\mu(a), \nu(a)).$$

$\square$

**Definition G.8** (Fokker-Planck Equation in Mou et al. (2018)). Let $\pi_t$ be the probability density function of distribution $\mu_t$, then Fokker-Planck Equation describes the evolution of $\pi_t$:

$$\frac{\partial \pi_t}{\partial t} = \frac{1}{\beta} \Delta \pi_t - \nabla \cdot (\pi_t \nabla L_E(\theta_{t-1}, \mathcal{W}_{\text{pre}}))$$

where $\nabla$ is gradient operator and $\Delta$ is Laplace operator.

**Definition G.9** (Gradient Langevin Dynamics and Continuous Langevin Dynamic Li et al. (2019)). LLMs perform Stochastic Gradient Descent (SGD) as optimization algorithm to update parameters $\theta$ in order to get the minimum $\hat{\theta}$. SGD can be viewed as gradient descent addition with gradient noise between full batch gradient and single/mini-batch gradient (Wang & Mao, 2022). The full batch gradient with $\theta_{t-1}$ can be denoted as $\nabla L_E(\theta_{t-1}, \mathcal{W}_{\text{pre}})$, and assume that the gradient noise follows an isotropic Gaussian distribution $\mathcal{N}(0, \frac{I_d}{\beta})$, thus the training dynamic of LLMs can be defined as

$$\theta_t \leftarrow \theta_{t-1} - \eta_t \nabla L_E(\theta_{t-1}, \mathcal{W}_{\text{pre}}) + \sqrt{\frac{\eta_t}{\beta}} \mathcal{N}(0, I_d), \tag{18}$$

which is called Gradient Langevin Dynamics (GLD). When the step size $\eta_t$ in GLD (see in equation 18) approaches zero, the Continuous Langevin Dynamics (CLD) is defined by the following Stochastic Differential Equation (SDE),

$$\mathrm{d}\theta_t = -\nabla L_E(\theta_{t-1}, \mathcal{W}_{\text{pre}})\mathrm{d}t + \sqrt{\beta^{-1}}\,\mathrm{d}B_t, \quad \theta_0 \sim \mu_0 \tag{19}$$

where $B_t$ is the standard brown emotion on $\mathbb{R}^d$.

**Lemma G.10** (Log-Sobolev Inequality (LSI) for Continuous Langevin Dynamic (CLD) in Li et al. (2019) Lemma 16). *Under Equation 19, let $q_t$ be the probability density function of parameter $\theta_t$ in CLD and the initial state obeys $\theta_0 \sim \mathcal{N}(0, \frac{I_d}{\beta})$. Let $p$ be any probability density function which is absolutely continuous with respect to $q_t$. Assume that the optimization objective $L_E(\theta, \mathcal{W}_{pre})$ is $C$-bounded, then we have*

$$D_{\text{KL}}\left(p \parallel q_t\right) \le \frac{\exp(8\beta S)}{2\beta} \int_{\mathbb{R}^d} \left\|\nabla \log \frac{p(\theta)}{q_t(\theta)}\right\|_2^2 p(\theta)\mathrm{d}\theta.$$

Many existing LSIs primarily focus on characterizing the stationary distribution of the Markov process. Contrastly, as shown in this lemma, we try to establish a LSI for $\mu_t$, which denotes the parameter distribution at each time step $t > 0$. It would be used in the proof of Theorem F.3 and F.7 which explores the upper bound of KL divergence carefully to get data-dependent and topic-dependent generalization bound. According to Fokker-Planck Equation in Definition G.8, the KL divergence between two probability density function can be built so that Lemma G.10 can be applied naturally.

**Lemma G.11** (McDiarmid's Inequality Variants for Markov Chains in Paulin (2015) Theorem 2.1). *Consider a Markov chain $X = (X_1, \cdots, X_N)$, which is not necessarily time homogeneous, taking values in a Polish state space $\Lambda = \Lambda_1 \times \cdots \times \Lambda_N$, with mixing time $\tau(\epsilon)$ (for $0 \le \epsilon \le 1$). Let $\tau_{\min} := \inf_{0 \le \epsilon < 1} \tau(\epsilon) \cdot \left(\frac{2-\epsilon}{1-\epsilon}\right)^2, c \in \mathbb{R}_+^N$. If $f : \Lambda \to \mathbb{R}$ satisfies $f(x) - f(y) \le \sum_{i=1}^N c_i \mathbf{1}[x_i \ne y_i]$, Then for any $\lambda \in \mathbb{R}$, we have*

$$\log \mathbb{E}_X\left[\exp\left(\lambda(\mathbb{E}_X[f(X)] - f(X))\right)\right] \le \frac{\lambda^2 \cdot \|c\|_2^2 \cdot \tau_{\min}}{8}.$$

**Proposition G.12** (Refer to Zhang et al. (2023b)). *Define $f(X) = \frac{1}{N}\sum_{i=1}^N f(X_i)$ where $X = (X_1, \cdots, X_N)$ is a Markov chain. With the condition in Lemma G.11, if $|f(X_i)| \le C$ and $f \in \mathcal{F}$, given a prior distribution $\nu$ on $\mathcal{F}$, with probability at least $1 - \delta$*

$$\mathbb{E}_\mu\left[\mathbb{E}_X[f(X)] - f(X)\right] \le \sqrt{\frac{C^2 \cdot \tau_{\min}}{2N \log 2}\left[D_{\text{KL}}(\mu \parallel \nu) + \log \frac{2}{\delta}\right]}$$

*Proof.* With the assumption $|f(X_i)| \leq C$, we have $c_i = \frac{2C}{N}$ in $f(x) - f(y) \leq \sum_{i=1}^{N} c_i \mathbf{1}[x_i \neq y_i]$. Then using Lemma G.11,

$$\log \mathbb{E}_X \left[ \exp \left( \lambda (\mathbb{E}_X[f(X)] - f(X)) \right) \right] \leq \frac{\lambda^2 C^2 \cdot \tau_{\min}}{2N}$$

$$\mathbb{E}_\nu \left[ \mathbb{E}_X \left[ \exp \left( \lambda (\mathbb{E}_X[f(X)] - f(X)) \right) \right] \right] \leq \exp \left( \frac{\lambda^2 C^2 \cdot \tau_{\min}}{2N} \right)$$

$$\mathbb{E}_X \left[ \mathbb{E}_\nu \left[ \exp \left( \lambda (\mathbb{E}_X[f(X)] - f(X)) \right) \right] \right] \leq \exp \left( \frac{\lambda^2 C^2 \cdot \tau_{\min}}{2N} \right) \tag{20}$$

According to Markov inequality $P(X \geq t) \leq \frac{E[X]}{t}$ for random variable $X$ and any $t > 0$, we have

$$P\left( \mathbb{E}_\nu \left[ \exp \left( \lambda (\mathbb{E}_X[f(X)] - f(X)) \right) \right] \geq t \right) \leq \frac{\mathbb{E}_X \left[ \mathbb{E}_\nu \left[ \exp \left( \lambda (\mathbb{E}_X[f(X)] - f(X)) \right) \right] \right]}{t} \tag{21}$$

then substitute inequality 20 into 21,

$$P\left( \mathbb{E}_\nu \left[ \exp \left( \lambda (\mathbb{E}_X[f(X)] - f(X)) \right) \right] \geq t \right) \leq \frac{\exp \left( \frac{\lambda^2 C^2 \cdot \tau_{\min}}{2N} \right)}{t} \tag{22}$$

Let $\lambda = \sqrt{\frac{2N \log 2}{C^2 \cdot \tau_{\min}}}$ and $t = \frac{2}{\delta}$ for any $0 < \delta < 1$, inequality 22 can be transformed into

$$P\left( \mathbb{E}_\nu \left[ \exp \left( \lambda (\mathbb{E}_X[f(X)] - f(X)) \right) \right] \geq \frac{2}{\delta} \right) \leq \delta$$

According to Lemma G.3,

$$D_{\mathrm{KL}}(\mu \parallel \nu) \geq \mathbb{E}_\mu [T] - \log(\mathbb{E}_\nu[e^T])$$

Let $T = \lambda (\mathbb{E}_X[f(X)] - f(X))$, then with probability at least $1 - \delta$, we have

$$\mathbb{E}_\mu \left[ \lambda (\mathbb{E}_X[f(X)] - f(X)) \right] \leq D_{\mathrm{KL}}(\mu \parallel \nu) + \log \left( \mathbb{E}_\nu \left[ \exp \left( \lambda (\mathbb{E}_X[f(X)] - f(X)) \right) \right] \right)$$

$$\mathbb{E}_\mu \left[ \mathbb{E}_X[f(X)] - f(X) \right] \leq \frac{1}{\lambda} \left[ D_{\mathrm{KL}}(\mu \parallel \nu) + \log \frac{2}{\delta} \right]$$

$$\mathbb{E}_\mu \left[ \mathbb{E}_X[f(X)] - f(X) \right] \leq \sqrt{\frac{C^2 \cdot \tau_{\min}}{2N \log 2}} \left[ D_{\mathrm{KL}}(\mu \parallel \nu) + \log \frac{2}{\delta} \right]$$

$\square$

**Lemma G.13** (McDiarmid's Inequality Variants in Luo et al. (2022)). *Consider a function $f$ : $[N]^{N'} \to \mathbb{R}^+$ that is order-independent, where $|f(J) - f(J')| \leq c$ holds for any adjacent sets $J, J' \in [N]^{N'}$ such that there is only one different elements in the two sets. Let $J$ consist of $N'$ indices sampled uniformly without replacement from $[N]$. Then, for any $t \geq 0$,*

$$P\left( |f(J) - \mathbb{E}_J[f(J)]| \geq t \right) \leq \exp \left( \frac{-2t^2}{N'c^2} \right)$$

**Proposition G.14.** *Define $f(X) = \frac{1}{N} \sum_{i=1}^{N} f(X_i)$ where $X = (X_1, \cdots, X_N)$ is a Markov chain. With the condition in Lemma G.11 and if $|f(X_i)| \leq C$, then with probability at least $1 - \delta$*

$$\mathbb{E}_X[f(X)] - f(X) \leq \sqrt{\frac{2C^2 \cdot \tau_{\min} \log \frac{1}{\delta}}{N}}$$

*Proof.* With the assumption $|f(X_i)| \leq C$, we have $c_i = \frac{2C}{N}$ in $f(x) - f(y) \leq \sum_{i=1}^{N} c_i \mathbf{1}[x_i \neq y_i]$. Then using Lemma G.11,

$$\log \mathbb{E}_X \left[ \exp \left( \lambda (\mathbb{E}_X[f(X)] - f(X)) \right) \right] \leq \frac{\lambda^2 C^2 \cdot \tau_{\min}}{2N}$$

$$\mathbb{E}_X \left[ \exp \left( \lambda (\mathbb{E}_X[f(X)] - f(X)) \right) \right] \leq \exp \left( \frac{\lambda^2 C^2 \cdot \tau_{\min}}{2N} \right) \tag{23}$$

According to Chernoff bound $P(X \geq t) \leq \frac{E[e^{\lambda X}]}{e^{\lambda t}}$ for random variable $X$ and any $\lambda > 0$, we have

$$P(\mathbb{E}_X[f(X)] - f(X) \geq t) \leq \frac{\mathbb{E}_X\left[\exp\left(\lambda(\mathbb{E}_X[f(X)] - f(X))\right)\right]}{\exp(\lambda t)} \tag{24}$$

then substitute inequality 23 into 24,

$$P(\mathbb{E}_X[f(X)] - f(X) \geq t) \leq \frac{\exp\left(\frac{\lambda^2 C^2 \cdot \tau_{\min}}{2N}\right)}{\exp(\lambda t)} = \exp\left(\frac{\lambda^2 C^2 \cdot \tau_{\min}}{2N} - \lambda t\right) \tag{25}$$

Let $\lambda = \frac{Nt}{C^2 \cdot \tau_{\min}}$, inequality 25 can be transformed into

$$P(\mathbb{E}_X[f(X)] - f(X) \geq t) \leq \exp\left(\frac{-Nt^2}{2C^2 \cdot \tau_{\min}}\right)$$

Let $t = \sqrt{\frac{2C^2 \cdot \tau_{\min} \log \frac{1}{\delta}}{N}}$ for any $0 < \delta < 1$,

$$P\left(\mathbb{E}_X[f(X)] - f(X) \geq \sqrt{\frac{2C^2 \cdot \tau_{\min} \log \frac{1}{\delta}}{N}}\right) \leq \delta$$

Finally, with probability at least $1 - \delta$,

$$\mathbb{E}_X[f(X)] - f(X) \leq \sqrt{\frac{2C^2 \cdot \tau_{\min} \log \frac{1}{\delta}}{N}}$$

$\square$

## G.2 Generalization of Sequences: The First-Level Expectation

### G.2.1 Proof of Theorem F.1

**Theorem** (Generalization Bound of the First-Level Expected Loss). *Let the auto-regressive LLM $\mathbb{P}_\theta$ be the empirical solution of Equation 1, and $\mathbb{P}(\cdot \mid w)$ denotes the true data distribution under topic $w$. Under Assumptions 4.1, for any $0 < \delta < 1$, with probability at least $1 - \delta$, the first-level expected loss with $K$ topics and infinite sequences per topic, denoted by $L(\theta, \mathcal{W}_{pre})$ (see in Equation 4 or Equation 13), satisfies,*

$$\mathbb{E}_\mu\left[L(\theta, \mathcal{W}_{pre})\right] = \mathcal{O}\left\{\sqrt{\frac{\log 1/\delta}{KNT}} + \sqrt{\frac{1}{KNT}\left(D_{\mathrm{KL}}(\mu \parallel \nu) + \log\frac{1}{\delta}\right)} - \epsilon_{opt}\right\},$$

*where $\epsilon_{opt}$ is the optimization error (see in Equation 3). $\mu$ and $\nu$ are the posterior and prior distribution of model parameters $\theta$, respectively. $K$, $N$ and $T$ denote the number of topics, the number of sequences per topic and the sequence length utilized in the optimization process of Equation 1.*

**Proof sketch.** Before the formal proof, we introduce the processing route to obtain the generalization error bound with handing the prompt token-dependency issue. First, we elaborate on the construction of ghost sequences $\tilde{E}_k$, which are constructed auto-regressively depending on the original sequence $E_k$ thus tokens in ghost sequences are independent. Additionally, we define the function $T = g(\theta, w_k) - \log \mathbb{E}_{\tilde{E}^k}\left[\exp(g(\theta, w_k)) \mid E^k\right]$, where $g(\theta, w_k) = \frac{1}{2}\sum_{n=1}^{N}\sum_{t=1}^{T} \log \frac{\mathbb{P}(x_{t+1}^{k,n} \mid E_t^{k,n}, w_k)}{\mathbb{P}_\theta(x_{t+1}^{k,n} \mid E_t^{k,n}, w_k)}$. It can be observed that this function links the original sequence $E_k$ (with dependent tokens), with the ghost sequences $\tilde{E}_k$ (with independent tokens). Substituting them into the Donsker-Varadhan Inequality facilitates establishing a connection between 'data' and the KL distance between 'model prior' and 'model posterior based on training data'. Furthermore, regarding the coupling term $\log \mathbb{E}_{\tilde{E}^k}\left[\exp(g(\theta, w_k)) \mid E^k\right]$ in the function $T$, we handle this part using the lemma provided in Agarwal et al. (2020), where this part is further transformed into a distribution measure of Total

Variance distance (TV distance). As we mentioned in Section 4, the primary optimization objective 'negative logarithm likelihood' naturally leads to 'KL divergence', thereby formalizing the expression of population loss. Therefore, it's necessary to introduce a relationship between TV distance and KL divergence (See in Lemma G.7), for obtaining our generalization bound. **Overall, the processing route can be summarized as:** 'original sequences $E_k \to$ ghost sequences $\tilde{E}_k \to$ Donsker-Varadhan Inequality $\to$ TV distance $\to$ KL divergence $\to$ the upper bound of population loss based on KL divergence'.

*Proof.* As we introduced before, all the pre-training sequences set is $E = \{E^{k,n}\}_{k,n=1}^{K,N}$, each sequence is denoted as $E^{k,n} = \{(E_t^{k,n}, x_{t+1}^{k,n})\}_{t=1}^{T_{k,n}-1}$ where $x_{t+1}^{k,n} \sim \mathbb{P}(\cdot \mid E_t^{k,n}, w_k)$. To decouple the dependence between tokens, we construct tangent/ghost sequences $\tilde{E} = \{\tilde{E}^{k,n}\}_{k,n=1}^{K,N}$ and each sequence is $\tilde{E}^{k,n} = \{(\tilde{E}_t^{k,n}, \tilde{x}_{t+1}^{k,n})\}_{t=1}^{T_{k,n}-1}$ where $\tilde{x}_{t+1}^{k,n}$ is generated depending on the partial original sequences $E_t^{k,n}$. The construction process of tangent/ghost sequences can be understood simply as duplicate sequences generated based on the original sequences. This proprietary term has been previously utilized in Agarwal et al. (2020); de la Peña et al. (1999); Kwapien & Woyczynski (1991). Therefore, by introducing the ghost sequences into our analysis, this will help decouple the token-dependency in auto-regressive sampling of sequences.

Notice that the following proof is first established under a fixed topic $w_k$.

According to Donsker-Varadhan Inequality (Lemma G.3), let $\mathcal{F}$ be any class of functions $T$ : $\Omega \to \mathbb{R}$ satisfying the integrability constraints of the lemma. Then for any defined function $T$, it's straightforward to get the lower-bound of the KL divergence between $\mu$ and $\nu$

$$D_{\mathrm{KL}}(\mu \parallel \nu) \geq \mathbb{E}_\mu[T] - \log(\mathbb{E}_\nu[e^T]),$$

Under fixed topic $w_k$, the posterior distribution of model parameter $\theta$ is depending on $E^k$ (see in Section 3.1) denoted by $\mu$ and the prior distribution of $\theta$ is denoted by $\nu$. Then, a simple deformation of Lemma G.3 leads to

$$\mathbb{E}_\mu[T] - D_{\mathrm{KL}}(\mu \parallel \nu) \leq \log \mathbb{E}_\nu[\exp(T)]$$
$$\exp\left(\mathbb{E}_\mu[T] - D_{\mathrm{KL}}(\mu \parallel \nu)\right) \leq \mathbb{E}_\nu[\exp(T)]$$

Taking expectation over data distribution $E^k \sim \mathbb{P}(\cdot \mid w_k)$, we have

$$\mathbb{E}_{E^k}\left[\exp\left\{\mathbb{E}_\mu[T] - D_{\mathrm{KL}}(\mu \parallel \nu)\right\}\right] \leq \mathbb{E}_{E^k}\mathbb{E}_\nu[\exp(T)] \tag{26}$$

Let $T = g(\theta, w_k) - \log \mathbb{E}_{\tilde{E}^k}\left[\exp(g(\theta, w_k)) \mid E^k\right]$ where

$$g(\theta, w_k) = \frac{1}{2} \sum_{n=1}^N \sum_{t=1}^T \log \frac{\mathbb{P}(x_{t+1}^{k,n} | E_t^{k,n}, w_k)}{\mathbb{P}_\theta(x_{t+1}^{k,n} | E_t^{k,n}, w_k)}$$

Then for the right hand of inequality 26, we have

$$\begin{aligned}
\mathbb{E}_{E^k}\mathbb{E}_\nu[\exp(T)] &= \mathbb{E}_{E^k}\mathbb{E}_\nu\left[\exp\left(g(\theta, w_k) - \log \mathbb{E}_{\tilde{E}^k}\left[\exp(g(\theta, w_k)) \mid E^k\right]\right)\right] \\
&= \mathbb{E}_\nu\mathbb{E}_{E^k}\left[\exp\left(g(\theta, w_k) - \log \mathbb{E}_{\tilde{E}^k}\left[\exp(g(\theta, w_k)) \mid E^k\right]\right)\right] \\
&= \mathbb{E}_\nu\mathbb{E}_{E^k}\left[\frac{\exp\left(g(\theta, w_k)\right)}{\mathbb{E}_{\tilde{E}^k}\left[\exp(g(\theta, w_k)) \mid E^k\right]}\right] = 1
\end{aligned}$$

Similarly to Agarwal et al. (2020), the last equality follows that the token in tangent sequence $\tilde{E}^k$ is independent conditional on $E^k$, so we have $\mathbb{E}_{\tilde{E}^k}\left[\exp(g(\theta, w_k)) \mid E^k\right] = \mathbb{E}_{\tilde{x}_{t+1}^{k,n} \sim \mathbb{P}(\cdot | E_t^{k,n}, w_k)}\left[\prod_{n=1}^N \prod_{t=1}^T \exp\left(\frac{1}{2} \log \frac{\mathbb{P}(\tilde{x}_{t+1}^{k,n}|E_t^{k,n}, w_k)}{\mathbb{P}_\theta(\tilde{x}_{t+1}^{k,n}|E_t^{k,n}, w_k)}\right)\right]$. Thus, inequality 26 can be transformed to

$$\mathbb{E}_{E^k}\left[\exp\left\{\mathbb{E}_\mu[T] - D_{\mathrm{KL}}(\mu \parallel \nu)\right\}\right] \leq 1 \tag{27}$$

With Markov Inequality $\mathbb{P}[X \geq a] \leq \frac{\mathbb{E}[X]}{a}$, we get the following high probability representation with probability at least $1 - \delta$,

$$\text{let } X = \mathbb{E}_\mu[T] - D_{\mathrm{KL}}(\mu \parallel \nu) \Rightarrow \mathbb{P}[e^X \geq e^a] \leq \frac{\mathbb{E}[e^X]}{e^a} \leq \frac{1}{e^a} \Rightarrow \mathbb{P}(X \leq \log \frac{1}{\delta}) \geq 1 - \delta$$

$$\mathbb{E}_\mu[g(\theta, w_k)] - \mathbb{E}_\mu\left[\log \mathbb{E}_{\tilde{E}^k}\left[\exp(g(\theta, w_k)) \mid E^k\right]\right] \leq D_{\mathrm{KL}}(\mu \parallel \nu) + \log \frac{1}{\delta} \tag{28}$$

For inequality 28, we mainly deal with the left hand in this Theorem and make more detailed analysis of KL divergence in Theorem F.3 to get data-dependent and optimization algorithm-dependent PAC-Bayesian generalization bound.

$$\mathbb{E}_\mu \left[ g(\theta, w_k) \right] - \mathbb{E}_\mu \left[ \log \mathbb{E}_{\tilde{E}^k} \left[ \exp(g(\theta, w_k)) \mid E^k \right] \right]$$

$$\geq \mathbb{E}_\mu \left[ \frac{1}{2} \sum_{n=1}^{N} \sum_{t=1}^{T} \log \frac{\mathbb{P}(x_{t+1}^{k,n} | E_t^{k,n}, w_k)}{\mathbb{P}_\theta(x_{t+1}^{k,n} | S_t^{k,n}, w_k)} \right] -$$

$$\mathbb{E}_\mu \left[ \sum_{n=1}^{N} \sum_{t=1}^{T} \log \mathbb{E}_{\tilde{E}^k} \left[ \exp \left( -\frac{1}{2} \log \frac{\mathbb{P}_\theta(\tilde{x}_{t+1}^{k,n} | E_t^{k,n}, w_k)}{\mathbb{P}(\tilde{x}_{t+1}^{k,n} | E_t^{k,n}, w_k)} \right) \mid E^k \right] \right] \tag{29}$$

Using Lemma G.6, the second term in the right hand can be transformed to the total variation distance (TV distance) of distribution $\mathbb{P}_\theta$ and $\mathbb{P}$.

$$\mathbb{E}_\mu \left[ g(\theta, w_k) \right] - \mathbb{E}_\mu \left[ \log \mathbb{E}_{\tilde{E}^k} \left[ \exp(g(\theta, w_k)) \right] \mid E^k \right]$$

$$\geq \mathbb{E}_\mu \left[ \frac{1}{2} \sum_{n=1}^{N} \sum_{t=1}^{T} \log \frac{\mathbb{P}(x_{t+1}^{k,n} | E_t^{k,n}, w_k)}{\mathbb{P}_{\hat{\theta}}(x_{t+1}^{k,n} | E_t^{k,n}, w_k)} \frac{\mathbb{P}_{\hat{\theta}}(x_{t+1}^{k,n} | E_t^{k,n}, w_k)}{\mathbb{P}_\theta(x_{t+1}^{k,n} | E_t^{k,n}, w_k)} \right] \tag{30}$$

$$+ \mathbb{E}_\mu \left[ \frac{1}{2} \sum_{n=1}^{N} \sum_{t=1}^{T} D_{\mathrm{TV}} \big( \mathbb{P}_\theta(\cdot \mid E_t^{k,n}, w_k), \mathbb{P}(\cdot \mid E_t^{k,n}, w_k) \big)^2 \right]$$

$$= \frac{1}{2} \sum_{n=1}^{N} \sum_{t=1}^{T} \log \frac{\mathbb{P}(x_{t+1}^{k,n} | E_t^{k,n}, w_k)}{\mathbb{P}_{\hat{\theta}}(x_{t+1}^{k,n} | E_t^{k,n}, w_k)} + \frac{1}{2} \sum_{n=1}^{N} \sum_{t=1}^{T} \mathbb{E}_\mu \left[ \log \frac{\mathbb{P}_{\hat{\theta}}(x_{t+1}^{k,n} | E_t^{k,n}, w_k)}{\mathbb{P}_\theta(x_{t+1}^{k,n} | E_t^{k,n}, w_k)} \right]$$

$$+ \mathbb{E}_\mu \left[ \frac{1}{2} \sum_{n=1}^{N} \sum_{t=1}^{T} D_{\mathrm{TV}} \big( \mathbb{P}_\theta(\cdot \mid E_t^{k,n}, w_k), \mathbb{P}(\cdot \mid E_t^{k,n}, w_k) \big)^2 \right]$$

$$\geq \frac{1}{2} \sum_{n=1}^{N} \sum_{t=1}^{T} \mathbb{E}_\mu \left[ \log \frac{\mathbb{P}_{\hat{\theta}}(x_{t+1}^{k,n} | E_t^{k,n}, w_k)}{\mathbb{P}_\theta(x_{t+1}^{k,n} | E_t^{k,n}, w_k)} \right]$$

$$+ \mathbb{E}_\mu \left[ \frac{1}{2} \sum_{n=1}^{N} \sum_{t=1}^{T} D_{\mathrm{TV}} \big( \mathbb{P}_\theta(\cdot \mid E_t^{k,n}, w_k), \mathbb{P}(\cdot \mid E_t^{k,n}, w_k) \big)^2 \right] \tag{31}$$

where $\hat{\theta}$ is the minimum of empirical loss 1. Thus, substitute inequality 31 into 28,

$$\mathbb{E}_\mu \left[ \frac{1}{2} \sum_{n=1}^{N} \sum_{t=1}^{T} D_{\mathrm{TV}} \big( \mathbb{P}_\theta(\cdot \mid E_t^{k,n}, w_k), \mathbb{P}(\cdot \mid E_t^{k,n}, w_k) \big)^2 \right]$$

$$+ \frac{1}{2} \sum_{n=1}^{N} \sum_{t=1}^{T} \mathbb{E}_\mu \left[ \log \frac{\mathbb{P}_{\hat{\theta}}(x_{t+1}^{k,n} | E_t^{k,n}, w_k)}{\mathbb{P}_\theta(x_{t+1}^{k,n} | E_t^{k,n}, w_k)} \right] \leq D_{\mathrm{KL}}(\mu \parallel \nu) + \log \frac{1}{\delta} \tag{32}$$

The result of inequality 32 is analysised under a fixed topic $w_k$, then combining all $w_k \in \mathcal{W}_{\mathrm{pre}}$ and taking average

$$\mathbb{E}_\mu \left[ \frac{1}{KNT} \sum_{k=1}^{K} \sum_{n=1}^{N} \sum_{t=1}^{T} D_{\mathrm{TV}} \big( \mathbb{P}_\theta(\cdot \mid E_t^{k,n}, w_k), \mathbb{P}(\cdot \mid E_t^{k,n}, w_k) \big)^2 \right]$$

$$\leq \frac{2}{KNT} \left( D_{\mathrm{KL}}(\mu \parallel \nu) + \log \frac{1}{\delta} \right) - \underbrace{\frac{1}{KNT} \sum_{k=1}^{K} \sum_{n=1}^{N} \sum_{t=1}^{T} \mathbb{E}_\mu \left[ \log \frac{\mathbb{P}_{\hat{\theta}}(x_{t+1}^{k,n} | E_t^{k,n}, w_k)}{\mathbb{P}_\theta(x_{t+1}^{k,n} | E_t^{k,n}, w_k)} \right]}_{\epsilon_{\mathrm{opt}}} \tag{33}$$

where the second term in the right hand is denoted as $\epsilon_{\mathrm{opt}}$ measuring the logarithmic distribution distance between the ideal minimum $\hat{\theta}$ and the trained model $\theta$ with empirical loss. Specially, we defer the analysis of optimization error to future work. Here, we assume that the results of the actual models obtained closely approximates the ideal minimum for empirical loss, implying that $\epsilon_{\mathrm{opt}}$ is a

very small value so that this item will be kept in the upper bounds of the first-level expected loss and two-level expected loss. Thus,

$$
\mathbb{E}_\mu \left[ \frac{1}{KNT} \sum_{k=1}^{K} \sum_{n=1}^{N} \sum_{t=1}^{T} D_{\mathrm{TV}} \big( \mathbb{P}_\theta(\cdot \mid E_t^{k,n}, w_k), \mathbb{P}(\cdot \mid E_t^{k,n}, w_k) \big) \right]
$$

$$
\leq \sqrt{ \mathbb{E}_\mu \left[ \frac{1}{KNT} \sum_{k=1}^{K} \sum_{n=1}^{N} \sum_{t=1}^{T} D_{\mathrm{TV}} \big( \mathbb{P}_\theta(\cdot \mid E_t^{k,n}, w_k), \mathbb{P}(\cdot \mid E_t^{k,n}, w_k) \big)^2 \right] }
$$

$$
\leq \sqrt{ \frac{2 \left( D_{\mathrm{KL}}(\mu \parallel \nu) + \log \frac{1}{\delta} \right)}{KNT} } - \epsilon_{\mathrm{opt}} \tag{34}
$$

Using Assumption 4.1, assume $\log \frac{\mathbb{P}(\cdot \mid E_t^{k,n}, w_k)}{\mathbb{P}_\theta(\cdot \mid E_t^{k,n}, w_k)}$ is upper bounded by $C$. Thus using Proposition G.14, with probability at least $1 - \delta$,

$$
\mathbb{E}_\mu \left[ \frac{1}{KNT} \sum_{k=1}^{K} \sum_{n=1}^{N} \sum_{t=1}^{T} \mathbb{E}_{E_t^{k,n}} \left[ D_{\mathrm{TV}} \big( \mathbb{P}_\theta(\cdot \mid E_t^{k,n}, w_k), \mathbb{P}(\cdot \mid E_t^{k,n}, w_k) \big) \right] \right.
$$

$$
\left. - \frac{1}{KNT} \sum_{k=1}^{K} \sum_{n=1}^{N} \sum_{t=1}^{T} D_{\mathrm{TV}} \big( \mathbb{P}_\theta(\cdot \mid E_t^{k,n}, w_k), \mathbb{P}(\cdot \mid E_t^{k,n}, w_k) \big) \right] \leq \sqrt{ \frac{2C^2 \cdot \tau_{\min} \log \frac{1}{\delta}}{KNT} } \tag{35}
$$

Finally, according to Equation 14, 13 and 1, the generalization error bound of the first-level expected loss is $\mathrm{gen}_{\mathrm{seq}} = L(\theta, \mathcal{W}_{\mathrm{pre}}) - L_E(\theta, \mathcal{W}_{\mathrm{pre}})$. Combining inequality 34, 35 and Lemma G.7, $L(\theta, \mathcal{W}_{\mathrm{pre}})$ can be bounded by

$$
\mathbb{E}_\mu \left[ \frac{1}{KNT} \sum_{k=1}^{K} \sum_{n=1}^{N} \sum_{t=1}^{T} \mathbb{E}_{E_t^{k,n}} \left[ D_{\mathrm{KL}} \big( \mathbb{P}(\cdot \mid E_t^{k,n}, w_k) \parallel \mathbb{P}_\theta(\cdot \mid E_t^{k,n}, w_k) \big) \right] \right]
$$

$$
\leq \frac{2C \log C}{C - 1} \cdot \mathbb{E}_\mu \left[ \frac{1}{KNT} \sum_{k=1}^{K} \sum_{n=1}^{N} \sum_{t=1}^{T} \mathbb{E}_{E_t^{k,n}} \left[ D_{\mathrm{TV}} \big( \mathbb{P}_\theta(\cdot \mid E_t^{k,n}, w_k), \mathbb{P}(\cdot \mid E_t^{k,n}, w_k) \big) \right] \right]
$$

$$
\leq \frac{2C \log C}{C - 1} \left( \sqrt{ \frac{2C^2 \cdot \tau_{\min} \log \frac{1}{\delta}}{KNT} } + \sqrt{ \frac{2 \left( D_{\mathrm{KL}}(\mu \parallel \nu) + \log \frac{1}{\delta} \right)}{KNT} } - \epsilon_{\mathrm{opt}} \right)
$$

$$
= \mathcal{O} \left\{ \sqrt{ \frac{\log \frac{1}{\delta}}{KNT} } + \sqrt{ \frac{D_{\mathrm{KL}}(\mu \parallel \nu) + \log \frac{1}{\delta}}{KNT} } - \epsilon_{\mathrm{opt}} \right\} \tag{36}
$$

Naturally, to simplify, for given any prefix sequence $P$, we have

$$
\mathbb{E}_\mu \left[ \frac{1}{K} \sum_{k=1}^{K} \mathbb{E}_P \left[ D_{\mathrm{KL}} \big( \mathbb{P}(\cdot \mid P, w_k) \parallel \mathbb{P}_\theta(\cdot \mid P, w_k) \big) \right] \right]
$$

$$
= \mathcal{O} \left\{ \sqrt{ \frac{\log \frac{1}{\delta}}{KNT} } + \sqrt{ \frac{1}{KNT} \left( D_{\mathrm{KL}}(\mu \parallel \nu) + \log \frac{1}{\delta} \right) } - \epsilon_{\mathrm{opt}} \right\} \tag{37}
$$

$\square$

### G.2.2 Proof of Theorem F.3

**Theorem** (Data-Dependent and Optimization-Dependent Generalization Bound of the First-Level Expected Loss)**.** *Under the conditions maintained in Theorem F.1 and Assumption 4.2, when considering data-dependent prior $\mu_J$, for any $0 < \delta < 1$, with probability at least $1 - \delta$, the first-level*

*expected loss with $K$ topics and infinite sequences per topic, denoted by $L(\theta, \mathcal{W}_{pre})$ (see in Equation 4 or Equation 13), satisfies,*

$$\mathbb{E}_\mu \left[ L(\theta, \mathcal{W}_{pre}) \right] = \mathcal{O} \left\{ \sqrt{\frac{\log 1/\delta}{K(N-N')T}} + \sqrt{\frac{1}{K(N-N')T} \left( D_{\mathrm{KL}}(\mu \| \nu_J) + \log \frac{1}{\delta} \right)} - \epsilon_{opt} \right\},$$

*then detailing the term $D_{\mathrm{KL}}(\mu \| \nu_J)$, $L(\theta, \mathcal{W}_{pre})$ further satisfies,*

$$\mathcal{O} \left\{ \sqrt{\frac{\log 1/\delta}{K(N-N')T}} + \sqrt{\frac{1}{K(N-N')T} \left[ \frac{L^2 C(\frac{1}{N_{param}}, T')}{N'} + \log \frac{1}{\delta} \right]} - \epsilon_{opt} \right\}, \quad (38)$$

*where $C(\frac{1}{N_{param}}, T') = \frac{\beta}{2} e^{8\beta S} \left( 1 - e^{-\frac{T'}{\exp(8\beta S)}} \right)$. $\epsilon_{opt}$ is the optimization error (see in Equation 3). $K$, $N(N')$ and $T$ denote the number of topics, the number of sequences per topic and the sequence length utilized in the optimization process of Equation 1. $T'$ denotes the total training iterations. $N_{param}$ denotes the number of model parameters.*

**Proof sketch of the use of continous mathematical analysis techniques.** We analyse the training dynamic of transformer via Continuous Langevin Dynamics (CLD) which is the continous form of Gradient Langevin Dynamics (GLD). To bound the KL divergence of two distribution, we transform the problem into measuring the KL divergence of pdfs. We first derive the derivative of KL divergence w.r.t. time $t$. This derivative can be decomposed into two parts, corresponding to the time derivatives of the two pdfs, which can be described by the Fokker-Planck Equation. Next, using Log-Sobolev Inequality, we bound the logorithm distance of two pdfs. By solvin the SDE, we obtain an upper bound for the KL divergence. Finally, referring to the proof of Lemma G.5 in Li et al. (2019), we demonstrate that the integral of the gradient difference of $\left\| \nabla L_{E_I}(\theta_t, \mathcal{W}_{\mathrm{pre}}) - \nabla L_{E_J}(\theta_t, \mathcal{W}_{\mathrm{pre}}) \right\|_2^2$. Consequently, we get data-dependent and optimization-dependent generalization bound.

*Proof.* In this Theorem, we analysis KL divergence to get data-dependent and optimization algorithm-dependent generalization bound. First, we analyse the training dynamic of transformer via Continuous Langevin Dynamics (CLD),

$$\mathrm{d}\theta_t = -\nabla L_{E_I}(\theta_{t-1}, \mathcal{W}_{\mathrm{pre}})\mathrm{d}t + \sqrt{\beta^{-1}}\,\mathrm{d}B_t, \quad \theta_0 \sim \mu_0$$

where $L_{E_I}(\theta, \mathcal{W}_{\mathrm{pre}}) = \frac{1}{K(N-N')T} \sum_{k=1}^{K} \sum_{n=1}^{N-N'} \sum_{t=1}^{T} \log \frac{\mathbb{P}(x_{t+1}^{k,n}|E_t^{k,n}, w_k)}{\mathbb{P}_\theta(x_{t+1}^{k,n}|E_t^{k,n}, w_k)}$, and $B_t$ is the standard brown emotion.

Split pre-training sequences under fixed topic $w_k$ into two parts $E_I^k$ and $E_J^k$ (where $J$ is a random sequence including $N'$ indexes uniformly sampled from $[N]$ without replacement and $I$ is $[N] \setminus J$). Under pre-training topics, we have $E_I = \{E_I^k\}_{k=1}^K$ and $E_J = \{E_J^k\}_{k=1}^K$. Assume that the prior distribution of model parameters $\theta$ is depending on the subset $E_J^k$, which is denoted by $\nu_J$ and the posterior distribution of $\theta$ is depending on $E_I^k$ denoted by $\mu$.

Let $\Theta = (\theta_t)_{t \geq 0}$ and $\Theta' = (\theta'_t)_{t \geq 0}$ be the trajectory trained on sequences $E_I^k$ and $E_J^k$ for fixed topic $w_k$, which are the parallel training process based on the same model architecture. Let $\mu$ and $\nu_J$ be the distribution of $\Theta$ and $\Theta'$ respectively, $p_t$ and $q_t$ be the pdf of $\Theta$ and $\Theta'$ and the total steps of iteration is $T'$. $D_{\mathrm{KL}}(\mu \| \nu_J)$ is equal to $D_{\mathrm{KL}}(p_{T'} \| q_{T'})$. To bound $D_{\mathrm{KL}}(p_{T'} \| q_{T'})$, we first apply Leibniz's rule and the chain rule on it:

$$\begin{aligned}
\frac{\mathrm{d}}{\mathrm{d}t} D_{KL}(p_t \| q_t) &= \frac{\mathrm{d}}{\mathrm{d}t} \int_{\mathbb{R}^d} p_t \log \frac{p_t}{q_t} \mathrm{d}\theta \\
&= \int_{\mathbb{R}^d} \left( \frac{\mathrm{d}p_t}{\mathrm{d}t} \log \frac{p_t}{q_t} + p_t \cdot \frac{q_t}{p_t} \cdot \frac{\frac{\mathrm{d}p_t}{\mathrm{d}t} q_t - p_t \frac{\mathrm{d}q_t}{\mathrm{d}t}}{q_t^2} \right) \mathrm{d}\theta \\
&= \underbrace{\int_{\mathbb{R}^d} \frac{\mathrm{d}p_t}{\mathrm{d}t} \log \frac{p_t}{q_t} \mathrm{d}\theta}_{(A)} - \underbrace{\int_{\mathbb{R}^d} \frac{p_t}{q_t} \frac{\mathrm{d}q_t}{\mathrm{d}t} \mathrm{d}\theta}_{(B)},
\end{aligned}$$

where the last equality follows from that $\int \frac{dp_t}{dt} d\theta = \frac{d}{dt} \int p_t d\theta = 0$, since $p_t$ is a probability measure. By Fokker-Planck Equation for $p_t$, $\frac{\partial p_t}{\partial t} = \frac{1}{\beta} \Delta p_t + \nabla \cdot (p_t \nabla L_{E_I}(\theta, \mathcal{W}_{\text{pre}}))$.

Then we bound term $A$,

$$
\begin{aligned}
A &:= \int_{\mathbb{R}^d} \left( \frac{dp_t}{dt} \log \frac{p_t}{q_t} \right) d\theta \\
&= \int_{\mathbb{R}^d} \left( \frac{1}{\beta} \Delta p_t + \nabla \cdot (p_t L_{E_I}(\theta, \mathcal{W}_{\text{pre}})) \right) \log \frac{p_t}{q_t} d\theta \\
&= \frac{1}{\beta} \left[ \int_{\mathbb{R}^d} \Delta p_t \log \frac{p_t}{q_t} d\theta \right] + \int_{\mathbb{R}^d} \nabla \cdot (p_t L_{E_I}(\theta, \mathcal{W}_{\text{pre}})) \log \frac{p_t}{q_t} d\theta \\
&= \frac{1}{\beta} \left[ \nabla p_t \log \frac{p_t}{q_t} - \int_{\mathbb{R}^d} \langle \nabla \log \frac{p_t}{q_t}, \nabla p_t \rangle d\theta \right] \\
&\quad + \left[ p_t \nabla L_{E_I}(\theta, \mathcal{W}_{\text{pre}}) \log \frac{p_t}{q_t} - \int_{\mathbb{R}^d} \langle \nabla \log \frac{p_t}{q_t}, p_t L_{E_I}(\theta, \mathcal{W}_{\text{pre}}) \rangle d\theta \right] \\
&= \frac{-1}{\beta} \int_{\mathbb{R}^d} \langle \nabla \log \frac{p_t}{q_t}, \nabla p_t \rangle d\theta - \int_{\mathbb{R}^d} \langle \nabla \log \frac{p_t}{q_t}, p_t L_{E_I}(\theta, \mathcal{W}_{\text{pre}}) \rangle d\theta
\end{aligned}
$$

Bound term $B$,

$$
\begin{aligned}
B &:= \int_{\mathbb{R}^d} \left( \frac{p_t}{q_t} \frac{dq_t}{dt} \right) dw \\
&= \int_{\mathbb{R}^d} \frac{p_t}{q_t} \left( \frac{1}{\beta} \Delta q_t + \nabla \cdot (q_t \nabla L_{E_J}(\theta, \mathcal{W}_{\text{pre}})) \right) dw \\
&= \frac{1}{\beta} \left[ \int_{\mathbb{R}^d} \frac{p_t}{q_t} \Delta q_t dw \right] + \int_{\mathbb{R}^d} \frac{p_t}{q_t} \nabla \cdot (q_t \nabla L_{E_J}(\theta, \mathcal{W}_{\text{pre}})) dw \\
&= \frac{1}{\beta} \left[ \frac{p_t}{q_t} \nabla q_t - \int_{\mathbb{R}^d} \langle \nabla \frac{p_t}{q_t}, \nabla q_t \rangle dw \right] + \left[ \frac{p_t}{q_t} q_t \nabla L_{E_J}(\theta, \mathcal{W}_{\text{pre}}) - \int_{\mathbb{R}^d} \langle \nabla \frac{p_t}{q_t}, q_t \nabla L_{E_J}(\theta, \mathcal{W}_{\text{pre}}) \rangle dw \right] \\
&= \frac{-1}{\beta} \int_{\mathbb{R}^d} \langle \nabla \frac{p_t}{q_t}, \nabla q_t \rangle dw - \int_{\mathbb{R}^d} \langle \nabla \frac{p_t}{q_t}, q_t \nabla L_{E_J}(\theta, \mathcal{W}_{\text{pre}}) \rangle dw
\end{aligned}
$$

In summary, the deviation of $D_{KL}(p_t || q_t)$ can be bounded,

$$
\begin{aligned}
&\frac{d}{dt} D_{KL}(p_t || q_t) \\
&= \frac{-1}{\beta} \int_{\mathbb{R}^d} \langle \nabla \log \frac{p_t}{q_t}, \nabla p_t \rangle dw - \int_{\mathbb{R}^d} \langle \nabla \log \frac{p_t}{q_t}, p_t \nabla L_{E_I}(\theta, \mathcal{W}_{\text{pre}}) \rangle dw \\
&\quad + \frac{1}{\beta} \int_{\mathbb{R}^d} \langle \nabla \frac{p_t}{q_t}, \nabla q_t \rangle dw + \int_{\mathbb{R}^d} \langle \nabla \frac{p_t}{q_t}, q_t \nabla L_{E_J}(\theta, \mathcal{W}_{\text{pre}}) \rangle dw \\
&= \frac{-1}{\beta} \int_{\mathbb{R}^d} \left( \langle \nabla \log \frac{p_t}{q_t}, \nabla p_t \rangle - \langle \nabla \frac{p_t}{q_t}, \nabla q_t \rangle \right) dw \\
&\quad - \int_{\mathbb{R}^d} \left( \langle \nabla \log \frac{p_t}{q_t}, p_t \nabla L_{E_I}(\theta, \mathcal{W}_{\text{pre}}) \rangle - \langle \nabla \frac{p_t}{q_t}, q_t \nabla L_{E_J}(\theta, \mathcal{W}_{\text{pre}}) \rangle \right) dw \\
&= \frac{-1}{\beta} \int_{\mathbb{R}^d} \left( \langle \frac{\nabla p_t}{p_t} - \frac{\nabla q_t}{q_t}, \nabla p_t \rangle - \langle \frac{\nabla p_t}{q_t} - \frac{p_t \nabla q_t}{q_t^2}, \nabla q_t \rangle \right) dw \\
&\quad - \int_{\mathbb{R}^d} \left( \langle \nabla \log \frac{p_t}{q_t}, p_t \nabla L_{E_I}(\theta, \mathcal{W}_{\text{pre}}) \rangle - \frac{p_t}{q_t} \langle \nabla \log \frac{p_t}{q_t}, q_t \nabla L_{E_J}(\theta, \mathcal{W}_{\text{pre}}) \rangle \right) dw \\
&= \frac{-1}{\beta} \int_{\mathbb{R}^d} p_t \| \nabla \log \frac{p_t}{q_t} \|_2^2 dw + \int_{\mathbb{R}^d} p_t \langle \nabla \log \frac{p_t}{q_t}, \nabla L_{E_I}(\theta, \mathcal{W}_{\text{pre}}) - \nabla L_{E_J}(\theta, \mathcal{W}_{\text{pre}}) \rangle dw
\end{aligned}
$$

Since for any constant $c \neq 0$, vector $\alpha$ and $\beta$, we have the inequality $\langle \frac{\alpha}{\sqrt{c}}, \beta \sqrt{c} \rangle \leq \frac{\|\alpha\|^2}{2c} + \frac{c\|\beta\|^2}{2}$, then we can transform the last equality into

$$
\frac{d}{dt} D_{KL}(p_t || q_t) \leq \frac{-1}{2\beta} \int_{\mathbb{R}^d} p_t \| \nabla \log \frac{p_t}{q_t} \|_2^2 dw + \frac{\beta}{2} \int_{\mathbb{R}^d} p_t \| \nabla L_{E_I}(\theta_t, \mathcal{W}_{\text{pre}}) - \nabla L_{E_J}(\theta_t, \mathcal{W}_{\text{pre}}) \|_2^2 dw
$$

According to Lemma G.10 (Log-Sobolev Inequality for CLD) and Assumption 4.1, then $|L_E(\theta, \mathcal{W}_{\text{pre}})| \leq S$ and $\theta_0 \sim \mathcal{N}(0, \frac{1}{\beta} I_d)$, we have $\int_{\mathbb{R}^d} p_t \| \nabla \log \frac{p_t}{q_t} \|_2^2 d\theta \geq \frac{2\beta}{\exp(8\beta S)} D_{KL}(p_t || q_t)$. Transform the first term in the

right hand with the LSI inequality,

$$\frac{\mathrm{d}}{\mathrm{d}t} D_{KL}(p_t\|q_t) \leq -\frac{1}{\exp(8\beta S)} D_{\mathrm{KL}}(p_t\|q_t) + \frac{\beta}{2}\mathbb{E}_{\theta_t}\left[\|\nabla L_{E_I}(\theta_t, \mathcal{W}_{\mathrm{pre}}) - \nabla L_{E_J}(\theta_t, \mathcal{W}_{\mathrm{pre}})\|_2^2\right]$$

Let $\phi(t) = D_{KL}(p_t\|q_t)$, $\delta(t) = \frac{\beta}{2}\mathbb{E}_{\theta_t}\left[\|\nabla L_{E_I}(\theta_t, \mathcal{W}_{\mathrm{pre}}) - \nabla L_{E_J}(\theta_t, \mathcal{W}_{\mathrm{pre}})\|_2^2\right]$, $\alpha = \frac{1}{\exp(8\beta S)}$, then we get the following difference equation:

$$\phi'(t) = -\alpha\phi(t) + \delta(t), \ \phi(0) = 0$$

Solve the equation:

$$D_{KL}(p_{T'}\|q_{T'}) \leq \frac{\beta}{2}\int_0^{T'} e^\alpha(t - T')\mathbb{E}_{\theta_t}\left[\|\nabla L_{E_I}(\theta_t, \mathcal{W}_{\mathrm{pre}}) - \nabla L_{E_J}(\theta_t, \mathcal{W}_{\mathrm{pre}})\|_2^2\right]\mathrm{d}t, \ \alpha = \frac{1}{\exp(8\beta S)}.$$

Furthermore, in order to get the upper bound of integral in the right hand, we first define that

$$G(J) = \sqrt{\mathbb{E}_{\theta_t}\left[\int_0^{T'} e^{\alpha(t - T')}\left[\|\nabla L_{E_I}(\theta_t, \mathcal{W}_{\mathrm{pre}}) - \nabla L_{E_J}(\theta_t, \mathcal{W}_{\mathrm{pre}})\|_2^2\right]\mathrm{d}t\right]}$$

Let $J$ and $J'$ be two neighboring collections, we first prove that $G(J) - G(J')$ is small. Let $X_t = \nabla L_{E_I}(\theta_t, \mathcal{W}_{\mathrm{pre}}) - \nabla L_{E_J}(\theta_t, \mathcal{W}_{\mathrm{pre}})$, $Y_t = \nabla L_{E_J}(\theta_t, \mathcal{W}_{\mathrm{pre}}) - \nabla L_{E_{J'}}(\theta_t, \mathcal{W}_{\mathrm{pre}})$. Then,

$$G(J')^2 = \mathbb{E}_{\theta_t}\left[\int_0^{T'} e^{\alpha(t-T')}\|X_t + Y_t\|_2^2\ \mathrm{d}t\right]$$

$$= \mathbb{E}_{\theta_t}\left[\int_0^{T'} e^{\alpha(t-T')}\left(X_t^\top X_t + Y_t^\top Y_t\right)\mathrm{d}t\right] + 2\mathbb{E}_{\theta_t}\left[\int_0^{T'} e^{\alpha(t-T')}X_t^\top Y_t\mathrm{d}t\right]$$

$$\leq \mathbb{E}_{\theta_t}\left[\int_0^{T'} e^{\alpha(t-T')}\left(\|X_t\|_2^2 + \|Y_t\|_2^2\right)\ \mathrm{d}t\right]$$

$$+ 2\sqrt{\mathbb{E}_{\theta_t}\left[\int_0^{T'} e^{\alpha(t-T')}\|X_t\|_2^2\mathrm{d}t\right]}\sqrt{\mathbb{E}_{\theta_t}\left[\int_0^{T'} e^{\alpha(t-T')}\|Y_t\|_2^2\mathrm{d}t\right]}$$

$$= \left(\sqrt{\mathbb{E}_{\theta_t}\left[\int_0^{T'} e^{\alpha(t-T')}\|X_t\|_2^2\mathrm{d}t\right]} + \sqrt{\mathbb{E}_{\theta_t}\left[\int_0^{T'} e^{\alpha(t-T')}\|Y_t\|_2^2\mathrm{d}t\right]}\right)^2$$

$$= \left(G(J) + \sqrt{\mathbb{E}_{\theta_t}\left[\int_0^{T'} e^{\alpha(t-T')}\|Y_t\|_2^2\mathrm{d}t\right]}\right)^2$$

For any fixed $J$ and $\theta_t$, under Assumption 4.2 that $\left\|\nabla L_{E_t^{k,n}}(\theta_t, \mathcal{W}_{\mathrm{pre}})\right\| \leq L$, then

$$\int_0^{T'} e^{\alpha(t-T')}\|Y_t\|_2^2\mathrm{d}t \leq \int_0^{T'} e^{\alpha(t-T')}\frac{4L^2}{(KN')^2}\mathrm{d}t = \frac{4L^2(1 - e^{-\alpha T'})}{(KN')^2\alpha}$$

Then,

$$\left|G(J) - G(J')\right| \leq \frac{2L}{KN'}\sqrt{\frac{1 - e^{-\alpha T}}{\alpha}}$$

Applying Lemma G.13 of concentration inequality and there are $N'$ indexes in $J$ or $J'$,

$$P_J\left[G(J) - \mathbb{E}_J[G(J)] \geq \epsilon\right] \leq \exp\left(\frac{-2\epsilon^2}{N'\frac{4L^2(1-e^{-\alpha T'})}{(KN')^2\alpha}}\right) = \exp\left(\frac{-K^2N'\alpha\epsilon^2}{2L^2(1 - e^{-\alpha T'})}\right)$$

We also have,

$$P_J\left[G(J)^2 \geq (\mathbb{E}_J[G(J)] + \epsilon)^2\right] \leq \exp\left(\frac{-K^2N'\alpha\epsilon^2}{2L^2(1 - e^{-\alpha T'})}\right)$$

Then referring to Li et al. (2019), we can easily get the upper bound of variance of $\nabla L_{E_J}(\theta_t, \mathcal{W}_{\text{pre}})$ which is $\mathbb{E}_J\left[\|\nabla L_{E_I}(\theta_t, \mathcal{W}_{\text{pre}}) - \nabla L_{E_J}(\theta_t, \mathcal{W}_{\text{pre}})\|_2^2\right] \leq \frac{4L^2}{N'}$, thus

$$\mathbb{E}_J[G(J)] = \mathbb{E}_J\sqrt{\mathbb{E}_{\theta_t}\left[\int_0^{T'} e^{\alpha(t-T')}\left[\|\nabla L_{E_I}(\theta_t, \mathcal{W}_{\text{pre}}) - \nabla L_{E_J}(\theta_t, \mathcal{W}_{\text{pre}})\|_2^2\right]\mathrm{d}t\right]}$$

$$\leq \sqrt{\mathbb{E}_{\theta_t}\left[\int_0^{T'} e^{\alpha(t-T')}\mathbb{E}_J\left[\|\nabla L_{E_I}(\theta_t, \mathcal{W}_{\text{pre}}) - \nabla L_{E_J}(\theta_t, \mathcal{W}_{\text{pre}})\|_2^2\right]\mathrm{d}t\right]}$$

$$\leq \sqrt{\int_0^{T'} e^{\alpha(t-T')}\frac{4L^2}{N'}}$$

$$= \frac{2L}{\sqrt{N'}}\sqrt{\frac{1-e^{-\alpha T'}}{\alpha}}$$

Let $\exp\left(\frac{-K^2 N'\alpha\epsilon^2}{2L^2(1-e^{-\alpha T'})}\right) = \delta$, then $\epsilon = \sqrt{\frac{2L^2(1-e^{-\alpha T'})\log\frac{1}{\delta}}{K^2 N'\alpha}}$. It follows that with probability at least $1-\delta$

$$G(J)^2 \leq \left(\frac{2L}{\sqrt{N'}}\sqrt{\frac{1-e^{-\alpha T'}}{\alpha}} + \sqrt{\frac{2L^2(1-e^{-\alpha T'})\log\frac{1}{\delta}}{K^2 N'\alpha}}\right)^2$$

Then,

$$\mathbb{E}_{\theta_t}\left[\int_0^{T'} e^{\alpha}(t-T')\left[\|\nabla L_{E_I}(\theta_t, \mathcal{W}_{\text{pre}}) - \nabla L_{E_J}(\theta_t, \mathcal{W}_{\text{pre}})\|_2^2\right]\mathrm{d}t\right]$$

$$\leq \left(\frac{2L}{\sqrt{N'}}\sqrt{\frac{1-e^{-\alpha T'}}{\alpha}} + \sqrt{\frac{2L^2(1-e^{-\alpha T'})\log\frac{1}{\delta}}{K^2 N'\alpha}}\right)^2$$

$$= \frac{4L^2}{N'}\left(1 + \sqrt{\frac{\log\frac{1}{\delta}}{2K^2}}\right)^2\frac{(1-e^{-\alpha T'})}{\alpha}$$

$$= \frac{4L^2}{N'}\left(1 + \sqrt{\frac{\log\frac{1}{\delta}}{2K^2}}\right)^2 e^{8\beta S}\left(1 - \exp\left(-\frac{T'}{e^{8\beta S}}\right)\right)$$

We bound the KL-divergence.

$$D_{\text{KL}}\left(p_{T'} \parallel q_{T'}\right) \leq \left(1 + \sqrt{\frac{\log\frac{1}{\delta}}{2K^2}}\right)^2\frac{2L^2\beta e^{8\beta S}(1 - \exp(-\frac{T'}{e^{8\beta S}}))}{N'} \tag{39}$$

$$= \left(1 + \sqrt{\frac{\log\frac{1}{\delta}}{2K^2}}\right)^2\frac{4L^2 C(\frac{1}{N_{\text{param}}}, T')}{N'} \tag{40}$$

where $C(\frac{1}{N_{\text{param}}}, T') = \frac{\beta}{2}e^{8\beta S}\left(1 - e^{-\frac{T'}{\exp(8\beta S)}}\right)$.

As introduced before, the prior distribution of model parameters $\theta$ is depending on the subset $E_J^k$, which is denoted by $\nu_J$ and the posterior distribution of $\theta$ is depending on $E_I^k$ denoted by $\mu$. Then Theorem F.1 can be transformed to (modify $N$ to $N - N'$)

$$\mathbb{E}_\mu\left[\frac{1}{K}\sum_{k=1}^K \mathbb{E}_P\left[D_{\text{KL}}\left(\mathbb{P}(\cdot \mid P, w_k) \parallel \mathbb{P}_\theta(\cdot \mid P, w_k)\right)\right]\right]$$

$$= \mathcal{O}\left\{\sqrt{\frac{\log\frac{1}{\delta}}{K(N-N')T}} + \sqrt{\frac{D_{\text{KL}}(\mu \parallel \nu_J) + \log\frac{1}{\delta}}{K(N-N')T}} - \epsilon_{\text{opt}}\right\} \tag{41}$$

Finally, with inequality 41 and 39, we get data-dependent and optimization algorithm-dependent PAC-Bayesian generalization error bound of the first-level expected loss.

$$
\mathbb{E}_\mu\left[\frac{1}{K}\sum_{k=1}^K \mathbb{E}_P\left[D_{\mathrm{KL}}\big(\mathbb{P}(\cdot\mid P, w_k)\parallel \mathbb{P}_\theta(\cdot\mid P, w_k)\big)\right]\right]
$$

$$
=\mathcal{O}\left\{\sqrt{\frac{\log 1/\delta}{K(N-N')T}} + \sqrt{\frac{1}{K(N-N')T}\left[\left(1+\sqrt{\frac{\log 1/\delta}{K^2}}\right)^2 \frac{4L^2 C(\frac{1}{N_{\mathrm{param}}}, T')}{N'} + \log\frac{1}{\delta}\right]} - \epsilon_{\mathrm{opt}}\right\}
$$

$$
=\mathcal{O}\left\{\sqrt{\frac{\log 1/\delta}{K(N-N')T}} + \sqrt{\frac{1}{K(N-N')T}\left(\frac{L^2 C(\frac{1}{N_{\mathrm{param}}}, T')}{N'} + \log\frac{1}{\delta}\right)} - \epsilon_{\mathrm{opt}}\right\} \tag{42}
$$

where $C(\frac{1}{N_{\mathrm{param}}}, T') = \frac{\beta}{2}e^{8\beta S}\left(1 - e^{-\frac{T'}{\exp(8\beta S)}}\right)$. $\qquad\square$

### G.3 GENERALIZATION OF SEQUENCES AND TOPICS: TWO-LEVEL EXPECTATION

#### G.3.1 PROOF OF THEOREM F.5

**Theorem** (Data-Dependent and Optimization-Dependent Generalization Bound of the Two-Level Expected Loss). *Let the auto-regressive LLM $\mathbb{P}_\theta$ be the empirical solution of Equation 1, and $\mathbb{P}(\cdot\mid w)$ is the true data distribution under topic $w$. Under Assumptions 4.1, 4.2 and 4.5, for any $0 < \delta < 1$, with probability at least $1 - \delta$, the two-level expected loss (population loss) with infinite topics and infinite sequences per topic, denoted by $L(\theta)$ (see in Equation 5), satisfies,*

$$
\mathbb{E}_\mu[L(\theta)] = \mathcal{O}\left\{\sqrt{\frac{1}{KT_p}\left(D_{\mathrm{KL}}(\mu\parallel\nu) + \log\frac{1}{\delta}\right)} + U(\mathcal{W}_{pre}, K, N, N', T)\right\},
$$

*where $U(\mathcal{W}_{pre}, K, N, N', T)$ denotes the right hand of equality 6 or equality 38. $\mu$ and $\nu$ are the posterior and prior distribution of model parameters $\theta$, respectively. $K$, $N(N')$ and $T$ denote the number of topics, the number of sequences per topic and the sequence length utilized in the optimization process of Equation 1.*

*Proof.* In this part, since we have gotten the generalization error bound when considering infinite sequences in Theorem F.1 and Theorem F.3. Our analysis is based on that there will be a sufficient number of sequences for each topic to enable thorough learning so that in the ideal case, the well-pretrained model can perform excellently on the seen topics. We try to get the upper bound of the two-level expected loss (population loss) so that the pre-trained model can also perform well on the unseen topics under the assumption of topic distribution.

For an ICL prompt prompt, we also establish auto-regressive loss based on the prefix sequence $\mathrm{prompt}_t$. Then according to Theorem 4.6, for topic $w$, we first have

$$
\mathbb{E}_\mu\left[\frac{1}{KT_p}\sum_{k=1}^K\sum_{t=1}^{T_p}\mathbb{E}_{\mathrm{prompt}_t}\left[D_{\mathrm{KL}}\big(\mathbb{P}(\cdot\mid \mathrm{prompt}_t, w_k)\parallel \mathbb{P}_\theta(\cdot\mid \mathrm{prompt}_t, w_k)\big)\right]\right]
$$

$$
= \mathcal{O}\left\{\sqrt{\frac{\log 1/\delta}{K(N-N')T}} + \sqrt{\frac{1}{K(N-N')T}\left(\frac{L^2 C(\frac{1}{N_{\mathrm{param}}}, T')}{N'} + \log\frac{1}{\delta}\right)} - \epsilon_{\mathrm{opt}}\right\}
$$

$$
= \mathcal{O}\left\{U(\mathcal{W}_{\mathrm{pre}}, K, N, N', T)\right\} \tag{43}
$$

Using Proposition G.12 and Assumption 4.1 of $\log \frac{\mathbb{P}(\cdot | E_t^{k,n}, w_k)}{\mathbb{P}_\theta(\cdot | E_t^{k,n}, w_k)}$ is upper bounded by $C$, thus with probability at least $1 - \delta$, we consider the generalization of topic so that ICL emerges,

$$
\mathbb{E}_\mu \left[ \frac{1}{T_p} \sum_{t=1}^{T_p} \mathbb{E}_w \mathbb{E}_{\text{prompt}_t} \left[ D_{\text{KL}} \big( \mathbb{P}(\cdot \mid \text{prompt}_t, w) \parallel \mathbb{P}_\theta(\cdot \mid \text{prompt}_t, w) \big) \right] \right.
$$

$$
\left. - \frac{1}{KT_p} \sum_{k=1}^{K} \sum_{t=1}^{T_p} \mathbb{E}_{\text{prompt}_t} \big( D_{\text{KL}} \big( \mathbb{P}(\cdot \mid \text{prompt}_t, w_k) \parallel \mathbb{P}_\theta(\cdot \mid \text{prompt}_t, w_k) \big) \big) \right]
$$

$$
\leq \sqrt{\frac{C^2 \cdot \tau_{\min}}{2KT_p \log 2} \left( D_{\text{KL}}(\mu \parallel \nu) + \log \frac{2}{\delta} \right)} = \mathcal{O} \left\{ \sqrt{\frac{1}{KT_p} \left[ D_{\text{KL}}(\mu \parallel \nu) + \log \frac{1}{\delta} \right]} \right\} \quad (44)
$$

Finally, we measure the generalization error of an auto-regressive pre-trained LLM, after which the ability of ICL will emerge with good generalization. It can be denoted as $\text{gen}_{\text{topic}} = L(\theta) - L(\theta, \mathcal{W}_{\text{pre}})$, then the two-level expected loss (population loss) $L(\theta)$ can be bounded by

$$
\mathbb{E}_\mu \left[ \frac{1}{T_p} \sum_{t=1}^{T_p} \mathbb{E}_w \mathbb{E}_{\text{prompt}_t} \left[ D_{\text{KL}} \big( \mathbb{P}(\cdot \mid \text{prompt}_t, w) \parallel \mathbb{P}_\theta(\cdot \mid \text{prompt}_t, w) \big) \right] \right]
$$

$$
= \mathcal{O} \left\{ \sqrt{\frac{1}{KT_p} \left[ D_{\text{KL}}(\mu \parallel \nu) + \log \frac{1}{\delta} \right]} + U(\mathcal{W}_{\text{pre}}, K, N, N', T) \right\} \quad (45)
$$

where $U(\mathcal{W}_{\text{pre}}, K, N, N', T)$ is the right hand of inequality 6.

$\square$

### G.3.2 PROOF OF THEOREM F.7

**Theorem** (Data-Dependent, Topic-Dependent and Optimization-Dependent Generalization Error Bound of the Two-Level Expected Loss.). *Under the conditions maintained in Theorem F.5 and Assumption 4.5, when further considering topic-dependent prior, for any $0 < \delta < 1$, with probability at least $1 - \delta$, the two-level expected loss (population loss) with infinite topics and infinite sequences per topic, denoted by $L(\theta)$ (see in Equation 5), satisfies,*

$$
\mathbb{E}_\mu \left[ L(\theta) \right] = \mathcal{O} \left\{ \sqrt{\frac{1}{(K - K')T_p} \left( D_{\text{KL}}(\mu \parallel \nu_J) + \log \frac{1}{\delta} \right)} + U(\mathcal{W}_{pre}, K, N, N', T) \right\},
$$

*then detailing the term $D_{\text{KL}}(\mu \parallel \nu_J)$, $L(\theta)$ further satisfies,*

$$
\mathcal{O} \left\{ \sqrt{\frac{1}{(K - K')T} \left( \frac{\sigma^2 C(\frac{1}{N_{param}}, T')}{K'} + \log \frac{1}{\delta} \right)} + U(\mathcal{W}_{pre}, K, N, N', T) \right\},
$$

*where $C(\frac{1}{N_{param}}, T') = \frac{\beta}{2} e^{8\beta S} \left( 1 - e^{-\frac{T'}{\exp(8\beta S)}} \right)$, $U(\mathcal{W}_{pre}, K, N, N', T)$ denotes the right hand of equality 6 or equality 17. $\mu$ and $\nu_J$ are the posterior and topic-dependent prior distribution of model parameters $\theta$, respectively. $K(K')$, $N(N')$ and $T$ denote the number of topics, the number of sequences per topic and the sequence length utilized in the optimization process of Equation 1. $T'$ denotes the total training iterations. $N_{param}$ denotes the number of model parameters.*

*Proof.* In this Theorem, we try to give a detail analysis of $D_{\text{KL}}(\mu \parallel \nu)$ to get data-dependent, topic-dependent and optimization-dependent generalization bound. Similarly, we analyze the training dynamic of transformer via Gradient Langevin Dynamics (GLD)

$$
\theta_t \leftarrow \theta_{t-1} - \eta_t \nabla L(\theta_{t-1}, \mathcal{W}_{\text{pre}, I}) + \sigma_t \mathcal{N}(0, I_d).
$$

when the step size approaches zero,

$$
d\theta_t = -\nabla L(\theta_{t-1}, \mathcal{W}_{\text{pre}, I}) dt + \sqrt{\beta^{-1}} \, dB_t, \quad \theta_0 \sim \mu_0
$$

where $\nabla L(\theta, \mathcal{W}_{\text{pre},I}) = \frac{1}{(K-K')T} \sum_{k=1}^{K-K'} \sum_{t=1}^{T} \mathbb{E}_{E_t^{k,n}} [\log \frac{\mathbb{P}(x_{t+1}^{k,n}|E_t^{k,n}, w_k)}{\mathbb{P}_\theta(x_{t+1}^{k,n}|E_t^{k,n}, w_k)}]$, and $B_t$ is the standard brown emotion.

Split pre-training topics into two parts $\mathcal{W}_{\text{pre},I}$ and $\mathcal{W}_{\text{pre},J}$ (where $J$ is a random sequence including $K'$ indexes uniformly sampled from $[K]$ without replacement and $I$ is $[K] \setminus J$). Then the total sequences are divided into $E^I = \{E^k\}_{k \in \mathcal{W}_{\text{pre},I}}$ and $E^J = \{E^k\}_{k \in \mathcal{W}_{\text{pre},J}}$. Assume that the prior distribution of model parameters $\theta$ is depending on the topic subset $E^J$, which is denoted by $\nu_J$ and the posterior distribution of $\theta$ is depending on $E^I$ denoted by $\mu$.

Let $\widetilde{\Theta} = (\theta_t)_{t \geq 0}$ and $\widetilde{\Theta}' = (\theta_t')_{t \geq 0}$ be the trajectory trained on $\mathcal{W}_{\text{pre},I}$ and $\mathcal{W}_{\text{pre},J}$ (the total sequences are divided into $E^I = \{E^k\}_{k \in \mathcal{W}_{\text{pre},I}}$ and $E^J = \{E^k\}_{k \in \mathcal{W}_{\text{pre},J}}$). Let $\mu$ and $\nu_J$ be the distribution of $\widetilde{\Theta}$ and $\widetilde{\Theta}'$ respectively, $p_t$ and $q_t$ be the pdf of $\widetilde{\Theta}$ and $\widetilde{\Theta}'$ and the total steps of iteration is $T'$. $D_{\text{KL}}(\rho \parallel \pi_J)$ is equal to $D_{\text{KL}}(p_{T'} \parallel q_{T'})$. To bound $D_{\text{KL}}(p_{T'} \parallel q_{T'})$, we first apply Leibniz's rule and the chain rule on it:

$$\frac{\mathrm{d}}{\mathrm{d}t} D_{KL}(p_t \| q_t) = \frac{\mathrm{d}}{\mathrm{d}t} \int_{\mathbb{R}^d} p_t \log \frac{p_t}{q_t} \mathrm{d}\theta$$

$$= \int_{\mathbb{R}^d} \left( \frac{\mathrm{d}p_t}{\mathrm{d}t} \log \frac{p_t}{q_t} + p_t \cdot \frac{q_t}{p_t} \cdot \frac{\frac{\mathrm{d}p_t}{\mathrm{d}t} q_t - p_t \frac{\mathrm{d}q_t}{\mathrm{d}t}}{q_t^2} \right) \mathrm{d}\theta$$

$$= \int_{\mathbb{R}^d} \frac{\mathrm{d}p_t}{\mathrm{d}t} \log \frac{p_t}{q_t} \mathrm{d}\theta - \int_{\mathbb{R}^d} \frac{p_t}{q_t} \frac{\mathrm{d}q_t}{p_t} \mathrm{d}\theta + \int_{\mathbb{R}^d} \frac{\mathrm{d}p_t}{\mathrm{d}t} \mathrm{d}\theta$$

$$= \underbrace{\int_{\mathbb{R}^d} \frac{\mathrm{d}p_t}{\mathrm{d}t} \log \frac{p_t}{q_t} \mathrm{d}\theta}_{(A)} - \underbrace{\int_{\mathbb{R}^d} \frac{p_t}{q_t} \frac{\mathrm{d}q_t}{p_t} \mathrm{d}\theta}_{(B)},$$

where the last equality follows from that $\int \frac{\mathrm{d}p_t}{\mathrm{d}t} \mathrm{d}\theta = \frac{\mathrm{d}}{\mathrm{d}t} \int p_t \mathrm{d}\theta = 0$, since $p_t$ is a probability measure. By Fokker-Planck Equation for $p_t$, $\frac{\partial p_t}{\partial t} = \frac{1}{\beta} \Delta p_t + \nabla \cdot (p_t \nabla L(\theta, \mathcal{W}_{\text{pre},I}))$.

Then we bound term $A$,

$$A := \int_{\mathbb{R}^d} \left( \frac{\mathrm{d}p_t}{\mathrm{d}t} \log \frac{p_t}{q_t} \right) \mathrm{d}\theta$$

$$= \int_{\mathbb{R}^d} \left( \frac{1}{\beta} \Delta p_t + \nabla \cdot (p_t \nabla L(\theta, \mathcal{W}_{\text{pre},I})) \right) \log \frac{p_t}{q_t} \mathrm{d}\theta$$

$$= \frac{1}{\beta} \left[ \int_{\mathbb{R}^d} \Delta p_t \log \frac{p_t}{q_t} \mathrm{d}\theta \right] + \int_{\mathbb{R}^d} \nabla \cdot (p_t \nabla L(\theta, \mathcal{W}_{\text{pre},I})) \log \frac{p_t}{q_t} \mathrm{d}\theta$$

$$= \frac{1}{\beta} \left[ \nabla p_t \log \frac{p_t}{q_t} - \int_{\mathbb{R}^d} \langle \nabla \log \frac{p_t}{q_t}, \nabla p_t \rangle \mathrm{d}\theta \right]$$

$$+ \left[ p_t \nabla L(\theta, \mathcal{W}_{\text{pre},I}) \log \frac{p_t}{q_t} - \int_{\mathbb{R}^d} \langle \nabla \log \frac{p_t}{q_t}, p_t \nabla L(\theta, \mathcal{W}_{\text{pre},I}) \rangle \mathrm{d}\theta \right]$$

$$= \frac{-1}{\beta} \int_{\mathbb{R}^d} \langle \nabla \log \frac{p_t}{q_t}, \nabla p_t \rangle \mathrm{d}\theta - \int_{\mathbb{R}^d} \langle \nabla \log \frac{p_t}{q_t}, p_t \nabla L(\theta, \mathcal{W}_{\text{pre},I}) \rangle \mathrm{d}\theta$$

Bound term $B$,

$$B := \int_{\mathbb{R}^d} \left( \frac{p_t}{q_t} \frac{\mathrm{d}q_t}{\mathrm{d}t} \right) \mathrm{d}w$$

$$= \int_{\mathbb{R}^d} \frac{p_t}{q_t} \left( \frac{1}{\beta} \Delta q_t + \nabla \cdot (q_t \nabla L(\theta, \mathcal{W}_{\text{pre},J})) \right) \mathrm{d}w$$

$$= \frac{1}{\beta} \left[ \int_{\mathbb{R}^d} \frac{p_t}{q_t} \Delta q_t \mathrm{d}w \right] + \int_{\mathbb{R}^d} \frac{p_t}{q_t} \nabla \cdot (q_t \nabla L(\theta, \mathcal{W}_{\text{pre},J})) \mathrm{d}w$$

$$= \frac{1}{\beta} \left[ \frac{p_t}{q_t} \nabla q_t - \int_{\mathbb{R}^d} \langle \nabla \frac{p_t}{q_t}, \nabla q_t \rangle \mathrm{d}w \right] + \left[ \frac{p_t}{q_t} q_t \nabla L(\theta, \mathcal{W}_{\text{pre},J}) - \int_{\mathbb{R}^d} \langle \nabla \frac{p_t}{q_t}, q_t \nabla L(\theta, \mathcal{W}_{\text{pre},J}) \rangle \mathrm{d}w \right]$$

$$= \frac{-1}{\beta} \int_{\mathbb{R}^d} \langle \nabla \frac{p_t}{q_t}, \nabla q_t \rangle \mathrm{d}w - \int_{\mathbb{R}^d} \langle \nabla \frac{p_t}{q_t}, q_t \nabla L(\theta, \mathcal{W}_{\text{pre},J}) \rangle \mathrm{d}w$$

In summary, the deviation of $D_{KL}(p_t||q_t)$ can be bounded,

$$
\begin{aligned}
&\frac{\mathrm{d}}{\mathrm{d}t}D_{KL}(p_t \parallel q_t) \\
=&\frac{-1}{\beta}\int_{\mathbb{R}^d}\langle\nabla\log\frac{p_t}{q_t},\nabla p_t\rangle\mathrm{d}w - \int_{\mathbb{R}^d}\langle\nabla\log\frac{p_t}{q_t},p_t\nabla L(\theta,\mathcal{W}_{\mathrm{pre},I})\rangle\mathrm{d}w \\
&+\frac{1}{\beta}\int_{\mathbb{R}^d}\langle\nabla\frac{p_t}{q_t},\nabla q_t\rangle\mathrm{d}w + \int_{\mathbb{R}^d}\langle\nabla\frac{p_t}{q_t},q_t\nabla L(\theta,\mathcal{W}_{\mathrm{pre},J})\rangle\mathrm{d}w \\
=&\frac{-1}{\beta}\int_{\mathbb{R}^d}\left(\langle\nabla\log\frac{p_t}{q_t},\nabla p_t\rangle - \langle\nabla\frac{p_t}{q_t},\nabla q_t\rangle\right)\mathrm{d}w \\
&-\int_{\mathbb{R}^d}\left(\langle\nabla\log\frac{p_t}{q_t},p_t\nabla L_{E_I}(\theta,\mathcal{W}_{\mathrm{pre}})\rangle - \langle\nabla\frac{p_t}{q_t},q_t\nabla L(\theta,\mathcal{W}_{\mathrm{pre},J})\rangle\right)\mathrm{d}w \\
=&\frac{-1}{\beta}\int_{\mathbb{R}^d}\left(\langle\frac{\nabla p_t}{p_t}-\frac{\nabla q_t}{q_t},\nabla p_t\rangle - \langle\frac{\nabla p_t}{q_t}-\frac{p_t\nabla q_t}{q_t^2},\nabla q_t\rangle\right)\mathrm{d}w \\
&-\int_{\mathbb{R}^d}\left(\langle\nabla\log\frac{p_t}{q_t},p_t\nabla L(\theta,\mathcal{W}_{\mathrm{pre},I})\rangle - \frac{p_t}{q_t}\langle\nabla\log\frac{p_t}{q_t},q_t\nabla L(\theta,\mathcal{W}_{\mathrm{pre},J})\rangle\right)\mathrm{d}w \\
=&\frac{-1}{\beta}\int_{\mathbb{R}^d}p_t\big\|\nabla\log\frac{p_t}{q_t}\big\|_2^2\mathrm{d}w + \int_{\mathbb{R}^d}p_t\langle\nabla\log\frac{p_t}{q_t},\nabla L(\theta,\mathcal{W}_{\mathrm{pre},I})-\nabla L(\theta,\mathcal{W}_{\mathrm{pre},J})\rangle\mathrm{d}w
\end{aligned}
$$

Since for any constant $c \neq 0$, vector $\alpha$ and $\beta$, we have the inequality $\langle\frac{\alpha}{\sqrt{c}},\frac{\beta}{\sqrt{c}}\rangle \leq \frac{\|\alpha\|^2}{2c} + \frac{c\|\beta\|^2}{2}$, then we can transform the last equality into

$$
\frac{\mathrm{d}}{\mathrm{d}t}D_{KL}(p_t \parallel q_t) \leq \frac{-1}{2\beta}\int_{\mathbb{R}^d}p_t\big\|\nabla\log\frac{p_t}{q_t}\big\|_2^2\mathrm{d}w + \frac{\beta}{2}\int_{\mathbb{R}^d}p_t\big\|\nabla L(\theta,\mathcal{W}_{\mathrm{pre},I})-\nabla L(\theta,\mathcal{W}_{\mathrm{pre},J})\big\|_2^2\mathrm{d}w
$$

According to Lemma G.10, we have $\int_{\mathbb{R}^d}p_t\left\|\nabla\log\frac{p_t}{q_t}\right\|_2^2\mathrm{d}w \geq \frac{2\beta}{\exp(8\beta S)}D_{\mathrm{KL}}(p_t||q_t)$, then transform the first term in the right hand of the above inequality,

$$
\frac{\mathrm{d}}{\mathrm{d}t}D_{KL}(p_t||q_t) \leq -\frac{1}{\exp(8\beta S)}D_{\mathrm{KL}}(p_t||q_t) + \frac{\beta}{2}\mathbb{E}_{\theta_t}\left[\big\|\nabla L(\theta,\mathcal{W}_{\mathrm{pre},I})-\nabla L(\theta,\mathcal{W}_{\mathrm{pre},J})\big\|_2^2\right]
$$

Let $\phi(t) = D_{KL}(p_t||q_t)$, $\delta(t) = \frac{\beta}{2}\mathbb{E}_{\theta_t}\left[\big\|\nabla L(\theta,\mathcal{W}_{\mathrm{pre},I})-\nabla L(\theta,\mathcal{W}_{\mathrm{pre},J})\big\|_2^2\right]$, $\alpha = \frac{1}{\exp(8\beta S)}$, then we get the following difference equation:

$$
\phi'(t) = -\alpha\phi(t) + \delta(t), \ \phi(0) = 0
$$

Solve the equation:

$$
D_{KL}(p_{T'} \parallel q_{T'}) \leq \frac{\beta}{2}\int_0^{T'}e^\alpha(t-T')\mathbb{E}_{\theta_t}\left[\big\|\nabla L(\theta,\mathcal{W}_{\mathrm{pre},I})-\nabla L(\theta,\mathcal{W}_{\mathrm{pre},J})\big\|_2^2\right]\mathrm{d}t, \ \alpha = \frac{1}{\exp(8\beta S)}.
$$

We first define that

$$
G(J) = \sqrt{\mathbb{E}_{\theta_t}\left[\int_0^{T'}e^\alpha(t-T')\left[\big\|\nabla L(\theta,\mathcal{W}_{\mathrm{pre},I})-\nabla L(\theta,\mathcal{W}_{\mathrm{pre},J})\big\|_2^2\right]\mathrm{d}t\right]}
$$

Let $J$ and $J'$ be two neighboring collections, we first prove that $G(J) - G(J')$ is small. Let $X_t = \nabla L(\theta, \mathcal{W}_{\text{pre},I}) - \nabla L(\theta, \mathcal{W}_{\text{pre},J})$, $Y_t = \nabla L(\theta, \mathcal{W}_{\text{pre},J}) - \nabla L(\theta, \mathcal{W}_{\text{pre},J'})$. Then,

$$
\begin{aligned}
G(J')^2 =& \mathbb{E}_{\theta_t}\left[\int_0^{T'} e^{\alpha(t-T')}\|X_t + Y_t\|_2^2 \, dt\right] \\
=& \mathbb{E}_{\theta_t}\left[\int_0^{T'} e^{\alpha(t-T')}\left(X_t^\top X_t + Y_t^\top Y_t\right) dt\right] + 2\mathbb{E}_{\theta_t}\left[\int_0^{T'} e^{\alpha(t-T')} X_t^\top Y_t dt\right] \\
\leq& \mathbb{E}_{\theta_t}\left[\int_0^{T'} e^{\alpha(t-T')}\left(\|X_t\|_2^2 + \|Y_t\|_2^2\right) dt\right] \\
&+ 2\sqrt{\mathbb{E}_{\theta_t}\left[\int_0^{T'} e^{\alpha(t-T')}\|X_t\|_2^2 dt\right]}\sqrt{\mathbb{E}_{\theta_t}\left[\int_0^{T'} e^{\alpha(t-T')}\|Y_t\|_2^2 dt\right]} \\
=& \left(\sqrt{\mathbb{E}_{\theta_t}\left[\int_0^{T'} e^{\alpha(t-T')}\|X_t\|_2^2 dt\right]} + \sqrt{\mathbb{E}_{\theta_t}\left[\int_0^{T'} e^{\alpha(t-T')}\|Y_t\|_2^2 dt\right]}\right)^2 \\
=& \left(G(J) + \sqrt{\mathbb{E}_{\theta_t}\left[\int_0^{T'} e^{\alpha(t-T')}\|Y_t\|_2^2 dt\right]}\right)^2
\end{aligned}
$$

For any fixed $J$ and $\theta_t$, using the Assumption 4.5 that $\|\nabla L(\theta_t, w_k)\| \leq \sigma$, then

$$
\int_0^{T'} e^{\alpha(t-T')}\|Y_t\|_2^2 \, dt \leq \int_0^{T'} e^{\alpha(t-T')}\frac{4\sigma^2}{K'^2} dt = \frac{4\sigma^2(1-e^{-\alpha T'})}{K'^2\alpha}
$$

Then,

$$
\left|G(J) - G(J')\right| \leq \frac{2\sigma}{K'}\sqrt{\frac{1-e^{-\alpha T'}}{\alpha}}
$$

Applying lemma of concentration inequality and there are $K'$ indexes in $J$ or $J'$,

$$
P_J\left[G(J) - \mathbb{E}_J[G(J)] \geq \epsilon\right] \leq \exp\left(\frac{-2\epsilon^2}{K'\frac{4\sigma^2(1-e^{-\alpha T'})}{K'^2\alpha}}\right) = \exp\left(\frac{-K'\alpha\epsilon^2}{2\sigma^2(1-e^{-\alpha T'})}\right)
$$

We also have,

$$
P_J\left[G(J)^2 \geq (\mathbb{E}_J[G(J)] + \epsilon)^2\right] \leq \exp\left(\frac{-K'\alpha\epsilon^2}{2\sigma^2(1-e^{-\alpha T'})}\right)
$$

then

$$
\begin{aligned}
\mathbb{E}_J[G(J)] = \mathbb{E}_J&\sqrt{\mathbb{E}_{\theta_t}\left[\int_0^{T'} e^{\alpha}(t-T')\left[\|\nabla L(\theta_t, \mathcal{W}_{\text{pre},I}) - \nabla L(\theta_t, \mathcal{W}_{\text{pre},J})\|_2^2\right] dt\right]} \\
\leq& \sqrt{\mathbb{E}_{\theta_t}\left[\int_0^{T'} e^{\alpha}(t-T')\mathbb{E}_J\left[\|\nabla L(\theta_t, \mathcal{W}_{\text{pre},I}) - \nabla L(\theta_t, \mathcal{W}_{\text{pre},J})\|_2^2\right] dt\right]} \\
\leq& \sqrt{\int_0^{T'} e^{\alpha(t-T')}\frac{4\sigma^2}{K'}} \\
=& \frac{2\sigma}{\sqrt{K'}}\sqrt{\frac{1-e^{-\alpha T'}}{\alpha}}
\end{aligned}
$$

Let $\exp\left(\frac{-K'\alpha\epsilon^2}{2\sigma^2(1-e^{-\alpha T'})}\right) = \delta$, then $\epsilon = \sqrt{\frac{4\sigma^2(1-e^{-\alpha T'})\log\frac{1}{\delta}}{2K'\alpha}}$. It follows that with probability at least $1-\delta$

$$
G(J)^2 \leq \left(\frac{2\sigma}{\sqrt{K'}}\sqrt{\frac{1-e^{-\alpha T'}}{\alpha}} + \sqrt{\frac{4\sigma^2(1-e^{-\alpha T'})\log\frac{1}{\delta}}{2K'\alpha}}\right)^2
$$

Then,

$$\mathbb{E}_{\theta_t} \left[ \int_0^{T'} e^\alpha (t - T') \left[ \|\nabla L(\theta_t, \mathcal{W}_{\text{pre},I}) - \nabla L(\theta_t, \mathcal{W}_{\text{pre},J})\|_2^2 \right] \mathrm{d}t \right]$$

$$\leq \left( \frac{2\sigma}{\sqrt{K'}} \sqrt{\frac{1 - e^{-\alpha T'}}{\alpha}} + \sqrt{\frac{4\sigma^2(1 - e^{-\alpha T'}) \log \frac{1}{\delta}}{2K'\alpha}} \right)^2$$

$$= \frac{4\sigma^2}{K'} \left( 1 + \sqrt{\log \frac{1}{\delta}} \right)^2 \frac{(1 - e^{-\alpha T'})}{\alpha}$$

$$= \frac{4\sigma^2}{K'} \left( 1 + \sqrt{\log \frac{1}{\delta}} \right)^2 e^{8\beta S} \left( 1 - \exp \left( -\frac{T'}{e^{8\beta S}} \right) \right)$$

We bound the KL-divergence.

$$D_{\text{KL}}\left( p_{T'} \| q_{T'} \right) \leq \left( 1 + \sqrt{\log \frac{1}{\delta}} \right)^2 \frac{2\sigma^2 \beta e^{8\beta S}(1 - \exp(-\frac{T'}{e^{8\beta S}}))}{K'} = \left( 1 + \sqrt{\log \frac{1}{\delta}} \right)^2 \frac{4\sigma^2 C(\frac{1}{N_{\text{param}}}, T')}{K'}$$

$$\tag{46}$$

where $C(\frac{1}{N_{\text{param}}}, T') = \frac{\beta}{2} e^{8\beta S} \left( 1 - e^{-\frac{T'}{\exp(8\beta S)}} \right)$.

As introduced before, the prior distribution of model parameters $\theta$ is depending on the subset $E^J$, which is denoted by $\nu_J$ and the posterior distribution of $\theta$ is depending on $E^I$ denoted by $\mu$. Then Theorem F.5 can be slightly changed.

$$\mathbb{E}_\mu \left[ \frac{1}{T_p} \sum_{t=1}^{T_p} \mathbb{E}_w \mathbb{E}_{\text{prompt}_t} \left[ D_{\text{KL}}\left( \mathbb{P}(\cdot \mid \text{prompt}_t, w) \| \mathbb{P}_\theta(\cdot \mid \text{prompt}_t, w) \right) \right] \right]$$

$$\leq \sqrt{\frac{C^2 \cdot \tau_{\min}}{2(K - K')T_p \log 2}} \left( D_{\text{KL}}(\mu \| \nu) + \log \frac{2}{\delta} \right) \tag{47}$$

$$+ \mathbb{E}_\mu \left[ \frac{1}{(K - K')T_p} \sum_{k=1}^{K-K'} \sum_{t=1}^{T_p} \mathbb{E}_{\text{prompt}_t} \left[ D_{\text{KL}}\left( \mathbb{P}(\cdot \mid \text{prompt}_t, w_k) \| \mathbb{P}_\theta(\cdot \mid \text{prompt}_t, w_k) \right) \right] \right]$$

$$= \mathcal{O} \left\{ \sqrt{\frac{1}{(K - K')T_p}} \left( D_{\text{KL}}(\mu \| \nu) + \log \frac{1}{\delta} \right) + U(\mathcal{W}_{\text{pre}}, K, N, N', T) \right\} \tag{48}$$

Finally, with inequality 46 and 48, we get data-dependent, topic-dependent and optimization-dependent PAC-Bayesian generalization bound of the two-level expected loss, *i.e.*, $L(\theta)$ is bounded by

$$\mathbb{E}_\mu \left[ \frac{1}{T_p} \sum_{t=1}^{T_p} \mathbb{E}_w \mathbb{E}_{\text{prompt}_t} \left[ D_{\text{KL}}\left( \mathbb{P}(\cdot \mid \text{prompt}_t, w_k) \| \mathbb{P}_\theta(\cdot \mid \text{prompt}_t, w_k) \right) \right] \right]$$

$$= \mathcal{O} \left\{ \sqrt{\frac{1}{(K - K')T_p}} \left[ \left( 1 + \sqrt{\log \frac{1}{\delta}} \right)^2 \frac{4\sigma^2 C(\frac{1}{N_{\text{param}}}, T')}{K'} + \log \frac{1}{\delta} \right] + U(\mathcal{W}_{\text{pre}}, K, N, N', T) \right\}$$

$$= \mathcal{O} \left\{ \sqrt{\frac{1}{(K - K')T_p}} \left( \frac{\sigma^2 C(\frac{1}{N_{\text{param}}}, T')}{K'} + \log \frac{1}{\delta} \right) + U(\mathcal{W}_{\text{pre}}, K, N, N', T) \right\} \tag{49}$$

where $C(\frac{1}{N_{\text{param}}}, T') = \frac{\beta}{2} e^{8\beta S} \left( 1 - e^{-\frac{T'}{\exp(8\beta S)}} \right)$. $\qquad\square$

