# OpenReview forum: "Towards Auto-Regressive Next-Token Prediction: In-context Learning Emerges from Generalization"
_ICLR.cc/2025/Conference — ICLR 2025 Poster_

### Official Review · Reviewer_JqeP · 2024-10-21

**Soundness:** 2
**Presentation:** 2
**Contribution:** 3
**Rating:** 5
**Confidence:** 2

**Summary:**

The paper proposes theoretical analysis to explain the ICL abilities exhibited by LLMs. The analysis establishes data-dependent, topic-dependent and optimization dependent generalization bounds for pre-trained LLMs, investigating that ICL merges from the generalization of sequence topics. The paper also present empirical evidence to support the theoretical implications.

**Strengths:**

1. The breakdown of pre-training loss into data and topic level optimization is interesting and inspiring. It is capable of identifying two sources of pre-training loss, sequences and topics.

2. The theoretical bounds and empirical evidence are aligned: with longer sequence length, more number of sequence and more number of topics, the ICL loss is smaller and performance is better.

**Weaknesses:**

1. One concern centers around the controlled experiment of the empirical investigation. Given the data is sampled from the topic and token distribution, with more topic or more sequence, or longer sequence, it implies more training data, so it is expected that the ICL loss drops and that performance increases. It is unclear that the loss change can be attributed to the three sources or simply data size, under the current experiment setup.

2. Although the theoretical analysis establish the bounds of ICL loss, but the results apply to general prompt, rather than in-context examples only. It also implies that the ICL loss should be lower and performance should be better with more training data (in terms of topic and sequence). But only well-structured in-context samples work empirically. I am uncertain if the presented theory is sufficiently tailored to address the problem of ICL.

3. Now the empirical investigation is omitted from the main paper but experiment setup discussion remains. I suggest to put the main analysis of empirical result into main paper and defer some theoretical analysis into appendix.

**Questions:**

1. Did you control the training size for various topic size, sequence length and sequence amount?

---

> ### Author Response · Authors · 2024-11-19
>
> Thank you sincerely for your suggestions and questions!
>
> Regarding your concerns about the impacts of $K,N,T$, we emphasize that we have already ***conducted ablation studies*** on linear dynamic systems and GINC dataset, to evaluate the individual impacts of number of training topics, sequences and sequence length. To further address your concern, during this discussion peoriod, ***we conduct more experiments on real-world language datasets*** to further reveal the impacts of $K,N$ beyond $T$. Meanwhile, with more insightful experiments, we observe the optimization process and the potential effects of prior model initialization (guided by our therorems). Additionally, as you suggested, we have moved some key empirical analysis into the main text and deferred some description in Section 3 into the Appendix B!
>
> **The revised paper has been updated and we strongly encourage to read the "Experiments" section for more details!** Below, we do our best to address your questions adequately!
>
> >**Q1, Q4: One concern centers around the controlled experiment of the empirical investigation [...] It is unclear that the loss change can be attributed to the three sources or simply data size [...] Did you control the training size for various topic size, sequence length and sequence amount?**
>
> **A:**  Thanks for your question! We strongly agree with you that it's necessary to conduct ablation studies on $K,N,T$, to separately investigate the impact of the number of pre-training topics, sequences and sequence length on the performance.
>
> In fact, we indeed have already conducted such experiments with controlling topic size, sequence amount and sequence length, on linear dynamic systems and GINC synthetic datatset. The details are introduced in Appendix C in the earlier version of paper. In the revised paper, as you suggested, we have moved some key empirical analysis into the main text and deferred some description in Section 3 into the Appendix B! Additionally, in this discussion period, we have producted **more experiments** on real-world datasets to further reveal the impacts of $K,N$ beyond $T$, and more **insightful experiments** that exploring the optimization process and the effects of prior model initialization induced by theory.
>
> Here, we briefly list the key experiments designed to evaluate the individual impacts of $K,N,T$, offering support for our theoretical insights:
> - **Numerical experiments on linear dynamic systems** (Old & New version: Appendix C.1).
> - **Experiments on synthetic language dataset GINC** (Old version: Appendix C.2; New version: Section 5).
> - **Experiments on real-world language datasets** (Old version: section 5; New version: Section 5 and Appendix C.3). Note that in this discussion period, we have conducted additional ablation experiments on $K,N$ and optimization process, as a complement to the original verificaition, which was only focused on sequence length $T$.

---

> > ### Author Response · Authors · 2024-11-19
> >
> > >**Q2: Although the theoretical analysis establish the bounds of ICL loss, but the results apply to general prompt, rather than in-context examples only. It also implies that the ICL loss should be lower and performance should be better with more training data (in terms of topic and sequence). But only well-structured in-context samples work empirically. I am uncertain if the presented theory is sufficiently tailored to address the problem of ICL.**
> >
> > **A:** Thanks for your thoughtful question! We agree with you that well-structured in-context samples can signicantly enhance performance. Although our theoretical results are derived for general prompts (with concatenated in-context examples, without assuming any specific order of concatenation), we emphasize that **the effects of sample order are already implicitly captured** within our framework and results. To clarify this point, we would like to make detailed discussions mainly from the following three aspects, as well as offer some inspiring guidance for future research on ICL.
> >
> > **(1) The characterization of conditional probability.** In the setting of a more realistic AR-NTP, individual tokens are generated based on the preceding tokens. Compared to most ICL research that simply considering $(x_i, y_i)$ pairs in each sequence/prompt, we capture the dependency between tokens by ultilizing conditional probability, i.e., $\mathbb{P}(x_t \mid x_1, x_2, \cdots, x_{t-1})$, which serves as our optimization objective. When considering the expectation over sequence (generalization of sequence), we specially divide it into two expections: one is the expection over token conditioned on its prefix sequence, and the other is the expectation over prefix sequence itself. Clearly, the order of samples affects the distribution of the prefix sequence, which in turn influences the optimization process of the conditional probability. Thus, **the representation of token-dependency provides a theoretical foundation for capturing the influence of sample order.**
> >
> > **(2) The impacts of sample order on generalization can be induced from our theorems.** As introduced above, in ICL, the prediction not only depends on the current input but is also influenced by the preceding context. When the model processes the in-context examples, if the sample order causes significant fluctuations (e.g., abruptly switching from a sample highly related to the query to one with very low relevance), the model may need to continuously adjust its parameters to adapt to the new input. This could lead to **increased gradient fluctuations and training instability**. In our theory, based on Assumption 4.5, **we use $\sigma$ to describe the upper bound of the expected gradient**. If $\sigma$ is large, it indicates that the magnitude of gradient changes is uncontrolled at each update, reflecting the instability of the optimization process.
> >
> > Combining the above analysis and the optimization insights based on $\sigma$, it demonstrates that, a good sample order can help the model converge more stably, which means a smaller $\sigma$, a smaller generalization upper bound, and better generalization performance. In total, the impacts of sample order on generalization can be induced from our theorems.
> >
> > **(3) Our theory provides guiding principles for in-context examples.** Our generalization theoretical analysis essentially provides an upper bound for the population loss when considering two-level expectation over topics and sequences. While this upper bound applies to general prompts, it can be used to guide the construction of better in-context examples. For instance, **reducing irrelevant samples in the context (optimizing the sample order)** can better satisfy Assumption 4.5, thus indirectly improving generalization performance.
> >
> > **Inspiring guidance for future research on ICL.** We strongly agree with you that a more detailed consideration of sample order is of significant research value for a deeper understanding of ICL. In our initial exploration of both considering a more realistic AR-NTP paradigm and establishing a systematic generalization framework, we hope to provide new insights for this field!
> >
> > Here, we leave a more detailed analysis of sample order for future work, but based on our current framework, we offer a few potential directions for further exploration:

---

> > > ### Author Response · Authors · 2024-11-19
> > >
> > > - **Utilize information theory to theoretically analyze the contribution of sample order.** Information theory is tightly related to PAC-Bayesian framework in this paper. By using mutual information metrics, we can quantify the contribution of each sample order to the gradient updates. Specifically, define the mutual information $I(\text{order}; \nabla L)$ to measure the relationship between sample order and gradients. Introducing this metric to our generalization bound provides a good chance for a better characterization of the impact of sample order on the gradients, and further on generalization.
> > >
> > > - **Use order-dependent regularization to practically improve optimization and generalization.** We can consider introducing order-dependent regularization to control the effects of sample order on gradient updates. For example, we could design a regularization term that penalizes large gradient fluctuations caused by poor sample order, which can improve the stability of the optimization process. Additionally, we can optimize the model to minimize the impact of sample order on gradients, by maximizing the mutual information discussed above. This would result in a model that is less sensitive to sample order and has better generalization performance.
> > >
> > > >**Q3: Now the empirical investigation is omitted from the main paper but experiment setup discussion remains. I suggest to put the main analysis of empirical result into main paper and defer some theoretical analysis into appendix.**
> > >
> > > **A:** Thanks for your suggestion! **Our current paper structure was designed with the following considerations:** our main contribution lies in establishing of a systemtic pre-training and ICL framework, as well as demonstrating that ICL emerges from generalization of pre-trained LLMs under the realistic AR-NTP training paradigm. Thus, we dedicated significant portions of Section 3 and 4 to modeling the learning process and presenting theoretical results, which led to a more concise description of the experiments in the main text (more experimental details are deferred to Appendix C).
> > >
> > > Yet, we now believe that **your suggestion is well-founded**, it may be more helpful for readers to grasp our main theoretical framework and Theorems. We greatly agree with you, and in the revised paper, we have put some key empirical analysis into the main text, and defer some description in Section 3 into the Appendix B. Additionally, we have producted more experiments on real-world datasets to further observe the impacts of $K,N,T$, optimization process and prior model initialization!
> > >
> > > In total, **the revised paper has been updated and we strongly encourage to read the "Experiments" section for more details!** Once again, we sincerely thank you for the detailed review and constructive feedback! We hope that our revisions satisfactorily address your concerns and we welcome any further discussions that may be helpful to the evaluation.

---

> > > > ### Comment · Reviewer_JqeP · 2024-11-24
> > > >
> > > > I would like to modify my comments on the order of ICL samples. Now I think it would be matter that much given that ICL are sampled from the same distribution as the training data. The remaining concern would be that I think there might some issue about controlling other confounding affect in empirical results. I would increase rating accordingly the reflect my opinion change on the effect or ICL sample order.

---

> > ### Comment · Reviewer_JqeP · 2024-11-24
> >
> > My point is not about the effect of K, N, T, but controlling the train data size when you use different K/N/T. When you use a larger K, the train size is larger than the one with smaller K, all else equal. The size of training data also impact the performance which cannot be attributed to the effect of large K.

---

> ### Author Response · Authors · 2024-11-24
> **Reply to Remaining Concern!**
>
> Dear Reviewer JqeP,
>
> Thank you sincerely for your responses and support! We are greatly encouraged by your score improvement! Regarding your remaining concern, we realize that before, we may not have fully understood your point about how independently changing $K,N,T$ could affect training size. After carefully considering your question, here is our current perspective:
>
> In our theoretical pre-training and ICL framework, $K$ pre-training topics determine whether the training data adequately covers the topic space (i.e., topic diversity), $N$ training sequences dictate whether each topic has sufficient sequences for effective training, and $T$ reflects the influence of sequence length. **These three factors collectively shape the properties of the sample space and contribute to the training sample size.** We strongly agree with you that the training size indeed affects model performance. However, it is **the result of the combined effects of $K,N,T$.** We understand the experiments in our paper as follows: For example, in experiments related to $K$, when we increase $K$ while keeping $N,T$ fixed, we observe improving performance. **For this improvement**, it is certainly a contribution of the increased training size, but indeed, it is the result of richer and more diverse topics being introduced.
>
> Regarding your suggestion to keep the total training size constant, we have considered the possibility of increasing $K$ while proportionally decreasing $N$. This approach would indeed fix the total training size but would also introduce new variable relationships, shifting the focus to the *relative effects* of $K$ and $N$. For example, we could design three pre-training regimes with $K \in \\{5, 10, 20\\}$ and correspondingly $N \in \\{2^{14}, 2^{13}, 2^{12}\\}$. However, this approach might also somewhat deviate from your original intent of isolating the independent effect of $K$.
>
> Once again, thank you for your valuable suggestions and for giving us the opportunity to clarify! We sincerely hope that our responses have addressed your concerns. Please don’t hesitate to let us know if you have any additional questions or suggestions. We would be more than happy to explore further experiments based on your specific ideas and incorporate them in our formal version!

---

### Official Review · Reviewer_x2ai · 2024-10-27

**Soundness:** 3
**Presentation:** 3
**Contribution:** 3
**Rating:** 8
**Confidence:** 2

**Summary:**

The paper critiques existing ICL theoretical analyses for relying on unrealistic i.i.d. (independent and identically distributed) assumptions and lacking an explanation of ICL emergence. To address these issues, the authors propose an auto-regressive next-token prediction (AR-NTP) framework, which mirrors real-world language learning by considering token dependencies within prompts. The paper introduces a pre-training and ICL framework, analyzing generalization bounds using PAC-Bayesian techniques and exploring how ICL arises through generalization of sequences and topics. The findings are validated through experiments on synthetic and real-world datasets, concluding that ICL capabilities emerge from the strong generalization of LLMs trained on diverse sequences and topics.

**Strengths:**

This paper presents an original contribution to the understanding of in-context learning (ICL) by introducing the auto-regressive next-token prediction (AR-NTP) framework. The paper seek to addresses two major limitations in the current literature: the i.i.d. assumption in prompt tokens and the lack of an explanation for how ICL emerges from pre-trained large language models (LLMs). By shifting the focus to token dependencies in AR-NTP and offering a novel pre-training and ICL framework, the authors provide a fresh perspective on ICL emergence through generalization. The originality stems from the use of PAC-Bayesian generalization techniques, and the proposed framework opens up new avenues for exploring token dependencies and generalization beyond previous work that relied on supervised function learning. The authors also provide theoretical results that are supported by empirical experiments on both synthetic and real-world datasets, enhancing the quality and rigor of their contributions.

the paper is well-structured and flows well, guiding the reader through complex concepts such as generalization bounds, topic-dependent priors, and PAC-Bayesian techniques without overwhelming them with technical jargon. The theoretical framework is detailed yet accessible, and the results are presented in a way that is easy to follow, particularly with well-placed diagrams and examples. The significance of the work lies in its ability to explain ICL as an emergent property of pre-trained LLMs, which has strong implications for both academia and industry. The results contribute to understanding why larger models exhibit better ICL performance and provide concrete recommendations for improving model generalization.

**Weaknesses:**

While the paper makes significant strides in addressing limitations in existing ICL literature, there are several areas where the work could be strengthened. One key weakness lies in the assumptions behind the AR-NTP paradigm. Although the shift from i.i.d. settings to token-dependent sequences is crucial, the paper could have provided a more thorough discussion on the potential challenges this poses for practical implementation. For example, the dependency between tokens may introduce computational inefficiencies in scaling to larger datasets or more complex sequences, which could affect the generalization results. A more detailed exploration of how to handle these challenges efficiently during training would improve the practical applicability of the proposed framework.

Altough the paper is mostly theoretical, there is no enough experiments in the main paper. Especially the claim that topic prior in pre-training would affect the ICL does not get justification. Appendix contains some synthetic topic results using GPT-2 but that is a bit too hand-wavy.
The authors validate their theoretical claims with experiments on synthetic and real-world datasets, which is commendable, but the diversity of the real-world datasets could be expanded. For example, the selection of datasets appears limited to common language tasks like sentiment analysis and topic classification. Including a wider variety of tasks such as training on Zipfian data but test on distributions like MMLU and GSM8k would better demonstrate the generalization and applicability of the proposed AR-NTP paradigm across different domains. Moreover, while the results show that increasing the number of pre-training topics, sequences per topic, and sequence length improves model performance, the experiments would benefit from more ablation studies.

Lastly, while the paper provides theoretical insights into how ICL emerges from pre-training, the connection between theory and practice could be more tightly integrated. The authors propose that ICL arises from excellent generalization of sequences and topics, but the specific impact of pre-training data diversity, model size, and optimization iterations could be made clearer. The paper could explore more explicitly how different pre-training regimes—such as training on more complex or diverse datasets [1] —might affect ICL performance in practice. This would provide practitioners with clearer guidelines on how to optimize their pre-training procedures for better ICL outcomes. In particular, the paper could explore limitations of their generalization bounds when models are applied in low-resource environments, where generalization may suffer in no enough ICL examples are given or the pre-training FLOPs are constrained.

[1] Data Distributional Properties Drive Emergent In-Context Learning in Transformers Stephanie C.Y. Chan, Adam Santoro, Andrew K. Lampinen, Jane X. Wang, Aaditya Singh, Pierre H. Richemond, Jay McClelland, Felix Hill

**Questions:**

Why particularly choose PAC-Bayesian framework for ICL? There are numerous works study ICL using statistical frameworks like [1]
Why not based on uniform convergence or algorithmic stability, could help situate the results within a broader landscape of generalization theory?

[1] Transformers as statisticians: Provable in-context learning with in-context algorithm selection

---

> ### Author Response · Authors · 2024-11-19
>
> Thank you sincerely for your support and thoughtful suggestions! We are so delighted that you found our paper present an original contribution and open up new avenues for exploring token dependencies and generalization.
>
> With carefully considering your suggestions, ***we have expanded the discussion*** on the potential challenges of AR-NTP for practical implementation and some handling methods guided by our current theory, in addition with ***complemental experiments*** as much as you suggested. Below, we do our best to address your questions adequately!
>
> >**Q: One key weakness lies in the assumptions behind the AR-NTP paradigm. Although the shift from i.i.d. settings to token-dependent sequences is crucial, the paper could have provided a more thorough discussion on the potential challenges this poses for practical implementation [...] A more detailed exploration of how to handle these challenges efficiently during training would improve the practical applicability of the proposed framework.**
>
> **A:** Thanks for your thoughtful suggestions! We would like to present our further considerations one by one!
>
> **1. Potential challenges of AR-NTP for practical implementation.** We strongly agree that the AR-NTP paradigm brings some challenges for practical implementation. AR-NTP generates the next token step by step in an auto-regressive manner, which means that the model must handle the context of the current token at each step. Especially for long sequences, this leads to a significant increase in computational and memory overhead. As the sequence length increases, the complexity of computing dependencies grows exponentially, which can make the training process extremely slow and resource-intensive. Furthermore, long sequences bring long-range dependency issues for models like RNNs, also for multi-layer transformers, when processing extremely long sequences, gradients may explode as they propagate through multiple layers.
>
> **2. Possible handling methods of the challenges.** To address the challenges posed by long-range token dependencies, including optimization stability, practical storage and computational efficiency, we propose the following potential methods guided by our theroy:
>
> **(1) Optimizing sample order.** In ICL, the prompt sequence consists of several concatenated exemplars. When a large number of exemplars are provided, the sequence can become excessively long. By optimizing the sample order, we can enhance the training process and maintain model performance.
>   - **Our theory provides a theoretical understanding for this:** We model the token generation process using the conditional probability $\mathbb{P}(x_{t+1} \mid E_t, w_k)$, indicating that the prediction depends not only on the current input but also on the preceding context. During the model's processing of in-context examples, if the sequence abruptly switches from a sample closely related to the query to one with very low relevance, it introduces significant fluctuations. This forces the model to continually adjust its internal parameters to adapt to the new input, potentially causing increased gradient fluctuations and *training instability*.
>   - According to our theoretical analysis, specifically Assumption 4.5, **we use $\sigma$ to represent the upper bound of the expected gradient variance**. A large $\sigma$ implies uncontrolled gradient magnitude changes during updates, reflecting instability in the optimization process, which can subsequently degrade the model's generalization performance from our generalization bounds. Thus, **optimizing the sample order (e.g., reducing irrelevant samples in the context)** can stabilize convergence and better satisfy Assumption 4.5, i.e., with a reduced $\sigma$. This, in turn, results in a tighter generalization upper bound and improved model generalization.

---

> > ### Author Response · Authors · 2024-11-19
> >
> > **(2) Local Attention Mechanism.**
> > - Building on the above discussion regarding optimizing sample order, when samples more closely related to the query are grouped together (a well sample-order prompt sequence), the dependencies between adjacent tokens become stronger. It enables the design of a **local attention mechanism**, which focuses only on a fixed-length prefix sequence of the current token during computation. This approach reduces the memory and efficiency overhead associated with long-range token dependencies.
> >
> > - Although our theoretical analysis of token dependency assumes reasoning over the entire sequence of preceding tokens, the existing framework can be extended to handle scenarios where the prefix sequence $E_t$ in $\mathbb{P}(x_{t+1} \mid E_t, w_k)$ is constrained to a certain length, akin to an n-gram model. Furthermore, the theoretical techniques for addressing token-dependency remain applicable. It provides great inspiration for future work that we can indeed extend the current theory to computation-limited scenarios!
> >
> > Thanks again for your suggestion! We have complement this analysis to Appendix D in the revised version! Hope that could address your concerns!
> >
> > >**Q: Why particularly choose PAC-Bayesian framework for ICL? There are numerous works study ICL using statistical frameworks like [1] Why not based on uniform convergence or algorithmic stability, could help situate the results within a broader landscape of generalization theory?**
> >
> > **A:** Thanks for your question! Uniform convergence and algorithmic stability are indeed great techniques to analysis generalization. We specially choose PAC-Bayesian technique mainly considering the following reasons.
> >
> > **1. Which characterization is relatively more accurate for modeling language learning, as well as much easier to theoretically deal with token-dependency under the AR-NTP setting?**
> >
> >   - **The characterization with conditional probability in PAC-Bayesian.** With language tasks like text generation or question answering, the next-token prediction paradigm is generally adapted in current LLMs. This method carefully considers the dependency in tokens for better context understanding. Thus, it's necessary to characterize this AR-NTP paradigm, and conditional probability in statistic is indeed an useful technique, i.e., $\mathbb{P}(x_t \mid x_1, x_2, \cdots, x_{t-1})$ serves as our optimization objective. As introduced in "Summary of Challenges" in Section 4, constructing ghost sequences provides a good chance that enables taking expectation over each token within all sequences, under the case of token-dependency.
> >   - **Uniform Convergence.** From the perspective of problem modeling, uniform convergence theory typically relies on assumptions about the hypothesis class, especially requiring that the hypothesis class exhibits some "controllable" complexity. For large models, defining the hypothesis class or achieving good generalization performance even with high-complexity hypothesis classes often limits the practical guidance of conclusions derived from this method. Additionally, and more critically, uniform convergence assumes the training data is i.i.d. When dealing with sequential data, modeling these dependencies and deriving corresponding generalization bounds is theoretically challenging and often infeasible based on current knowledge on uniform convergence technique.
> >   - **Algorithmic Stability.** The algorithmic stability technique has been ultilized by Li et al.[1] which focuses on ensuring that small changes in training data do not lead to significant fluctuations in predictions. From this perspective, compared to uniform convergence, algorithmic stability at least considers data aspect. However, it seems to bypass the modeling of token dependencies by characterizing parameter changes caused by perturbing a single token. The traditional approach can be relatively easily extended to ICL scenarios. While this method is feasible for providing generalization bounds in AR-NTP settings, it appears to skip detailed modeling of token-dependency, making the characterization of NTP less precise and somewhat limited.

---

> > > ### Author Response · Authors · 2024-11-19
> > >
> > > **2. What insights can different generalization bounds provide?**
> > > - **Uniform Convergence.** The primary limitation of this method lies in its neglect of the optimization process inherent in the learning algorithm itself. For instance, the impact of optimization algorithms such as gradient descent on generalization ability is not adequately considered within this theoretical framework. The relevant theoretical results are typically constrained by the sample size and the hypothesis class.
> > >
> > > - **Algorithmic Stability.** The most notable feature of this method is its ability to establish a close connection between small perturbations during the optimization process and generalization. The theoretical results often include gradient constraint terms and provide insights into the effect of sample size.
> > >
> > > - **PAC-Bayesian.** Within our framework, starting from a statistical perspective, we can similarly derive the influence of sample size on generalization. Additionally, the Bayesian framework uniquely incorporates the KL divergence between the posterior and prior distributions of the model into the generalization upper bound, highlighting the impact of the prior on model generalization. We further consider data-dependent and topic-dependent priors, thus we can also introduce *continous optimization trajectories analysis* when detailing the KL term, giving similar optimization insights to algorithm stability. We here specially point out the KL term $D_{KL}(\mu \parallel \nu)$, it provides more insights on model training and data selection strategies. **These practical implications are introduced in Appendix D.**
> > >
> > > **3. How does our PAC-Bayesian framework differ from other statistical analysis?**
> > >
> > > As you mentioned, the ICL research by Bai et al. [2] indeed explores properties of transformers that it can implement a broad class of standard ML algorithms in context, just like statisticians. They view the ICL process as successful algorithm selection process, which is *a novel perspective to understand ICL process itself*. However, it's in the scope of **exploring what ICL does**, as we introduced in Introduction Section, yet **has some limitations in bridging pre-training and ICL to explain ICL emergence**. Our detailed established pre-training and ICL framework particularly speed up the analysis from the perspective of ICL emerges from good generalization.

---

> > > > ### Author Response · Authors · 2024-11-19
> > > >
> > > > >**Q: About Experiments--the topic prior in pre-training would affect the ICL does not get justification.**
> > > >
> > > > **A:** Thanks for your suggestions! We find this weakness and in this discussion period, **we further explore the impact of prior via a carefully designable experiment on the real-world language datasets**.  About **more practical insights** from the KL divergence between model posterior and prior in our generalization results, we have detailed discussion in Appendix D, in the revised paper! It indeed guides us to verify whether leveraging prior model initialization brings benefits to model training or performance. Specifically, consider the following setup: our training data consists of $K=20$ pre-training topics, $N=2^{14}$ training sequences per topic and sequence length $T=256$.
> > > >
> > > >   - **Step 1:** Train the GPT2-small model for 15,000 steps using $K=5$ pre-training topics, $N=2^{14}$ training sequences per topic and sequence length $T=256$.
> > > >   - **Step 2:** Transfer the weights from GPT2-small model to the corresponding weight matrices of GPT2-large, ensuring dimension compatibility. Initialize the weights randomly for the additional transformer layers in GPT2-large.
> > > >   - **Step 3:** Train the GPT2-large model for an additional 30,000 steps using the full pre-training data ($K=20$, $N=2^{14}$, $T=256$).
> > > >
> > > > According to our experimental results, the random initialization regime with all pre-training data requires nearly ***7 hours*** on four A100 GPUs to complete 30,000 steps. However, under the prior model initialization regime, where a smaller model is used for warmup and serves as the prior model initialization for the larger model, training the GPT2-large model takes only ***4 hours*** for 30,000 steps on four A100 GPUs under the same setting of $K,N,T$ (with 0.5 hours needed for training the GPT2-small model for 15,000 steps).
> > > >
> > > > Furthermore, as shown in the optimization loss curve in Figure 2(h), the prior model initialization not only accelerates training but also stablizes the training process (especially at the early stage), leading to comparable or even improved model performance. This approach demonstrates how effectively leveraging prior knowledge can contribute to the training process and performance, supporting the KL term in our generalization bounds and presenting more practical insights.
> > > >
> > > >
> > > > >**Q: About Experiments--Including a wider variety of tasks, such as multi-modal learning or tasks requiring reasoning over longer contexts, would better demonstrate the generalization [...] The experiments would benefit from more ablation studies.**
> > > >
> > > > **A:** Thanks for your great suggestions! We examine the existing expeiments and in this discussion period, refer to your suggestions, we design more refined ablation experiments to verify the impacts of $K,N,T$ and observe the possible limits of these parameters, as well as observing the optimization process. Concretely,
> > > > - **Datasets, Model and Hyperparameter Settings.** In the pre-training phase, we consider a mixture of a series of language tasks, mainly including 20 datasets. Classified by task types, including sentiment analysis (glue-sst2, poem_sentiment, yelp_polarity and emotion), linguistic analysis (glue-cola, blimp), text classification (ag_news, dbpedia_14, ethos), question answering (tweet_qa) and commonsense reasoning (swag). We especially add linguistic analysis, QA and reasoning tasks, compared to the original experiments. Different datasets are considered as different topics (reflected in $K$ from our framework). In ICL phase, we test ICL performance with different datasets. All the datasets are obtained from Huggingface. We train the GPT2-large model with a batch size of 16 and a learning rate of 1e-4 for total 30,000 iterations. All experiments are conducted using four 24GB NVIDIA GeForce RTX 3090 and 40GB A100 GPUs.
> > > > - Compared to the original experiments, **we add the verification of $K,N$.** Specifically, we empirically explore the separate effects of the number of topics ($K$), number of sequences per topic ($N$) and sequence length ($T$). By detailing $K \in \\{5,10,15,20\\}$, $N \in \\{2^{8}, 2^{10}, 2^{12}, 2^{14}\\}$, $T \in \\{48, 64, 128, 256\\}$, we arrange groups of comparative experiments to verify that increasing $K, N, T$ individually improves the model's generalization performance as demonstrated in our Theorems.
> > > >
> > > > **In total, the revised paper has been updated and we strongly encourage to read the "Experiments" section for more details!**

---

> > > > > ### Author Response · Authors · 2024-11-19
> > > > >
> > > > > **Q: The specific impact of pre-training data diversity, model size, and optimization iterations could be made clearer [...] In particular, the paper could explore limitations of their generalization bounds when models are applied in low-resource environments, where generalization may suffer in no enough ICL examples are given or the pre-training FLOPs are constrained.**
> > > > >
> > > > > **A:** Thanks for your thoughtful suggestions!
> > > > > - **Theoretically**, by analyzing continous optimization trajectories, we gain insights into optimization iterations and model size in the generalization bounds. This reveals that faster optimization and larger model indeed lead to better performance, in addition to the statistical insights regarding the number of pre-training topics, sequences and sequence length. Your suggestion to characterize the pre-training data diversity is an excellent research direction, we consider that techniques such as using statistical measures like the KL divergence $D_{KL}(P(w_i) \parallel P(w_j))$ to capture distribution differences across topics, extending the current analysis of topic distribution from the high level. We plan to this theoretical exploration for future work, building on our current framework.
> > > > > - **Empirically**, in this discussion period, we have conducted more experiments on real-world language datasets observing the optimization process, as introduced in Section 5 in the revised paper: In Figure 2(f), we present four training processes where varying $N \in \\{2^{8},2^{10},2^{12},2^{14}\\}$ and keeping $K=20$ and $T=256$ fixed. We observe that larger $N$ brings faster convergence in addition to better performance. Similarly, Figure 2(g) takes varied $T \in \\{48,64,128,256\\}$ and keeping $K=20$ and $T=256$. All these aligns with our Theorems that `trains faster, generalize better'. As for the resource-limited case, our additional experiments that using prior model initialization brings new insights, which provides accelerated training strategies to address resource constraints.
> > > > >
> > > > > In summary, we have supplemented more experiments to **Section 5 and Appendix C**, as well as relevanet practical implications discussed above to **Appendix D**. **The revised paper has been updated accordingly**! Once again, we sincerely thank you for the detailed review and constructive feedback! We hope that our revisions satisfactorily address your concerns and we welcome any further discussions that may be helpful to the evaluation!
> > > > >
> > > > > Reference:
> > > > >
> > > > > [1] Li, Y., Ildiz, M. E., Papailiopoulos, D., & Oymak, S. (2023, July). Transformers as algorithms: Generalization and stability in in-context learning.
> > > > >
> > > > > [2] Bai, Y., Chen, F., Wang, H., Xiong, C., & Mei, S. (2024). Transformers as statisticians: Provable in-context learning with in-context algorithm selection.

---

> > > > > > ### Comment · Reviewer_x2ai · 2024-11-23
> > > > > >
> > > > > > Thank you for updating the draft and take into account the review feedback. I am happy to raise my score as most of my concerns were addressed. I still think the empirical results should be made stronger. Especially there are too many works on the theory of ICL but less empirical studies on its origin. There is another related work that might be relevant - by controlling the distribution to be Zipfian the emergence of ICL is more likely to occur.
> > > > > >
> > > > > > [1] Data Distributional Properties Drive Emergent In-Context Learning in Transformers
> > > > > > Stephanie C.Y. Chan, Adam Santoro, Andrew K. Lampinen, Jane X. Wang, Aaditya Singh, Pierre H. Richemond, Jay McClelland, Felix Hill

---

> > > > > > > ### Author Response · Authors · 2024-11-23
> > > > > > > **Reply to Remaining Concern**
> > > > > > >
> > > > > > > Dear Reviewer x2ai,
> > > > > > >
> > > > > > > Thank you sincerely for your responses and support! We are greatly encouraged by your score improvement! However, we noticed that the updated score has not yet been reflected on the platform. **Could you kindly take a moment to confirm the final score on the platform?**
> > > > > > >
> > > > > > > We greatly enjoy the opportunity to explain our work to those interested and refine it based on thoughtful suggestions! In the following, we are glad to discuss and clarify your remaining concerns.
> > > > > > >
> > > > > > > > **Q:** For example, the selection of datasets appears limited to common language tasks like sentiment analysis and topic classification. Including a wider variety of tasks such as training on Zipfian data but test on distributions like MMLU and GSM8k would better demonstrate the generalization and applicability of the proposed AR-NTP paradigm across different domains.
> > > > > > >
> > > > > > > **A:** We have carefully reconsidered your suggestions! It is an excellent idea to validate our theory in a broader domain and to explore the origin of ICL empirically. However, we would like to point out the following, related to the work [1]:
> > > > > > >
> > > > > > > As introduced in question (a): **"How can we model language tasks with token dependency, going beyond the *i.i.d.* limitation?"** Here are some key points:
> > > > > > >
> > > > > > > **(1)** Previous ICL work has invested significant effort into characterizing training sequences containing $(x, y)$ pairs, aiming to study supervised function learning tasks. This approach precisely aligns with the *i.i.d.* limitation we mentioned, including the empirical work you referred to in [1]. In this setting, the training sequences are formatted as $\\{(x_1, y_1, x_2, y_2, \cdots, x_T, y_T)\\}_{i=1}^N$ where $N$ sequences are *i.i.d.*, and $(x_i, y_i)$ are *i.i.d.* as well.
> > > > > > >
> > > > > > > Regarding the "data distributional properties" or "Zipfian distribution" particularly considered in [1], this is indeed an empirical complement to the study of distribution shift in prior ICL theoretical research.
> > > > > > >
> > > > > > > **(2)** In contrast, when modeling **language** tasks, the auto-regressive next-token prediction (AR-NTP) training paradigm is widely adopted in modern LLMs. Natural language learning is indeed more challenging than supervised image classification, as reflected in the phenomenon of token-dependency in this paradigm. Specifically, in our ICL setting for language modeling tasks, the training data format is $\\{(x_1, x_2, \cdots, x_T)\\}_{i=1}^N$ where $N$ sequences are i.i.d., but tokens within each sequence are interdependent. Thus, the empirical validation in image domain is somewhat going beyond our current research focus.
> > > > > > >
> > > > > > > **In total,** in previous setting with $(x, y)$-pair-formatted training sequences, the impact of distributional shifts has been explored. However, the AR-NTP training paradigm is still in the early stages of theoretical research, as well as our work provides potential inspiration to this field.
> > > > > > >
> > > > > > > We greatly appreciate your suggestion, which is indeed an excellent direction! Building upon our current work, **exploring distributional properties in the AR-NTP paradigm (beyond the supervised learning framework studied in [1])** represents a highly promising avenue for future research. We hope that our work serves as an inspiring foundation for advancing emergent capabilities such as ICL and COT reasoning under the AR-NTP training paradigm.
> > > > > > >
> > > > > > > Thank you again for raising these suggestions and giving us a chance to clarify this. We sincerely hope that our responses satisfactorily addressed your concerns! Please let us know if you have any more questions!
> > > > > > >
> > > > > > >
> > > > > > > [1] Data Distributional Properties Drive Emergent In-Context Learning in Transformers Stephanie C.Y. Chan, Adam Santoro, Andrew K. Lampinen, Jane X. Wang, Aaditya Singh, Pierre H. Richemond, Jay McClelland, Felix Hill

---

> > > > > > > > ### Author Response · Authors · 2024-11-23
> > > > > > > > **We sincerely thank you for raising the score!**
> > > > > > > >
> > > > > > > > Thanks again for your valuable feedback and for recommending a higher score! We are greatly encouraged by your recognition of our work and its contributions. Your positive comments motivate us to continue refining and improving the manuscript!

---

### Official Review · Reviewer_ARXg · 2024-11-02

**Soundness:** 3
**Presentation:** 2
**Contribution:** 2
**Rating:** 5
**Confidence:** 3

**Summary:**

This paper gives a generalization bound of sequences for in-context learning in a language model under the auto-regressive next-token prediction paradigm. It models the practical AR-NTP paradigm for an In-context Learning Framework, considering the two-level expectation loss over sequence and topic. Then it presents Theorems for the generalization of sequences and topics, provides the upper bound of the two-level expected loss. Thus, it indicates that ICL emerges from number of topics, number of sequences and sequence length, and demonstrated by limited numerical experiments.

**Strengths:**

1. This paper considers the Auto-Regressive Next-Token Prediction paradigm, which models token dependencies more realistically in actual language models, addressing a limitation in previous ICL studies.
2. This paper presents data-dependent and topic-dependent PAC-Bayesian generalization bounds for auto-regressive next-token prediction loss.
3. This paper presents a theoretical view about ICL emergence, and demonstrates that ICL emerges from the generalization of sequences and topics.

**Weaknesses:**

1. Some motivation of the paper is confusing and not clarified enough. The author claims that the i.i.d. setting is unrealistic in language tasks, but Section 3 directly shifts the paradigm towards the language model without an explanation of how to break through the i.i.d. limitation.
2. The experimental setting is not fully discussed, including the detailed information about GINC datasets (number of entries; example; split;), the initialization of GPT-2 (from-scratch or pretrained), the method to change sequence length on real-world datasets (directly mask the token that exceeds the specified length, ignoring the semantic?)

**Questions:**

1. Can this result be directly adapted to the in-context learning paradigm in recent large language models? Since the difference in the empirical loss, is there still a gap between them?
2. Is the Bounded Gradient a strong assumption for the Transformer and recent large language model? After multiple layers of nonlinear superposition in the Transformer, the network output may be highly sensitive to input changes.
3. How do you define the topic examples in your experiments? Can you give some examples of the topic in linear dynamic systems and GINC?

---

> ### Author Response · Authors · 2024-11-19
>
> Thank you sincerely for your helpful comments! Regarding your concerns about confusion in some motivation, ***we infer that the statement of question (a) caused this regret***. To further address your concern, ***we have re-emphasized our research motivation about i.i.d., slightly modifying to the statement of question (a) to be more accurate and avoid misunderstandings.*** Additionally, as you suggested, we have detailed the experimental settings and moved GINC experiments to the main text. Meanwhile, we have conducted more new experiments on language datasets. **The revised paper has been updated accordingly**! Below, we do our best to address your questions adequately.
>
>
> >**Q1: Some motivation of the paper is confusing and not clarified enough. The author claims that the i.i.d. setting is unrealistic in language tasks, but Section 3 directly shifts the paradigm towards the language model without an explanation of how to break through the i.i.d. limitation.**
>
> **A:** Thanks for your thoughtful question very much! We would like to express our regret for the confusion regarding the statement of question (a). In the following, we would re-emphasize our research motivation about i.i.d. and make some modification.
>
> **Re-emphasize our research motivation about i.i.d.:** We emphasize that most of the existing ICL literature is based on the i.i.d. assumption (i.e., supervised learning tasks with prompts composed of i.i.d. $(x,y)$ pairs). This assumption is unrealistic, and the theoretical analysis techniques and results under this assumption are difficult to extend to more general language modeling tasks (where tokens are interdependent, i.e. $(x_1, x_2, ...)$, rather than $(x,y)$ pairs). Therefore, we propose "break through the i.i.d. limitation", which means we need to conduct research under a setting that considers token-dependency. In total, for question (a):
> - **Our core task is to model language tasks** with a systematic framework (which leads to question (b), where we ICL explore emergence within this new framework).
> - **The core challenge of modeling language tasks is to consider token-dependency**.
>
> As such, the explanation of this limitation in Line 47-50 of the original text is quite accurate. However, for question (a), the word "and" might have caused you to think that "breaking through the i.i.d. limitation" and "modeling language tasks" are two separate tasks. In fact, this is indeed one task. The original version of question (a) was intended to simultaneously emphasize that the core task -- modeling language tasks, and the core challenge -- without i.i.d. assumption.
>
> **Modification:** Your question and suggestion are excellent! Perhaps a simple modification to question (a) could reduce misunderstandings and more clearly express our research goal:
> - Original question (a)
>
> "How can we break through the i.i.d. limitation and shift toward modeling language tasks?"
>
> - Modified question (a):
>
> "How can we model language tasks with token-dependency, going beyond the i.i.d. limitation?"
>
> - Add clarification in Introduction Section:
>
> "Our core task is to model language tasks, and the core challenge of modeling language tasks is to consider prompt token-dependency."

---

> > ### Author Response · Authors · 2024-11-19
> >
> > >**Q2, Q5: The experimental setting is not fully discussed, including the detailed information about GINC datasets [...] How do you define the topic examples in your experiments? Can you give some examples of the topic in linear dynamic systems and GINC?**
> >
> > **A:** Thanks for your great consideration! In the revised paper, we have added detailed experimental settings and moved GINC experiments to the main text. Meanwhile, we have supplemented more insightful experiments on real-world language datasets, observing the optimization process and the potential effects of prior model initialization (guided by our therorems). **The revised paper has been updated and we strongly encourage to read the "Experiments" section for more details!** In the following, we would like to answer your questions one by one!
> >
> > **More experimental details about GINC.**
> > - **GINC Dataset.** GINC is a small-scale language dataset generated from a uniform mixture of Hidden Markov Models (HMMs) over a family of topics/concepts. The generation steps are as follows: *(1) Prepare transition matrix for HMMs:* The topic/concept determines the state transition matrix in HMM. For simulation, the transition matrix is randomly generated for each topic (each HMM), respectively; *(2) Prepare vocabulary:* The vocabulary is generated as combinations of letters starting from a' to z', aa' to az', and so on. We can obtain vocabularies with different sizes; *(3) Prepare memory matrix:* A unique matrix is created that records the mapping of vocabulary and state; *(4) Generate sequences:* Given a fixed topic and an initial state, generate the next state based on the transition matrix, and then obtain the observed token using the memory matrix. In total, each sequence is sampled from a random HMM in the family.
> >
> > - **Model and Hyperparameter Settings.** Our transformer model is based on the GPT-2 architecture with 4 layers, 12 attention heads, and 768-dimensional embeddings. Limited by computation resources and training data, we use "AutoModelForCausalLM.from_pretrained("gpt2")" to load a pretrained version from the HuggingFace model hub, not training from scratch, like most ICL research does. We train the model for 5 epochs using the AdamW optimizer with a batch size of 8 and a linear learning rate schedule. The schedule includes a warmup phase of 1000 steps, up to the learning rate of 8e-4. All experiments on GINC are conducted using a single 24GB NVIDIA GeForce RTX 3090.
> >
> > - We empirically explore the separate effects of the number of topics ($K$), number of sequences per topic ($N$), sequence length ($T$) and prompt length ($T_p$). We detail $K \in \\{10,20,30\\}$, $N \in \\{20,40,60,80,100\\}$, $T \in \\{1280, 2560, 5120, 10240\\}$, $T_p \in \\{8, 16, 32, 64\\}$, where ranging $T$ with directly masking the token that exceeds the specified length and do not taking special consideration. In totoal, we arrange groups of comparative experiments to verify that increasing $K, N, T, T_p$ individually improves the model's generalization performance as demonstrated in our Theorems. Additionally, we discuss the effect of vocabulary size and provide an interesting case involving with a failed ICL.
> > - More details are provided in Section 5 and Appendix C.
> >
> > **Topics in linear dynamic systems and GINC.**
> > - For linear dynamic systems, we provide a detailed discussion in Appendix C.1. We adopt numerical experiments to simulate the gold state equation $x_{t+1} = W x_t + \zeta_t$, where different $W \in \mathbb{R}^{d\times d}$ represent different tasks and different dimensions represent the varying difficulities of tasks. Thus, the topics indeed correspond to the randomly generated weight matrices $W$.
> > - For the sythetic dataset GINC, given the above introduction of its generating process, we would prepare transition matrix for each HMM. For simulation, the transition matrix is randomly generated for each topic (each HMM), respectively. Since the topic/concept uniquely determines the state transition matrix in HMM, we would use the transition matrix to represent the concept, rather than explicitly defining the concept.

---

> > > ### Author Response · Authors · 2024-11-19
> > >
> > > >**Q3: Can this result be directly adapted to the in-context learning paradigm in recent large language models? Since the difference in the empirical loss, is there still a gap between them?**
> > >
> > > **A:** Thanks for your insightful question! In our analysis, we emphasize that the training method of auto-regressive next-token prediction is critical. The pre-training and in-context learning framework presented in this paper is closely related to such auto-regressive generation, and is independent of specific model architectures. Therefore, for recent large language models, such as Llama series, where the loss function is also defined as maximum likelihood estimation, as in this paper, existing theoretical results can be directly applied. Surely, different model sizes, according to our theorem, will lead to varying generalization performance and differentiated ICL capabilities. For modern rapidly evolving large language models, the foundational training objective is generally consistent, but there may be more focus on other aspects of the model's performance, such as harmlessness, honesty, alignment (e.g., RLHF). These capabilities may require the development of new theories to explain, which would be an exciting research direction!
> > >
> > >
> > > >**Q4: Is the Bounded Gradient a strong assumption for the Transformer and recent large language model? After multiple layers of nonlinear superposition in the Transformer, the network output may be highly sensitive to input changes.**
> > >
> > > **A:** Thank you for your question! We would like to discuss in detail how this assumption of bounded gradient reasonable, including: theoretically research that prove Lipschitz constant of transformers, its alignment with practical training and its commonality in generalization and optimization theory.
> > >
> > > **Proved Lipschitz constant**: Recent studies [1-2] have theoretically provided a detailed study on the Lipschitz constant of self-attention in several practical scenarios (eg, with layer normalization), which can be rigorously upper-bounded. Their experiments on pretrained and randomly initialized BERT and GPT-2 also support the theoretical findings.
> > >
> > > **Align with practical training:** Despite the depth and nonlinear structure of transformers, which may make the gradients sensitive to input changes. However, in practical training, especially with large models, *various strategies are commonly employed to effectively avoid gradient explosion or vanishing issues and ensure the stability of the training process*. For example, gradient clipping [3] is a common technique used to prevent gradient explosion in the training of large neural networks. It ensures that the gradient norm does not exceed a predefined threshold at each training step. Modern optimization algorithms such as Adam [4] adjust the learning rates adaptively for each parameter, effectively smoothing gradient updates and helping to prevent large fluctuations in gradients. Regularization methods [5] such as L2 regularization and gradient penalties also help control the magnitude of gradients during training. By penalizing the gradient size, these techniques prevent gradients from growing too large, stabilizing the training process.
> > >
> > > **Common in generalization and optimization theory:** The Lipschitz continuity [6-9] assumption is commonly used in theoretical analyses to explain the generalization ability of models. Bounded gradients imply that the model will not suffer from instability due to extreme gradient fluctuations, which could otherwise harm its generalization ability. By excluding the influence of the gradient factor, it becomes possible to explore other factors that are more likely to affect the model's generalization.
> > >
> > > Once again, we sincerely thank you for the detailed review and constructive feedback! We hope that our revisions satisfactorily address your concerns and we welcome any further discussions that may be helpful to the evaluation.
> > >
> > >
> > > Reference:
> > >
> > > [1] Castin, V., Ablin, P., & Peyré, G. (2024). How Smooth Is Attention?. In ICML 2024.
> > >
> > > [2] Collins, L., Parulekar, A., Mokhtari, A., Sanghavi, S., & Shakkottai, S. (2024). In-context learning with transformers: Softmax attention adapts to function lipschitzness. arXiv preprint arXiv:2402.11639.
> > >
> > > [3] Mai, V. V., & Johansson, M. (2021, July). Stability and convergence of stochastic gradient clipping.
> > >
> > > [4] Kingma, D. P. (2014). Adam: A method for stochastic optimization.
> > >
> > > [5] Moradi, R., Berangi, R., & Minaei, B. (2020). A survey of regularization strategies for deep models.
> > >
> > > [6] Eriksson, K., Estep, D., Johnson, C., Eriksson, K., Estep, D., & Johnson, C. (2004). Lipschitz continuity.
> > >
> > > [7] Bousquet, O., & Elisseeff, A. (2002). Stability and generalization.
> > >
> > > [8] Zhou, Y., Liang, Y., & Zhang, H. (2022). Understanding generalization error of SGD in nonconvex optimization.
> > >
> > > [9] Li, Y., Ildiz, M. E., Papailiopoulos, D., & Oymak, S. (2023, July). Transformers as algorithms: Generalization and stability in in-context learning.

---

> ### Author Response · Authors · 2024-11-25
> **A Sincere Invitation to Further Discuss Our Revised Paper!**
>
> Dear Reviewer ARXg,
>
> Thanks again for your valuable feedback! We have carefully considered your suggestions and made corresponding revisions. **The revised paper has been updated accordingly!**
>
> We sincerely hope you will continue to engage in this discussion, if you have further questions or concerns, we are more than willing to provide additional clarifications or supporting materials! Your insights are crucial for refining our research!

---

> ### Comment · Reviewer_ARXg · 2024-11-26
>
> Thank you for your feedback. I still don't know why generalization performance leads to the emergence of ICL. In addition, you don't fully answer "Q4: Is the Bounded Gradient a strong assumption for the Transformer and recent large language model?". In a word, your work may be valuable. But I'm not sure about the significance and impact of your work.

---

> > ### Author Response · Authors · 2024-11-27
> > **Reply to Further Questions!**
> >
> > Dear Reviewer ARXg,
> >
> > Thanks for your further questions! We are more than willing to address all of your concerns! Regarding the assumption of bounded gradient, we find that **there exists a mismatch between our views**. For the connection of "generalization" and "ICL emergence", we further dissect the complete process from pre-training to ICL, including the role played by generalization performance metrics. Finally, we summarize the impact of our paper from research question, theoretical framework and techniques, and practical applications supported by theoretical and experimental evidence.
> >
> > > **Further clarification for Q4:** Is the Bounded Gradient a strong assumption for the Transformer and recent large language model? After multiple layers of nonlinear superposition in the Transformer, the network output may be highly sensitive to input changes.
> >
> > **A:** Thanks for your question! We have reconsidered your questions that ***there exists a mismatch*** between your concerns about "the network output being highly sensitive to input changes" and our assumption of bounded gradient. These correspond to ***the gradient over input data*** ($\nabla_x L(x; \theta)$) and ***the gradient over model parameters*** ($\nabla_\theta L(x; \theta)$), respectively. Below, we will provide a more detailed explanation!
> >
> > **1. The gradient over input data reflects model robustness.**
> >
> > The gradient over input data ($\nabla_x L(x; \theta)$) reflect the sensitivity of model outputs to input changes. As you mentioned, transformers are built on multiple layers of complex nonlinear transformations, causing large changes in the model's output. In practice, there are many methods to improve a model's stability to *input perturbations*, such as data augmentation and adversarial training. Theoretically, if the gradient over input data is bounded, the model would be robust to input noise, allowing it to generalize better to unseen data.
> >
> > However, we have not analyzed model generalization from the perspective of robustness. Instead, we consider optimization-dependent generalization bounds, which are closely linked to the optimization process. Our assumption, the boundedness of gradients over model parameters reflects the stability of the training process.
> >
> > **2. The bounded gradient over model parameters is reasonable.**
> >
> > Our bounded gradient assumption focuses on the gradient over model parameters, $\nabla_\theta \mathcal{L}(x;\theta)$, as we answered before, we mainly explain the reasonableness of the bounded gradient over model parameters. From both practical and theoretical perspectives:
> >
> > - **Practically**, gradient explosion is a common issue in deep learning optimization, leading to unstable convergence or failure to converge. To address this, many training strategies (e.g., gradient clipping and regularization) explicitly or implicitly enforce constraints on gradient magnitudes. This highlights that bounding parameter gradients is a widely adopted engineering practice. For LLMs which involve numerous parameters, controlling gradient magnitudes is especially critical. Bounded gradients on parameters ensure training stability and avoid numerical issues.
> >
> > - **Theoretically**, in the optimization process, especially when using gradient descent, assuming that the gradient is bounded helps ensure the convergence of the algorithm. If the gradient is too large, it can lead to excessively large parameter updates, causing instability in the training process and preventing convergence. When the gradient is bounded, it ensures that each update step is within a reasonable range, allowing the algorithm to gradually approach the optimal solution and ultimately achieve convergence.
> >
> > **In summary**, the boundedness of gradients over input data (as you considered) is more directly related to model robustness. However, our bounded gradient assumption pertains specifically to gradient over parameters and is justified by its role in ensuring optimization stability, when considering optimization-dependent generalization bounds.
> >
> > We hope the above effectively addresses the mismatch between our understandings and resolves your concerns!

---

> > > ### Author Response · Authors · 2024-11-27
> > > **Reply to Further Questions!**
> > >
> > > > **Q:** I still don't know why generalization performance leads to the emergence of ICL.
> > >
> > > **A:** Thanks for your question! Let’s dissect the complete process from pre-training to ICL, including the role played by generalization performance metrics:
> > >
> > > **First**, considering that we have a pre-trained LLM, trained on a vast amount of multi-task data [1]. We observe that it develops in-context learning (ICL) capabilities. This refers to the ability to predict ***new, unknown topics/tasks*** based on just a few examples in the ***prompt***, without requiring parameter adjustments [2].
> > >
> > > According to the basic definition of ICL, there are two key aspects:
> > > **(1) Predict based on prompt**.
> > > - This requires the model to possess the ability to handle new sequences (ICL prompts) as well as to master next-token prediction (thus can utilize in-context examples in the ICL prompt).
> > >
> > > - ***To characterize the model's ICL capability when encountering new sequences***, we use the expectation defined over sequences, which represents covering all sequences in the sample space. This is what we define as the **first-level expectation**: generalization of sequences, as introduced in Section 4.1. The expected loss with infinite training sequences can be bounded by the sum of empirical loss with finite training sequences and a small value, which is common to consider generalization of sequences. We aim to use as many training sequences $N$ as possible to cover the entire sequence space so that optimizing the empirical loss approximates optimizing the first-level expected loss. A smaller expected loss ensures good ICL capabilities when the model encounters new sequences under topics that are seen during pre-training.
> > >
> > > - This process covers **the learning of AR-NTP paradigm**, which ensures that the model performs predictions utilizing the in-context examples in ICL prompt: as introduced in Lines 258-263, each token is generated based on its prefix sequence. Due to the token-dependency, when considering the expectation over sequences, it is necessary to decompose it into two parts: expectation over each token when given prefix sequences $\mathbb{E}
> > > \_{x^{k,n}\_{t+1} \sim \mathbb{P}(\cdot \mid E^{k,n}\_t, w_k)}$ and expectation over prefix sequences $\mathbb{E}\_{E^{k,n}\_t}$. This dealment is indeed our key contribution, compared to other works studying ICL primarily on regression tasks!
> > >
> > > **(2) Predict on new, unknown ICL topics.**
> > > - The above, we only consider the sequence level. However, another crucial aspect is that ***we focus on the ICL ability when faced with new, unknown ICL topics***. Here, unknown ICL topics naturally include both *seen* and *unseen* during pre-training.
> > > - If the model is pretrained with sufficient sequences for each pre-training topic, it will perform well during the ICL phase when encountering topics already seen in pre-training. This is indeed considered in the first-level expectation: generalization of sequences.
> > > - Furthermore, if the model is pre-trained on enough topics under the assumption of topic distribution, the probability of encountering unseen pretraining topics during the ICL phase decreases. This leads to smaller population loss over the topic distribution, capturing the model’s generalization performance to unseen topics in an expected sense. This forms our **second-level expectation**: generalization of topics, as introduced in Section 4.2.
> > >
> > > **In summary**, from a macro perspective, our research involves two key aspects, accordingly:
> > > - Corresponding to our **Question (a)**, AR-NTP paradigm represents the internal challenge on ICL prompt to complete the ICL prediction (capturing token-dependency, which is covered in the partial first-level expected loss).
> > > - Corresponding to our **Question (b)**, we bridge pre-training and ICL phase, and then the generalization of topics and sequences leads to ICL emergence!
> > >
> > > Reference.
> > >
> > > [1] Radford, A. (2019). Language models are unsupervised multitask learners.
> > >
> > > [2] Brown, T. B. (2020). Language models are few-shot learners.

---

> > > > ### Author Response · Authors · 2024-11-27
> > > > **Reply to Further Questions!**
> > > >
> > > > > **Q:** I'm not sure about the significance and impact of your work.
> > > >
> > > > **A:** Thanks for your question! We would like to further summarize the significance of our research from the following three aspects!
> > > >
> > > > **(1) The research question we address.**
> > > > - **Related to question (a)**: We greatly appreciate that previous work assuming i.i.d. $(x_i, y_i)$ pairs within sequences to study supervised classification or regression problems, has been beneficial for initial explorations in ICL. *Their setup simplifies the analysis, as introducing dependencies between tokens would significantly increase the complexity of the proof* (as discussed in Introduction, there has already been a considerable amount of work). However, we should not limit our investigations to this i.i.d. assumption alone. **More realistic tasks**, such as language modeling with the AR-NTP paradigm, and **more realistic setup**, where $x_i$ is generated based on its preceding tokens, are crucial. Notably, there has been relatively little work focusing the NTP problem, making the motivation for our work quite compelling.
> > > > - **Related to question (b)**: Most literature analysis answers *what* ICL does but falls short in explaining *how* pre-trained LLMs can be good enough to emerge ICL ability as well as the impact of pre-training phase on ICL. Our research serves as a valuable complement to the existing ICL research.
> > > >
> > > > **(2) The theoretical framework and techniques we have developed.**
> > > > - **Pre-training and ICL framework:** We view the systematic pre-training and ICL framework as a significant contribution (summarized in Contributions), one that may be intuitive yet has not been formalized in such detail (establish the layer-wise structure of sequences and topics), compared to the previous work.
> > > > - **Technical contributions:** As "Summary of challenges" introduced in Section 4, under the setting of AR-NTP, we make great efforts to address the dependency between the current token and its preceding tokens by constructing ghost sequences (see the detailed construction in Appendix G.2.1, where we summarize the proof sketch), thereby enabling the possibility of taking expectation over each token within all possible sequences. In the following, the token-dependency introduces a crucial connection of negative logarithm likelihood, KL divergence and TV distance (which is important for the definition and upper bounds of population loss). Specifically, we begin by examining the primary optimization objective: negative logarithm likelihood. Naturally, this leads to a connection with KL divergence, thereby formalizing the expression of population loss. Furthermore, in addressing the aforementioned token-dependency, we establish connections between TV distance and the expectation over a single token when given its predecessors. Therefore, it's necessary to establish connections between the two key distribution metrics: TV distance and KL divergence (see in Lemma G.7), to obtain our final generalization error bounds. **The AR-NTP setup necessitates the establishment of the above series of connections, which are not considered in the previous ICL work.**
> > > >
> > > > **(3) The practical applications, supported by both theoretical and experimental evidence.**
> > > > - We study the emergence of ICL from the perspective of generalization (the reason that we have further clarified in the above answer). As a result, we present data-dependent, topic-dependent and optimization-dependent generalization bounds. These guide the selection of more training topics (training sequences and sequence length) to obtain a pre-trained model with excellent generalization, thus ICL emerges from the generalization. Meanwhile, in addition to guiding the training data, $N_{\text{param}}$ also suggests the effect of model parameter scales on ICL performance. The term $D_{KL} (\mu \parallel \nu)$ motivates more specific training strategies from the perspective of optimization, such as how prior model initialization brings benefits to training efficiency and stability (which has been experimentally verified).
> > > > - The practical implications discussed above have been complemented in Appendix D, as well as relevant experiments to Section 5 and Appendix C. We greatly encourage to read the relevant section in the revised paper!
> > > >
> > > >
> > > > Once again, thank you for your valuable comments and for giving us the opportunity to clarify! We sincerely hope that our responses have addressed all your concerns. **Please don’t hesitate to let us know if you have any additional questions or suggestions. Your insights are crucial for refining our research!**

---

> > > > > ### Author Response · Authors · 2024-12-02
> > > > > **Have our responses satisfactorily addressed your concerns?**
> > > > >
> > > > > Dear Reviewer ARXg,
> > > > >
> > > > > Thanks again for your valuable feedback! We have carefully answered your additional questions, **particularly clarifying the mismatch between our views on the bounded gradient assumption.**
> > > > >
> > > > > We sincerely hope that our responses have addressed all of your concerns! Please don’t hesitate to let us know if you have any additional questions or suggestions. **Your insights are crucial for refining our research!**

---

### Official Review · Reviewer_fg9c · 2024-11-02

**Soundness:** 3
**Presentation:** 2
**Contribution:** 2
**Rating:** 6
**Confidence:** 3

**Summary:**

- This paper seeks to provide a theoretical understanding of in-context learning for language models.
 - A simple topic model is assumed, where a set of topics are drawn independently and a sequence is generated given each topic; this is done both for the pretraining data as well as the in-context learning data; note that the pretraining and in-context learning topics are different in general, but are drawn i.i.d. from a shared topic distribution.
 - The main result is a PAC-Bayes bound that controls the log-loss on new in-context learning instances, as a function of the number of topics, number of sequences per topic, and sequence length. (There are actually two results - one conditioned on a single topic, and one that averages over topics).
 - A few experiments demonstrate that accuracy improves as the sequence length increases.

**Strengths:**

- Obtaining theoretical analysis of in-context learning is a worthwhile goal, given the general lack of rigorous theory in this area.
 - The actual result that's derived is non-trivial (analyzing both sample complexity and optimization); and sensible in the sense that the bound scales fairly cleanly with the typical quantities (sequence length, number of sequences, number of topics).

**Weaknesses:**

- Some of the claims/framing in the paper seemed a bit odd. For example, 'emergence' is highlighted a few times, but I don't think it's necessary to invoke that: the setting here is a much more conventional transfer learning + domain generalization.
 - The paper contrasts previous analyses's i.i.d. approach compared to the autoregressive one in this paper, but this paper still assumes that we have i.i.d. sequences (it's just that the tokens in each sequence are not independent, which is not remarkable).
 - The essence of in-context learning as it manifests in LLMs is that the pretraining data is meant to just be raw text from the Internet, and that the in-context data is supposed to be examples of some task. The surprising part of ICL is that the pretraining and in-context distributions are very different. The fact that this paper assumes the two are the similar I think misses the spirit of in-context learning.

**Questions:**

- It seems like Theorem 4.3 is optimization-dependent, but it seems like we are just analyzing the solution to Equation 1, so does the choice of optimization algorithm come in? Assumption 4.2 bounds the gradient by S, which I see appears in (6). But what is assumed on the optimization algorithm? How can we even guarantee that it converges to the global optimum?
 - Theorem 4.6: If you have a topic at ICL time that you didn't see at pretraining time, it seems like your loss could be arbitrarily bad (since I don't see any other distributional assumptions). How does this manifest in the bound? Indeed, I think this is the interesting part of ICL (which one can think of as a transfer learning problem).
 - In general, it seemed like a lot of the paper defines fairly basic things, and it's only at page 7 when things get interesting. But by that time, there's really no space in the paper to discuss the actual theorems (not to mention that the experiments are very abbreviated).
 - I'm not sure what insight to take away from the experiments (other than that increasing sequence length helps). It would have been nice to see some even more synthetic experiments that try to test the theory (how things actually scale with K, N, T).

---

> ### Author Response · Authors · 2024-11-19
>
> Thank you sincerely for your helpul comments! We are delighted that you found our theoretical results to be nontrival (with model size, optimization and sample complexity). To further address your concerns regarding the significance of token-dependency setting and the claim of "emergence", ***we have re-emphasized our research motivation about i.i.d. limitation and re-clarified why claim emergence of ICL, which cannot be simply answered by traditional transfer learning and domain generalization theory.*** Additionally, regarding the organization of the paper, as you suggested, we have removed some rebundant statements in Section 3 while retaining critical components, such as the framework establishment and the transition from empirical loss to population loss. Here, ***we position Section 3 and 4 as equally important.*** We view the systemetic modeling process as a significant contributuion (summarized in Contributions), one that may be intuitive yet has not been formalized in such detail compared to the previous work.
>
> Below, we do our best to address your questions adequately. **The revised paper has been updated accordingly**!
>
> >**Q2: The paper contrasts previous analyses's i.i.d. approach compared to the autoregressive one in this paper, but this paper still assumes that we have i.i.d. sequences (it's just that the tokens in each sequence are not independent, which is not remarkable).**
>
> **A:** Thank you for your question! We would like to clarify that the shifting from i.i.d. $(x_i,y_i)$ pairs in each sequence, to dependent tokens in each sequence, is greatly significant and widely recognized improvement in ICL theoretical analysis. Specifically, we provide more discussions on the i.i.d. assumption from traditional machine learning to current ICL research, and emphasize that the token-dependency setting introduces great proof difficulties, making our work a valuable contribution to this field.
>
> **(1) I.I.D. assumption from traditional machine learning to ICL research.** In traditional machine learning, training samples are commonly assumed to be i.i.d., a consensus widely accepted in the field [1-2]. When each sample is generated independently, it means that the feature and label of one sample do not influence the generation of another sample,simplifying the modeling process. With all samples come from the same probability distribution, it ensures that the training data is representative of the target distribution, allowing the model to generalize effectively to unseen data. In total, the i.i.d. assumption is often the foundational hypothesis used for estimation and testing, and also helps provide convenience for theoretical analysis, such as bounds on generalization error.
>
> Here, **the "i.i.d. training samples", in comparison to our training data, should correspond to "i.i.d. training sequences".** Thus, we have the following summary:
>
> - In traditional machine learning, supervised training data is $$\{(x_i, y_i)\}_{i=1}^N$$ where $N$ samples are i.i.d.
> - In previous ICL setting, study supervised function learning tasks, training data is $$\{(x_1, y_1, x_2, y_2,\cdots,x_T, y_T)\}_{i=1}^N$$ where $N$ sequences are i.i.d., $(x_i,y_i)$ are i.i.d..
> - In our ICL setting, study language modeling tasks, training data is $$\{(x_1, x_2, \cdots, x_T)\}_{i=1}^N$$ where $N$ sequences are i.i.d., tokens are interdependent.
>
> It's easy to find that, much previous ICL research has focused on constructing prompts/sequences with i.i.d. samples to study supervised classification or regression problems. The learning task and i.i.d. assumption in each sequence are somewhat limited, particularly in the current era of LLMs. In practice, we rarely ultilize LLMs solely to fit a regression function, using such $(x_1, y_1, x_2, y_2,\cdots,x_T, y_T)$ sequences.
>
> **We greatly appreciate that previous work assuming i.i.d. $(x_i, y_i)$ pairs within sequences has been beneficial for initial explorations in ICL.** This assumption simplifies the analysis, as introducing dependencies between tokens would significantly increase the complexity of the proof. Starting theoretical analysis with simpler setups is both reasonable and practical (as discussed in Introduction, there has already been a considerable amount of work).
>
> **However, we should not limit our investigations to this i.i.d. assumption alone.** More realistic tasks, such as language modeling with the next-token prediction (NTP) paradigm, and more realistic assumptions, where $x_i$ is generated based on its preceding tokens, are crucial. Notably, there has been relatively little work addressing the NTP problem, making the motivation for our work quite compelling.
>
> In summary, we emphasize that, in modern language modeling tasks, **the i.i.d. within sequences is more critical than the i.i.d. between sequences.**

---

> > ### Author Response · Authors · 2024-11-19
> >
> > **(2) Token-dependency setting brings great proof difficulties.** As **"Summary of challenges"** introduced in Section 4, under the setting of AR-NTP, we make great efforts to address the dependency between the current token and its preceding tokens by constructing ghost sequences (see the detailed construction in Appendix G.2.1, where we summarize the proof sketch), thereby enabling the possibility of taking expectation over each token within all possible sequences.
> >
> > In the following, **the token-dependency introduces a crucial connection of negative logarithm likelihood, KL divergence and TV distance** (which is important for the definition and upper bounds of population loss). Specifically, we begin by examining the primary optimization objective: negative logarithm likelihood. Naturally, this leads to a connection with KL divergence, thereby formalizing the expression of population loss. Furthermore, in addressing the aforementioned token-dependency, we establish connections between TV distance and the expectation over a single token when given its predecessors. Therefore, it's necessary to establish connections between the two key distribution metrics: TV distance and KL divergence (see in Lemma G.7), to obtain our final generalization error bounds. **The AR-NTP setup necessitates the establishment of the above series of connections, which are not considered in the previous ICL work.**

---

> > > ### Author Response · Authors · 2024-11-19
> > >
> > > >**Q1: Some of the claims/framing in the paper seemed a bit odd. For example, 'emergence' is highlighted a few times, but I don't think it's necessary to invoke that: the setting here is a much more conventional transfer learning + domain generalization.**
> > >
> > > **A:** Thanks for your thoughtful concern! We would like to further clarify our consideration here, and we sincerely hope to reach a consensus with you.
> > >
> > > **(1) ICL generalization analysis under AR-NTP cannot be entriely equivalent to domain generalization.** As we introduced in Related Work section (Section 2 and Appendix E), training LLMs to perform ICL can indeed be viewed as an approach for addressing the broader tasks of meta-learning or learning-to-learn, similar to the perspective on transfer learning that you mentioned.
> > >
> > > Drawing inspiration from the assumption of an unknown task distribution in meta-learning analysis, we establish a pre-training and ICL framework with topic/task distribution and data distribution, to describe the model’s generalization ability to new test prompts and unseen topics. However, it is worth emphasizing that our ICL generalization analysis under AR-NTP cannot be equivalent to domain generalization. In traditional domain generalization, training data entirely satisfies the i.i.d. assumption, which is quite different from our setting. Even though in recent ICL research, most assumes i.i.d. $(x_i, y_i)$ pairs in each sequence, if we were to conduct generalization analysis under this assumption, we would agree with your point that there would be no remarkable distinction compared to traditional analysis.
> > >
> > > However, our analysis is conducted under the unique setup of auto-regressive pre-trained LLMs. Overall, while the pre-training and ICL learning framework remains intuitive, considering token-dependency represents a critical improvement over traditional domain generalization. This new setting explored in ICL research is much more aligned with current LLM language learning tasks. Theoretically, it introduces significant challenges for analysis, as discussed in the above answers **"Token-dependency setting brings great proof difficulties"**.
> > >
> > > Delving deeper into our theoretical analysis, the notable difference is that we define a two-layer expectation with respect to topic and sequence. **The inner expectation over sequence is specially split into two parts due to prompt token-dependency:** one part is the expectation over token conditioned on its prefix sequence, and the second part is the expectation over the prefix sequence itself. **This consideration of token-dependency was not addressed in previous domain generalization work.**
> > >
> > >
> > > **(2) The claim of "emergence".**
> > > We would emphasize that the claim of "emergence" is tightly related to the second limitation compared to recent ICL research. As introduced in **Introduction section**, most literature analysis answers ***what*** ICL does but falls short in explaining ***how*** pre-trained LLMs can be good enough to emerge ICL ability as well as the impact of pre-training phase on ICL. This leads to our question (b): How can ICL emerge from pre-trained LLMs? Intuitively, for a well-pretrained LLM like ChatGPT, it can generate good responses to users' prompts, showing the model's ICL ability and excellent generalization performance. Therefore, it is reasonable to explain the origin of ICL from the perspective of measuring generalization error.
> > >
> > > In total, in relation to this question, the statement of "emergence" indicates that we are **focusing on the surprising ICL ability** of LLMs. Generalization serves as the tool to study the emergence of ICL and is the underlying reason for ICL emergence. Therefore, if we only emphasize domain generalization, our research motivation would not be accurately framed.

---

> > > > ### Author Response · Authors · 2024-11-19
> > > >
> > > > > **Q3, Q5: The essence of in-context learning as it [...] The surprising part of ICL is that the pretraining and in-context distributions are very different. The fact that this paper assumes the two are the similar [...] Theorem 4.6: If you have a topic at ICL time that you didn't see at pretraining time, it seems like your loss could be arbitrarily bad. How does this manifest in the bound?**
> > > >
> > > > **A:** Thank you for question! We would like to clarify more on the assumption of topic distribution and further some misunderstandings about our theorems. Sincerely hope to address your concerns!
> > > >
> > > > **(1) Assumption of Topic distribution.**
> > > > We would first mention a famous work by Xie et al. [3], they assert that both pre-training and ICL inference are potentially inferring some concept, and suggest that ICL fails to extrapolate to unseen concepts. Notably, their work is established on the extremely general cases (as you suggested that the pretraining and in-context distributions are very different). Thus, in the general case, from the observation by Xie et al. [3] and the basic meaning of generalization, **we think that we can reach such a consensus:** if there is no any relationship between pre-training and ICL, the model cannot complete the downstream tasks.
> > > >
> > > > Now return to our setting, we emphasize that there is topic distribution assumption over pre-training and ICL topics. As we introduced in Related work, we draw this inspiration from traditional meta-learning or transfer learning analysis, yet further extend it to NTP setting with token-dependency. Notice that we do not make any other assumptions on this distribution, such as it represents a strong or weak relation between pre-training and ICL topics. If the relation is strong, only a relatively small amount of training data is needed for the model to generalize well on ICL tasks. Here, no matter strong or weak relation, we only require as much training data as possible to ensure the model learns more knowledge, in order to handle unknown ICL topics (seen or unseen during pre-training).
> > > >
> > > > **(2) If the ICL topic is unseen during pre-training.**
> > > >
> > > > As for this question, it's necessary to re-understanding our Theorems that try to characterize. **In our theorem 4.3**, we first describe the first-level expected loss, defined on finite topics and infinite training sequences per topic, this loss decribes the ability to predict new sequences if ICL topics are seen topics during pre-training. If the first-level expected loss is small, it means that relatively sufficient knowledge for pre-training topics has been learned, so that completing the ICL topics although with new sequences (when ICL topics are seen during pre-training).
> > > >
> > > > **Further in theorem 4.6**, we further consider the second-level expectation as the final population loss defined by two-level expectations, with infinite topics and infinite sequences per topic. This population loss naturally represents the ability to handle any topic under a given topic distribution, including **both seen and unseen topics**. Due to the fact that in actual training, we cannot sample all topics from the topic distribution, only a finite number ($K$) of topics are sampled for model training. The limitations of finite topic samples lead to the first term of in the upper bound of population loss, $\sqrt{\frac{1}{KT_p}}$, which indicates that when we sample more topics (larger $K$) to better cover the topic space, the empirical loss based on the finite training data becomes very close to the population loss. This allows us to **characterize the model's performance across the entire topic distribution (indeed including any seen or unseen topics)**. Therefore, our theorem mainly uses the statistic $K$ to characterize the model's ability on new topics, including unseen topics, without needing to introduce additional symbols to specifically represent unseen topics. In total, it demonstrates that larger $K$, better generalization performance, and ICL emerges with more possibilities. We are sincerely hoping to reach a consensus with you!

---

> > > > > ### Author Response · Authors · 2024-11-19
> > > > >
> > > > > >**Q4: It seems like Theorem 4.3 is optimization-dependent, but it seems like we are just analyzing the solution to Equation 1, so does the choice of optimization algorithm come in? Assumption 4.2 bounds the gradient by S, which I see appears in (6). But what is assumed on the optimization algorithm? How can we even guarantee that it converges to the global optimum?**
> > > > >
> > > > > **A:** Thanks for your question! We would like to make more clarifications!
> > > > >
> > > > > **Manifestations about "optimization-dependent".**  We say that Theorem 4.3 is optimization-dependent, meaning that we consider the optimization process to give an approximation of $D_{KL}(\mu \parallel \nu_J)$, through continuous optimization trajectories analysis, rather than just analyzing the solution to Equation (1).
> > > > >
> > > > > In more detail, in Theorem 4.3, the main difference between the two equations is that equation (6) details the KL divergence between the posterior and prior distributions, particularly by considering a data-dependent prior. As introduced in section 4.1, this data-dependent prior provides a more informative starting point, leading to a posterior that is better aligned with the true data distribution. Overall, we can describe the generalization bounds with gradual refinement: from $D_{KL}(\mu \parallel \nu)$ with arbitary prior $\nu$, to $D_{KL}(\mu \parallel \nu_J)$ with data-dependent prior $\nu_J$, and then we have an approximation
> > > > > $$
> > > > > D_{KL}(\mu \parallel \nu_J) \approx \frac{L^2 C(\frac{1}{N_{param}}, T^\prime)}{N^\prime}.
> > > > > $$
> > > > > where $C(\frac{1}{N_{param}}, T^\prime) = \frac{\beta}{2} e^{8\beta S}(1-e^{-\frac{T^\prime}{exp(8\beta S)}})$ and $N^\prime$ is the size of the subset used to get a data-dependent prior. This approximation is based on the continuous analysis of optimization trajectories using SDE and Fokker-Planck Equation techniques, which explicitly reveal the impact of optimization process on the KL divergence and furthermore, on the model generalization. Specifically, we provide analysis based on the continuous form of classic optimization algorithm SGD, i.e., gradient langevin dynamics with noise. The impact of optimization algorithm is reflected in the gradient term, where we assume that the gradients are upper-bounded by $L$. When more data are used to obtain prior and adopt some training techniques such as gradient clipping or regularization to ensure more stable training, i.e., larger $N^\prime$ and smaller $L$, this leads to better generalization performance. Furthermore, $T^\prime$ in $C(\frac{1}{N_{param}}, T^\prime)$ also suggests that "train faster, generalize better", aligning with the famous conclusion by Hardt et al. [4].
> > > > >
> > > > > According to the above analysis, we consider the continuous dynamics of SGD -- gradient langevin dynamics with noise -- and assume that the gradients during training are bounded (stable). Aside from this, we have not made additional assumptions about the optimization algorithms.
> > > > >
> > > > > **Prior model initialization provides chances to better converge around the global minimum.** Randomly initialized parameters typically follow uniform or standard normal distributions, which lack any specific information about the data. In contrast, during pre-training, begin by a small-scale subset of data to train a prior model. The parameters of this prior model can then serve as an informative starting point for longer and more sufficient training with greatly-large-scale pre-training data. Compared to random initialization, prior model initialization brings a smaller $D_{KL}(\mu \parallel \nu)$, which means that this favorable initialization brings a stable training (with reduced gradient norm $\sigma$) and avoids exploring the entire parameter space (with fewer optimization iterations $T^\prime$). This aligns with our understandings that data patterns guides the model toward appropriate directions during training, reducing the likelihood of encountering unsuitable local minima or saddle points.

---

> > > > > > ### Author Response · Authors · 2024-11-19
> > > > > >
> > > > > > >**Q6, Q7: In general, it seemed like a lot of the paper defines fairly basic things [...] the experiments are very abbreviated [...] It would have been nice to see some even more synthetic experiments that try to test the theory (how things actually scale with K, N, T).**
> > > > > >
> > > > > > **A:** Thanks for your suggestion! Regarding the organization of the paper, we have removed some rebundant statements in Section 3, as you suggested. While, we retain critical components, such as the framework establishment and the transition from empirical loss to population loss. Here, ***we think that Section 3 and 4 are equally important.*** We view the systemetic modeling process as a significant contributuion (summarized in Contributions), one that may be intuitive yet has not been formalized in such detail compared to the previous work.
> > > > > >
> > > > > > Regarding experiments, as you suggested, we have moved some detailed experiments (on synthetic language dataset GINC) from Appendix C to the main text. Here, we briefly list the key experiments that attempt to test the impact of $K,N,T$ as you suggested, supporting our theory.
> > > > > > - **Numerical experiments on linear dynamic systems** (Old & New version: Appendix C.1).
> > > > > > - **Experiments on synthetic language dataset GINC** (Old version: Appendix C.2; New version: Section 5).
> > > > > > - **Experiments on real-world language datasets** (Old version: section 5; New version: Section 5 and Appendix C.3). Note that in this discussion period, we have conducted additional ablation experiments on $K,N$ and optimization process, as a complement to the original verificaition, which was only focused on sequence length $T$.
> > > > > >
> > > > > > Meanwhile, we have supplemented more insightful experiments on real-world language datasets, observing the optimization process and the potential effects of prior model initialization (guided by our therorems). **The revised paper has been updated and we strongly encourage to read the "Experiments" section for more details!**
> > > > > >
> > > > > > Once again, we sincerely thank you for the detailed review and constructive feedback! We hope that our revisions satisfactorily address your concerns and we welcome any further discussions that may be helpful to the evaluation.
> > > > > >
> > > > > >
> > > > > > Reference:
> > > > > >
> > > > > > [1] Bishop, C. M. (2006). Pattern recognition and machine learning.
> > > > > >
> > > > > > [2] Mohri, M. (2018). Foundations of machine learning.
> > > > > >
> > > > > > [3] Xie et al.(2021). An Explanation of In-context Learning as Implicit Bayesian Inference.
> > > > > >
> > > > > > [4] Hardt, M. (2016). Train faster, generalize better: Stability of stochastic gradient descent.

---

> > > > > > > ### Author Response · Authors · 2024-11-25
> > > > > > > **A Sincere Invitation to Further Discuss Our Revised Paper!**
> > > > > > >
> > > > > > > Dear Reviewer fg9c,
> > > > > > >
> > > > > > > Thanks again for your valuable feedback! We have carefully considered your suggestions and made corresponding revisions. **The revised paper has been updated accordingly!**
> > > > > > >
> > > > > > > We sincerely hope you will continue to engage in this discussion, if you have further questions or concerns, we are more than willing to provide additional clarifications or supporting materials! Your insights are crucial for refining our research!

---

> > > > > > > > ### Comment · Reviewer_fg9c · 2024-11-30
> > > > > > > > **response**
> > > > > > > >
> > > > > > > > Thank you for the detailed responses and updating the paper - this is very helpful.
> > > > > > > >
> > > > > > > > I agree that the dependent token case is interesting (and challenging).
> > > > > > > >
> > > > > > > > I still disagree with the need to use the term 'emergence', which is a vague non-technical term that gets thrown around in the era of LLMs, and I would prefer to keep everything more grounded to precise terms. I don't think we disagree over substance here, only the terminology.
> > > > > > > >
> > > > > > > > I am updating my rating from 5 to 6.

---

> > > > > > > > > ### Author Response · Authors · 2024-12-01
> > > > > > > > > **We sincerely thank you for raising the score!**
> > > > > > > > >
> > > > > > > > > Thank you sincerely for your valuable feedback and for recommending a higher score! We are greatly encouraged by your recognition of our work and its contributions. We are also delighted that we have no disagreements on the substance so far! Your positive comments motivate us to continue refining and improving the manuscript!

---

### Official Review · Reviewer_y9tb · 2024-11-08

**Soundness:** 3
**Presentation:** 3
**Contribution:** 3
**Rating:** 6
**Confidence:** 3

**Summary:**

This paper derives generalization bounds for in context learning using a topic-dependent and data-dependent view of pre-training,

**Strengths:**

- This paper looks at in-context learning through a new lens of generalizing over topics and sequences, which is a novel contribution.
- The generalization bounds arrived at in Theorem 4.3 are similar but tighter than the bound from [Li et al](https://proceedings.mlr.press/v202/li23l.html), which is valuable.
- The paper has a comprehensive appendix that readers will find helpful while grokking the paper.

Overall I think the paper has a new view that adds to recent work on the theory of in context learning.

**Weaknesses:**

- I am not entirely certain of the impact of this paper. Sure the bounds derived are new and tighter than previous similar ones in [Li et al](https://proceedings.mlr.press/v202/li23l.html), but I am not sure this contribution (alongside the view of modeling pre-training topics) is significant enough for ICLR. Perhaps a more fleshed out discussion section in the paper could remedy this, I am open to changing my opinion given a solid argument of why this result is significant compared to bounds from adjacent literature.

**Questions:**

- From what I can tell, the experiments in Appendix C.2 on GINC with varying number of topics are far more relevant than those in Section 5 which just show better ICL performance with more context length. Why did the authors choose to organize the paper in such a way? If there is a reason, I think the authors should highlight it, as it definitely stood out to me that the experiments in section 5 are common knowledge from existing in-context learning literature.

---

> ### Author Response · Authors · 2024-11-19
>
> Thank you sincerely for your support and insightful comments. We are delighted that you found our paper to be both novel and valuable. To further address your concerns regarding the impact of this paper, ***we have expanded the discussion to offer more practical guidance for model training, training data selection and deduplication, informed by our theoretical results.*** These insights are unique compared to previous ICL analysis and **serve as a complement to the practical guidance in Appendix D** where we have discussed number of training topics, sequences and sequence length. Below, we do our best to address your questions adequately.
>
> > **Q1:** I am not entirely certain of the impact of this paper. Sure the bounds derived are new and tighter than previous similar ones in Li et al, but I am not sure this contribution (alongside the view of modeling pre-training topics) is significant enough for ICLR. Perhaps a more fleshed out discussion section in the paper could remedy this, I am open to changing my opinion given a solid argument of why this result is significant compared to bounds from adjacent literature.
>
> **A:** Thanks for your question! Some practical guidance on the number of training topics, sequences and sequence length were provided in Appendix "Practical Implications" of the origin version. Here, we have expanded the discussion on the significance of this paper, primarily including ***practical guidance for model training, training data selection and deduplication*** from our theoretical results, as well as a ***comparision with relevant ICL research***.
>
> It's first worthy to emphasize that, in contrast to the algorithmic stability techniques adopted by Li et al., Bayesian analysis evaluates model performance from the perspective of probability distributions. Probability distributions have long been foundational in defining information entropy [1] and relative entropy (i.e., KL divergence), making them essential tools for measuring information. In our PAC-Bayesian generalization bounds, the key term $D_{KL}(\mu \parallel \nu)$ surely offers possibilities to quantify the information contained in the model and data, thereby providing practical guidance for model training, training data selection and deduplication.
>
> **1. Practical Guidance for Model Training.**
> We first summarize the model training principles informed by our theory:
>
> **(1) Prior Model Initialization:** Typically, randomly initialized parameters follow uniform or standard normal distributions, which lack any specific information about the data. In contrast, we begin with a small-scale subset of data to train a prior model during pre-training. The parameters of this prior model can then serve as an informative starting point for longer and more sufficient training with greatly-large-scale pre-training data. When using a data-dependent prior rather than random initializations, this results in a smaller $D_{KL}(\mu \parallel \nu)$, which in our theorems represents the distance between model posterior $\mu$ and prior $\nu$, contributing to a better generalization.
>
> Furthermore, a lower $D_{KL}(\mu \parallel \nu)$ also enhances the optimization, by detailing this term with continuous optimization trajectory analysis. Specifically for example in Theorem 4.6, when with topic-dependent prior $\nu_J$,
> $$
> D_{KL}(\mu \parallel \nu_J) \approx \frac{\sigma^2 C(\frac{1}{N_{param}}, T^\prime)}{K^\prime},
> $$
> where $C(\frac{1}{N_{param}}, T^\prime) = \frac{\beta}{2} e^{8\beta S}(1-e^{-\frac{T^\prime}{exp(8\beta S)}})$. A smaller $D_{KL}(\mu \parallel \nu)$ means that this favorable initialization brings a stable training (with reduced gradient norm $\sigma$) and avoids exploring the entire parameter space (with fewer optimization iterations $T^\prime$). This aligns with our understanding that data patterns guide the model toward appropriate directions during training, reducing the likelihood of encountering unsuitable local minima or saddle points.
>
> In total, using a data-dependent and topic-dependent prior for model initialization can significantly **improve training stability, model convergence, and generalization.** This approach is particularly useful in multi-task learning, where it helps establish relevant priors for each task in advance. Although employing more strategies to choose the subset $K^\prime$ can further refine the prior, excessive refinement may introduce new computational costs and efficiency trade-offs. We emphasize that even without careful data selection for prior model learning, a data-dependent prior generally outperforms random initialization. Particularly, when random initialization does not yield good performance, a data-dependent prior model may provide a new opportunity.

---

> > ### Author Response · Authors · 2024-11-19
> >
> > **(2) Using Small Model Training as Warm-up for Large Models:** The prior model initialization strategy discussed above considers training the model once in advance with the same architecture as the formal training. This approach can be further extended to provide insights for training large models.
> >
> > Specifically, prior knowledge can be acquired by first training a relatively smaller model with a different architecture. It enables effective initial parameters at a lower computational cost, providing a solid foundation for larger models and avoiding the instability and non-convergence issues that may arise from random initialization. The detailed analysis of $D_{KL}(\mu \parallel \nu)$ presented earlier serves as the theoretical understanding for the "small model warm-up" strategy. Furthermore, this approach has been successfully applied in engineering practices, including AutoML [2], Neural Architecture Search (NAS) [3] and current LLMs training.
> >
> >
> > **(3) Gradual Expansion of Training Data:** The strategy of expanding the training data involves beginning training with a small subset and gradually increasing the dataset size. In this process, the model's initial learning can be seen as based on a "data-dependent prior", and each expansion of the training data can be understood as the injection of a new model prior. Based on the analysis of $D_{KL}(\mu \parallel \nu)$ above, gradual expansion of training data similarly leads to improved generalization, faster convergence, and better handling of complex features. This guiding principle is also reflected in Curriculum Learning [4] and Progressive Networks [5].
> >
> >
> > **2. Practical Guidance for Training Data Selection and Deduplication.**
> > It is well-known that the vast amount of data obtained from the internet serves as input for compressing world knowledge into LLMs [6]. In the redundant data, high-quality data determines the upper limit of the performance of LLMs. Therefore, considering a data-dependent pre-training and ICL generalization framework has immense potential for guiding data. In our theory, to explicitly show the impact of data, we adopt a data-dependent and topic-dependent prior $\nu_J$ and further detail $D_{KL}(\mu \parallel \nu)$ with optimization analysis. We have discussed this in detail before: in ***Practical Guidance for Model Training*** part, we emphasize the advantages of prior model initialization over random initialization in model training. Here, we aim to further explore its guidance for training data **from the perspective of compression**.
> >
> > Specifically, we can select a subset of size $K^\prime$ from the $K$ pre-training topics to estimate a prior distribution. If a smaller $K^\prime$ can estimate a prior that is very close to the posterior distribution, it indicates that the information from the $K$ topics can actually be compressed into a smaller subset of $K^\prime$ topics. This reflects the compressibility of the data, and can thus **backward guide pre-training data** to further undergo data selection and deduplication, such as through topic clustering, data diversity, or information gain metrics (e.g., $D_{KL}(\nu(D) \parallel \nu(D_i))$, if this value is small, the data block $D_i$ is considered redundant and can be reduced in weight or removed to decrease the model's reliance on redundant information.) The reprocessed pretraining data may exclude some noise interference, further improving model performance, saving computational resources, and facilitating training for new models.
> >
> > **In summary:**
> > Based on the analysis above, the bounds derived by Li et al.[7], based on algorithmic stability techiniques, mainly reveal the impact of multi-task learning and sequence length. Zhang et al.[8] approach the problem by considering model distribution, however their results involve uniform priors, which are similar to random initialization, relatively having limitations in practical guidance. In contrast, our Bayesian approach characterizes model performance from a statistical perspective. In addition to the statistical insights regarding the number of pre-training topics, sequences and sequence length, the data-dependent prior also provides practical guidance on model training, data selection and deduplication, which is a unique contribution.
> >
> > **The above analysis of practical impacts has been supplemented in Appendix D. The revised paper has been updated accordingly!**
> >
> > Reference:
> >
> > [1] Shannon, C. E. (1948)A mathematical theory of communication.
> >
> > [2] He, X. (2021)AutoML:A survey of the state-of-the-art.
> >
> > [3] Elsken, T. (2019)Neural architecture search: A survey.
> >
> > [4] Bengio, Y.(2009)Curriculum learning.
> >
> > [5] Rusu, A. A. (2016)Progressive Neural Networks.
> >
> > [6] Delétang, G.(2023)Language modeling is compression.
> >
> > [7] Li, Y. (2023)Transformers as algorithms: Generalization and stability in in-context learning.
> >
> > [8] Zhang, Y. (2023)What and how does in-context learning learn? bayesian model averaging, parameterization, and generalization.

---

> > > ### Author Response · Authors · 2024-11-19
> > >
> > > > **Q2:** From what I can tell, the experiments in Appendix C.2 on GINC with varying number of topics are far more relevant than [...] the authors should highlight it, as it definitely stood out to me that the experiments in section 5 are common knowledge from existing in-context learning literature.
> > >
> > > **A:** Thanks for your suggestion! We strongly agree that experiments on GINC are interesting and reflect our theoretical results. Additionally, we have supplemented more insightful experiments on real language datasets with various learning tasks, observing the optimization process and the potential effects of **prior model initialization** (as we mentioned before, it provides guidance for practical implications). In light of this, as well as combining with other reviewers' opinions, we have increased the proportion of the "Experiments" section in the main text. **The revised paper has been updated and we strongly encourage to read the **"Experiments"** section for more details!**
> > >
> > >
> > > Once again, we sincerely thank you for the detailed review and constructive feedback! We hope that our revisions satisfactorily address your concerns and we welcome any further discussions that may be helpful to the evaluation.

---

> > > > ### Comment · Reviewer_y9tb · 2024-11-23
> > > > **Response**
> > > >
> > > > Thanks for your response. After reading your comment, I am keeping my recommendation the same.

---

> > > > > ### Author Response · Authors · 2024-11-25
> > > > > **We sincerely thank you for your valuable comments!**
> > > > >
> > > > > Thanks again for your thoughtful feedback! Your valuable comments motivate us to continue refining and improving the manuscript!

---

### Meta-Review · Area_Chair_UqVp · 2024-12-24

**Metareview:**

The paper introduces a theoretical framework for analyzing in-context learning (ICL) in pre-trained autoregressive next-token prediction language models. It assumes a generative process in which topics are sampled, and sequences are generated for each topic. This assumption applies both to the training data and to ICL. Consequently, ICL examples in this framework consist of sequences sampled from this process, rather than pairs of (input, output) examples, as is typical in the standard ICL setting. The authors refer to the latter as supervised ICL and their approach as unsupervised ICL. The model assumes that sequence distributions are independent and identically distributed (i.i.d.), while the tokens within each sequence are interdependent. This contrasts with the standard ICL setting, where ICL examples are generally considered independent. The theoretical results include PAC-Bayes bounds on the log-loss for new input sequences. The first bound addresses the case of a single or finite number of training topics, while the second considers the expectation over topics. These bounds depend on the number of topics, sequences, and sequence lengths. Experiments conducted on synthetic and real-world corpora analyze the influence of these factors, as well as the impact of training convergence. The findings confirm that increasing the number of sequences, their length, and the number of topics improves generalization performance. This aligns well with the theoretical results, though the observation that more data leads to better performance is not particularly surprising and somewhat trivial.


The paper elicited significant comments and questions from the reviewers, focusing on two main concerns: (i) the need for clarification of the theoretical framework and results, and (ii) the practical insights derived from the work beyond the straightforward observation that more samples improve performance, as well as questions about why ICL emerges. The authors responded extensively, providing detailed clarifications and explanations regarding their assumptions and theoretical implications. They also added new appendix sections to emphasize the potential practical relevance of their findings, significantly enhancing the paper’s content. These additions addressed many initial misunderstandings, particularly regarding the importance of the assumptions and results. However, the second concern remains unresolved, leaving open the question of the broader impact and practical insights gained from the experiments. Nonetheless, as the paper introduces a new direction and novel results for the theoretical investigation of ICL, potentially laying the foundation for further practical studies, I recommend acceptance.

**Additional Comments On Reviewer Discussion:**

There were extensive discussions and lengthy responses from the authors during the rebuttal. This allowed them to clarify the description of the theoretical part of the paper, while leaving the question of its practical impact largely open.

---

### Decision · Program_Chairs · 2025-01-22

Accept (Poster)